



# Solar Forcing for CMIP6 (v3.1)

Katja Matthes[1,2], Bernd Funke[3], Monika E. Andersson[18], Luke Barnard[4], Jürg Beer[5],
Paul Charbonneau[6], Mark A. Clilverd[7], Thierry Dudok de Wit[8], Margit Haberreiter[9], Aaron Hendry[14],
Charles H. Jackman[10], Matthieu Kretzschmar[8], Tim Kruschke[1], Markus Kunze[11], Ulrike Langematz[11],
Daniel R. Marsh[19], Amanda Maycock[12], Stergios Misios[13], Craig J. Rodger[14], Adam A. Scaife[15],
Annika Seppälä[18], Ming Shangguan[1], Miriam Sinnhuber[16], Kleareti Tourpali[13], Ilya Usoskin[17],
Max van de Kamp[18], Pekka T. Verronen[18], and Stefan Versick[16]

[1]GEOMAR Helmholtz Centre for Ocean Research Kiel, Kiel, Germany
[2]Christian-Albrechts Universität zu Kiel, Kiel, Germany
[3]Instituto de Astrofísica de Andalucía (CSIC), Granada, Spain
[4]University of Reading, Reading, United Kingdom
[5]EAWAG, Dübendorf, Switzerland
[6]University of Montreal, Canada
[7]British Antarctic Survey (NERC), Cambridge, UK
[8]LPC2E, CNRS and University of Orléans, France
[9]Physikalisch-Meteorologisches Observatorium Davos/World Radiation Center, Davos, Switzerland
[10]Emeritus, NASA Goddard Space Flight Center, Greenbelt, MD, U.S.A.
[11]Freie Universität Berlin, Berlin, Germany
[12]University of Leeds, Leeds, UK
[13]Laboratory of Atmospheric Physics, Aristotle University of Thessaloniki, Thessaloniki, Greece
[14]Department of Physics, University of Otago, Dunedin, New Zealand
[15]Met Office Hadley Centre, Fitz Roy Road, Exeter, Devon, UK
[16]Karlsruhe Institute of Technology, Karlsruhe, Germany
[17]ReSoLVE Centre of Excellence and Sodankylä Geophysical Observatory, University of Oulu, Finland
[18]Finnish Meteorological Institute, Helsinki, Finland
[19]National Center for Atmospheric Research, Boulder, CO, USA

*Correspondence to:* Katja Matthes (kmatthes@geomar.de)

**Abstract.** This paper describes the solar forcing dataset for CMIP6 and highlights in particular changes with respect to the CMIP5 recommendation. The solar forcing is provided for radiative properties, i.e., total solar irradiance (TSI) and solar spectral irradiance (SSI), and F10.7cm radio flux, as well as particle forcing, i.e., geomagnetic indices Ap and Kp, and ionisation rates to account for effects of solar protons, electrons and galactic cosmic rays. This is the first time that a recommendation

5   for solar-driven particle forcing is provided for a CMIP exercise. The solar forcing dataset is provided at daily and monthly resolution separately for the CMIP6 Historical Simulation (1850–2014), for the future (2015–2300), including an additional extreme Maunder Minimum-like sensitivity scenario, as well as for a constant and a time-varying forcing for the preindustrial control simulation. The paper not only describes the forcing dataset, but also provides detailed recommendations for how to implement the different forcing components in climate models.

10      The TSI and SSI time series are defined as averages of two (semi-) empirical solar irradiance models, namely the NRLTSI2/ NRLSSI2 and SATIRE-TS. A new and lower TSI value is recommended: the contemporary solar cycle-average is now



1361.0 W/m$^2$. The slight negative trend in TSI during the last three solar cycles in CMIP6 is statistically indistinguishable from available observations and only leads to a small global radiative forcing of -0.04 W/m$^2$. In the 200–400 nm range, which is also important for ozone photochemistry, CMIP6 shows a larger solar cycle variability contribution to TSI than CMIP5 (50% as compared to 35%).

The CMIP6 dataset is tested and compared to its CMIP5 predecessor using timeslice experiments of two chemistry-climate models and a reference radiative transfer model. The changes in the background SSI in the CMIP6 dataset, as compared to CMIP5, impact on climatological stratospheric conditions (lower shortwave heating rates (-0.35 K/day at the stratopause), cooler stratospheric temperatures (-1.5 K in the upper stratosphere), lower ozone abundances in the lower stratosphere (-3%), and higher ozone abundances (+1.5% in the upper stratosphere and lower mesosphere). Between the maximum and minimum

phases of the 11-year solar cycle, there is an increase in shortwave heating rates (+0.2 K/day at the stratopause), temperatures (∼1 K at the stratopause), and ozone (+2.5% in the upper stratosphere) in the tropical upper stratosphere using the CMIP6 forcing dataset. This solar cycle response is slightly larger, but not statistically significantly different from that for the CMIP5 forcing dataset.

    CMIP6 models with a well-resolved shortwave radiation scheme are encouraged to use SSI, as well as solar-induced ozone

signals, in order to better represent solar climate variability compared to models that only prescribe TSI and/or exclude the solar-ozone response. Monthly mean solar-induced ozone variations will also be incorporated into the CCMI CMIP6 Ozone Database for climate models that do not calculate ozone interactively. CMIP6 models with interactive chemistry are encouraged to use the particle forcing which will allow the potential long-term effect of particles to be addressed for the first time. The consideration of particle forcing has been shown to significantly improve the representation of reactive nitrogen and ozone

variability in the polar middle atmosphere, eventually resulting in further improvements of the representation of solar climate variability.

## 1   Introduction

Solar variability affects the Earth's atmosphere in numerous, and often intricate ways through changes in the radiative and energetic particle forcing (Lilensten et al., 2015). For many years, the role of the Sun in climate model simulations was reduced

to its sole total radiative output, named Total Solar Irradiance (TSI), and this situation prevailed in the assessment reports of the IPCC until 2007 (Alley et al., 2007). However, there has been growing evidence for other aspects of solar variability to be major players for climate, in particular Solar Spectral Irradiance (SSI) variations and, more recently, Energetic Particle Precipitation (EPP).

    For about a decade, studies involving stratospheric resolving (chemistry) climate models have included SSI variations (e.g.,

Haigh, 1996; Matthes et al., 2003, 2006; Austin et al., 2008; Gray et al., 2010). Whereas relative TSI variations in the 11-year solar cycle are small, about 0.1%, SSI changes are wavelength dependent, and may vary by up to 10% at 200 nm in the ultraviolet (UV) wavelength range (Lean, 1997). Variations in UV radiation over the solar cycle have significant impacts on the radiative heating and ozone budget of the middle atmosphere (Haigh, 1994).



Through dynamical feedback mechanisms solar forcing can also influence the lower atmosphere and the ocean (e.g., Gray et al., 2010). Therefore, its importance in particular for regional climate variability is becoming increasingly evident (e.g., Gray et al., 2010; Seppälä et al., 2014). Together with volcanic activity, solar variability is an important external source of natural climate variability. Because of its prominent 11-year cycle, solar variability may offer a degree of predictability for regional
climate and could therefore help reduce uncertainties in decadal climate predictions.

However, there are still uncertainties in the observed atmospheric signals of solar variability (Mitchell et al., 2015a) and its transfer mechanism(s) to the surface. Proposed transfer mechanisms include changes in TSI and SSI, as well as in solar-driven energetic particles (e.g., Seppälä et al., 2014). In addition, recent work suggests a lagged response in the North Atlantic/European sector due to atmosphere-ocean coupling (e.g., Gray et al., 2013; Scaife et al., 2013), as well as a synchro-
nisation of decadal variability in the North Atlantic Oscillation (NAO) by the solar cycle (Thieblemont et al., 2015). Lagged responses have been also attributed to particle effects (Seppälä and Clilverd, 2014) and hence the observed solar surface signal could be a combination of top-down solar UV and particle as well as bottom-up atmosphere-ocean mechanisms.

Since some of the climate models run under the previous fifth Coupled Model Intercomparison Project (CMIP5) included the stratosphere and the mesosphere for the first time, and were thus able to capture the so-called "top-down" mechanism
for solar-climate coupling, both TSI and SSI variations were recommended by the WCRP/SPARC SOLARIS-HEPPA activity (http://solarisheppa.geomar.de/cmip5). Recent modeling efforts have made progress in defining the pre-requisites to simulate solar influence on regional climate more realistically (e.g., Gray et al., 2013; Scaife et al., 2013; Thieblemont et al., 2015), but the lessons learned from CMIP5 show that a more process based analysis of climate models within CMIP6 is required to better understand the differences in model responses to solar forcing (e.g., Mitchell et al., 2015b; Misios et al., 2016; Hood et al.,
2015). In particular, the role of solar-induced ozone changes and the need for a suitable resolution of climate model radiation schemes to capture SSI variations is becoming increasingly evident, and will be touched upon in this paper. In addition we will for the first time provide the solar-driven energetic particle forcing together and consistent with the radiative forcing.

The quantitative assessment of radiative solar forcing has been systematically hampered so far by the large uncertainties and the instrumental artifacts that plague TSI and SSI observations (e.g., Ermolli et al., 2013; Solanki et al., 2013), and also by
the sparsity of the observations, which started in the late 1970s with the satellite era. These problems have deprived us of the hindsight that is needed to properly assess variations on time scales that are relevant for climate studies. Another issue is the uncertainty regarding their absolute level. Since CMIP5, the nominal TSI has been reduced to $1361.0 \pm 0.5$ W/m$^2$ (Mamajek et al., 2015). This adjustment has inevitable implications for understanding the Earth's radiation budget.

On multi-decadal time scales, proxy reconstructions of the TSI exhibit occasional phases of unusually low or high activity,
which are respectively called grand solar minima and maxima (Usoskin et al., 2014). Of particular interest in this regard is the future evolution of long-term solar activity. There is growing evidence for the Sun to enter a phase of low activity near 2050, after a grand maximum that peaked during the 20th century. However, how deep and how long this new phase of low solar activity is likely to be is still uncertain (Abreu et al., 2008; Barnard et al., 2011; Steinhilber et al., 2012). Recent studies have investigated the climate impacts of a large reduction in solar forcing over the 21st century, revealing only a small impact
on a global scale (Feulner and Rahmstorf, 2010; Anet et al., 2013; Meehl et al., 2013). However, a systematic assessment of



the regional impacts of a more realistic future solar forcing is still to be done. For example, on regional scales, a future grand solar minimum could potentially reduce Arctic amplification (Chiodo et al., 2016) and reduce long-term warming trends over western Europe (Ineson et al., 2015).

The above-mentioned uncertainty in the SSI is particularly challenging in the UV band (Ermolli et al., 2013). All climate
model intercomparison studies relied so far on the NRLSSI1 dataset (Lean, 2000). However, it is becoming increasingly evident that its solar cycle variability in the UV part of the spectrum may be too low as compared to updated and more recent SSI reconstructions by models such as NRLSSI2 (Coddington et al., 2015) and SATIRE-TS (Yeo et al., 2014). Recent studies have emphasized the sensitivity to UV forcing changes due to top down effects (Ermolli et al., 2013; Langematz et al., 2013; Ineson et al., 2015; Thieblemont et al., 2015; Maycock et al., 2015; Ball et al., 2016), thereby stressing the need for a state-of-
the-art representation of the SSI, and in particular the UV band, in the CMIP6 solar forcing recommendation. For that reason, we will in particular focus on the SSI uncertainty and possible impacts of the higher SSI variability in CMIP6 with respect to the CMIP5 solar forcing recommendation.

Analysis of model simulations and observations have shown a response of global surface temperature to TSI variations over the 11-year solar cycle of about 0.1 K (Lean and Rind, 2008; Misios et al., 2016). However, the observed lag and the spatial
pattern of the solar cycle response are poorly represented in CMIP5 models (e.g., Mitchell et al., 2015b; Misios et al., 2016; Hood et al., 2015). In addition, (Gray et al., 2010) report that previous long-term variations in solar forcing used in some experiments (Alley et al., 2007) may be too weak due to an unfortunate choice of epoch (around 1750) for the preindustrial solar forcing, as this was a period of relatively high solar activity.

More recently, it has become better established that there is a solar response in the Arctic Oscillation (AO) and NAO from
the top-down mechanism (Shindell et al., 2001; Kodera, 2002; Matthes et al., 2006; Woollings et al., 2010; Lockwood et al., 2019; Ineson et al., 2011; Langematz et al., 2013; Ineson et al., 2015; Maycock et al., 2015; Thieblemont et al., 2015). Earlier models often employed a lower vertical domain, missing key physical processes by which solar signals in the stratosphere couple to surface winter climate. However, some of the more recent studies using stratosphere resolving (chemistry) climate models confirm a stratospheric downward influence on the NAO from solar variability, in particular associated with changes
in UV radiation and possibly through interaction with stratospheric ozone (e.g., Matthes et al., 2006; Rind et al., 2008; Ineson et al., 2011; Chiodo et al., 2012; Langematz et al., 2013; Thieblemont et al., 2015; Ineson et al., 2015). Some of these studies also suggest weaker model responses than are apparent in observations, although with large uncertainty (e.g., Gray et al., 2013; Scaife et al., 2013).

Another very important solar forcing mechanism after electromagnetic radiation is energetic particle precipitation (Gray
et al., 2010; Lilensten et al., 2015). Although the impact of EPP on the atmosphere is well documented, it had been ignored in solar forcing recommendations for earlier phases of CMIP. The term EPP encompasses particles with very different origins: solar, magnetospheric, and from beyond the solar system. These particles are mainly protons and electrons, and occasionally $\alpha$–particles, and heavier ions.

Solar protons with energies of 1 MeV to several hundreds of MeV are accelerated in interplanetary space during large solar
perturbations called coronal mass ejections (Reames, 1999; Richardson and Cane, 2010). These sporadic events, also known





as solar proton events (SPEs), are associated with the presence of complex sunspots, and are therefore more frequent during solar maximum.

Auroral electrons originate from the Earth's magnetosphere, and are accelerated to energies of 1-30 keV during auroral substorms (Fang et al., 2008). Sudden enhancements of their flux occurs during geomagnetic active periods, which are more

frequent 1-2 years after peak of the 11-year solar cycle. Medium-energy electrons are accelerated to energies of a few hundred keV during geomagnetic storms in the terrestrial radiation belts (Horne et al., 2009). Precipitation of middle energy electrons can be triggered both by solar coronal mass ejections, and high-speed solar wind streams, leading to more frequent events near solar maximum and during the declining phase of the solar cycle. Particle precipitation, regardless of its origin, is thus modulated by solar activity, and varies with the solar cycle. However, these intermittent variations take place on different

timescales, and at different altitude regions. Their sources and variability have recently been reviewed by Mironova et al. (2015).

EPP affects the ionisation levels in the polar middle and upper atmosphere, leading to significant changes of the chemical composition. In particular, the production of odd nitrogen and odd hydrogen species causes changes in ozone abundances via catalytic cycles, potentially affecting temperature and winds (see, e.g., the review by Sinnhuber et al., 2012). Recent model

studies and the analysis of meteorological data have provided evidence for a dynamical coupling of this signal to the lower atmosphere, leading to particle-induced surface climate variations on a regional scale (e.g., Seppälä et al., 2009; Baumgaertner et al., 2011; Rozanov et al., 2012; Maliniemi et al., 2014).

The third, and most energetic components of EPP is represented by galactic cosmic rays (GCR), which mainly consist of protons with energies from hundreds of MeV to TeV. This continuous flux of particles is the main source of ionisation in the

troposphere and lower stratosphere. GCRs are deflected by the solar magnetic field, and hence their flux is anti-correlated with the solar cycle. Laboratory-based studies have confirmed the existence of ion-mediated aerosol formation and growth rates; however, the connection between GCR ionization and cloud production, and therefore convection, is still under debate. Meanwhile, the chemical impact via ozone-depleting catalytic cycles and subsequent dynamical forcing is rather well understood (Calisto et al., 2011; Rozanov et al., 2012).

The effect of various components of EPP on surface climate is an emerging research topic. However, the particle impact on regional climate may be comparable to that of the UV forcing (Seppälä and Clilverd, 2014). One of the major challenges here is to quantify the long-term climate impact of such local and mostly intermittent particle precipitations.

The uncertainties in the solar forcing itself are compounded by possible errors in the simulated climate response to this forcing in models (e.g., Stott et al., 2003; Scaife et al., 2013). Possible errors in climate model responses could be related

to biases in the representation of dynamical processes and dynamical variability, the inability of model radiation schemes to properly resolve SSI changes (Forster et al., 2011), or to the missing or inadequate representation of UV and particle-induced ozone signals (Hood et al., 2015). Any comparison of climate model simulations with observations could be affected by a combination of these possible sources of error. In addition, the comparison of models with observations is inhibited by the insufficient length of the observational records, and in some cases model simulations.





This paper will provide the first complete overview on solar forcing (radiative, particle and ozone forcing) recommendations for CMIP6 from preindustrial times to the future and provides in this respect an advance to earlier MIPs (CMIP5, CCMVal, CCMI) as it gives a complete and state-of-the-art overview on our current understanding of solar variability and provides the dataset in a user-friendly way.

Section 2 presents the historical to present-day solar forcing dataset with individual subsections on solar irradiance (Section 2.1) and particle forcing (Section 2.2). Section 3 provides a description of the future solar forcing recommendation, section 4 on the pre-industrial control forcing and section 5 finally a description of the solar induced ozone signal. A summary with respect to differences to the CMIP5 recommendation is given in section 6.

## 2   Historical (to Present) Forcing Data (1850-2014)

In this section we first describe the solar irradiance dataset (including the TSI, the F10.7 decimetric radio index, and the SSI, see Sec. 2.1), and subsequently address the energetic particle datasets (including solar protons, auroral electrons, medium-energy electrons, and and galactic cosmic rays, see Sec. 2.2).

### 2.1   Solar Irradiance (TSI, SSI, and F10.7)

This subsection starts with a description of the available TSI and SSI datasets from two different solar irradiance models
(NRLSSI and SATIRE), and one observational estimate (SOLID), before introducing the CMIP6 recommendation. Afterwards a recommendation on how to implement the solar irradiance forcing in CMIP6 models is provided. An evaluation of the comparison between different SSI forcing datasets with a focus on CMIP5 and CMIP6 solar irradiance recommendations in a line-by-line model and two state-of-the-art CCMs, i.e. CESM1(WACCM) and EMAC, is performed at the end to highlight the effects of solar irradiance variability on the atmosphere, and possible effects on atmospheric dynamics all the way to the ocean.

#### 2.1.1   Description of Solar Irradiance Datasets

**NRLTSI2 and NRLSSI2**

The Naval Research Laboratory (NRL) family of SSI models (Lean, 2000; Lean et al., 2011) is based on the premise that changes in solar irradiance from background quiet Sun conditions can be described by a balance between bright facular, and dark sunspot features on the solar disk. These two contributions are determined by linear regression between solar proxies, and
direct observations of TSI and SSI by satellite missions such as SORCE (Rottman, 2005). These models are thus empirical.

    Both the TSI and the SSI consist of a baseline solar contribution, with a wavelength-dependent contribution from bright faculae (i.e., the Mg II index, see below) and dark sunspots (i.e., sunspot area, see below). The time dependency in TSI and SSI thus emerges from the temporal variability in the solar proxies. SORCE measurements at solar minimum conditions are the basis for the adopted quiet Sun irradiance (Kopp and Lean, 2011) in NRLSSI2.





The recently updated version of the NRL models, named NRLTSI2 (for TSI only) and NRLSSI2, have been transitioned to the National Centers for Environmental Information (NCEI) as part of their Climate Data Record (CDR) Program (see http://www.ngdc.noaa.gov), and operational updates are provided on a near quarterly basis. Coddington et al. (2015) describe the model algorithm, the uncertainty estimation approach, and comparisons to observations in detail.

In NRLSSI2, a multiple linear regression approach of solar proxy inputs with observations of TSI from SORCE/TIM (Kopp et al., 2005), and observations of SSI from the SORCE/SOLSTICE (McClintock et al., 2005) and SORCE/SIM (Harder et al., 2005) instruments is used to determine the scaling coefficients that convert the proxy indices to irradiance variability. Because of concerns regarding the long-term stability of the SORCE SSI observations Lean and DeLand (2012) the wavelength-dependent scaling coefficients used in NRLSSI2 are derived from solar rotation time scales (i.e., the SSI observations and the

proxy indices are detrended with an 81-day running mean). A separate adjustment is made to extend the SSI variability to solar cycle time scales by a linear scaling that is constrained by the TSI variability. This adjustment is made in the separate facular and sunspot proxy records and the magnitude of the adjustment is smaller than the assumed uncertainty in the proxy indices themselves. In this approach, the integral of the SSI tracks the TSI; however, the relative facular and sunspot contributions at any given wavelength are not constrained to match their specific TSI contributions.

The NRLTSI2 and NRLSSI2 irradiances also include a speculated long-term facular contribution that produces a secular (i.e., underlying the solar activity cycle) net increase in irradiance from a small accumulation of total magnetic flux. This secular impact is specific to historical time scales (i.e., prior to 1950) and is consistent with simulations from a magnetic flux transport model (Wang et al., 2005).

**SATIRE-TS**

The SATIRE (Spectral And Total Irradiance REconstruction) family of semi-empirical models assumes that the changes in the solar spectral irradiance are driven by the evolution of the photospheric magnetic field (Fligge et al., 2000; Krivova et al., 2003, 2011). The model makes use of the calculated intensity spectra of the quiet Sun, faculae and sunspots generated from model solar atmospheres with a radiative transfer code (Unruh et al., 1999). SSI at a particular time is given the sum of these spectra, weighted by the fractional solar surface that is covered by faculae and sunspots, as apparent in solar observations.

The implementation of SATIRE employing solar images in visible light, and solar magnetograms (magnetic field intensity and polarity) is termed SATIRE-S (Wenzler et al., 2005; Ball et al., 2012; Yeo et al., 2014), and that based on the sunspot number is SATIRE-T (Krivova et al., 2010; Dasi-Espuig et al., 2014). Individual records are accessible at http://www2.mps. mpg.de/projects/sun-climate/. The dataset we consider here is a combination of SATIRE-S for observations between 1974 and 2014, where full-disc magnetograms are available, and a reconstruction based on SATIRE-T from 1850 to 1974, and from

2014 onwards. SATIRE-S has been demonstrated to be consistent with SSI measurements where the latter are reliable, see Yeo et al. (2015). On decadal to centennial time scales, SATIRE reproduces observations such as: the composite of the Lyman-$\alpha$ line at 121.5 nm (since 1947, Woods et al., 2000), the measured solar photospheric magnetic flux (since 1967), the empirically reconstructed solar open magnetic flux (since 1845, Lockwood et al., 2014), and the $^{44}$Ti activity in stony meteorites (Krivova et al., 2010; Vieira et al., 2011; Yeo et al., 2014).





Let us stress that SATIRE-TS and NRLSSI2 are internally consistent, in the sense that the integral of the modeled spectral irradiance equals the TSI. In NRLSSI2, this internal consistency also applies to the integral of the facular and sunspot contributions to SSI to their respective counterparts in TSI (see Coddington et al. (2015) for more details).

**Proxies Used**

Both NRLSSI2 and SATIRE-TS rely on the sunspot number when no other solar proxies are available. For the CMIP6 composite, we decided to rely on version 1.0 of the international sunspot number (from http://www.sidc.be/silso/versionarchive), even though a newer version 2.0 recently came out (Clette et al., 2014). Indeed, SSI models have not yet been thoroughly trained and tested with this new sunspot number. Recent results by Kopp et al. (2016) suggest that this revision has little impact after 1885, and leads to greater solar-cycle fluctuations prior.

In NRLSSI2 the proxy index for facular brightening is the composite Magnesium (Mg) II index from the University of Bremen. The Mg II index (Viereck et al., 2001) is a disk-integrated ratio of the core to the wings of the Mg II emission line at 280 nm, and used by many models as a UV proxy. The Mg II index is available from 1978 onwards; values prior to that are estimated from the sunspot number.

    In NRLSSI2, the proxy index for sunspot darkening is the sunspot area as recorded by ground-based observatories in white
light images since 1882 (Lean et al., 1998). Values prior to that are estimated from the sunspot number.

**SOLID Composite**

The task at hand – determine the most likely temporal variation in SSI – is challenged by the paucity of direct SSI observations, which, in addition, suffer from numerous instrumental artefacts. Recently, this task has been addressed by an international consortium to produce an observational SSI composite. The SOLID[1] SSI composite is the first of its kind in the sense that
it is based on a probabilistic approach. More specifically it is derived as the weighted mean of all available SSI observations in the satellite era. This approach leads to a new observational SSI composite that serves here as an independent data set to validate the SSI reconstruction models described above. However, the observations are restricted to the satellite era and the extension using proxies is limited back to 1950. Thus the SOLID composite cannot be employed for the CMIP6 solar forcing recommendation time period.

The making of such a composite involves several steps: first, the raw data are preprocessed (Schöll et al., 2016). Next, each individual time series is decomposed into different time scales (typically, from daily to annual). For each time scale, their uncertainty is estimated. All these records are then merged scale-wise, by computing their average, weighted by their scale-dependent uncertainties. Finally, the composite is obtained by adding up the average obtained for each time scale.

    Let us stress that the foremost aim has been to keep the observations, and ultimately the composite fully independent from
existing models. This means that no SSI models have been used to correct the observational data, which were taken at their face value, without any correction.

---

[1]FP7 SPACE Project *First European SOLar Irradiance Data Exploitation (SOLID)*; http://projects.pmodwrc.ch/solid/



One challenge of this - as with any statistical - approach is its dependence on the number of independent datasets. While for the past decades several missions were dedicated to measuring the UV band of the solar spectrum, the picture becomes more bleak when it comes to the visible part of the spectrum. In that band, only observations from SORCE/SIM are available. Clearly, this means that the SOLID composite relies entirely on the SORCE/SIM dataset in this spectral region. The controversial out-of-phase behavior of SORCE/SIM observations in that band (Harder et al., 2009) are likely to be an instrumental artefact (Lean and DeLand, 2012; Ermolli et al., 2013) but this has not yet been corrected in the SOLID composite.

### 2.1.2 CMIP6 Recommended Solar Irradiance Forcing

NRLSSI and SATIRE are not the only available models for reconstructing the SSI (Ermolli et al., 2013). However, they are the only ones that have been widely tested, and can easily cover the 1850–2300 time span for CMIP6 with one single and continuous record. The resulting homogeneity in time is a major asset of our reconstructions, and a necessary condition for obtaining a realistic solar forcing.

NRLSSI and SATIRE agree well on time scales of days to months, but exhibit more pronounced differences on longer time scales, which have fueled a debate (e.g. Yeo et al., 2015) that is unlikely to settle soon. In this context, and until additional information may help us better constrain long-term variability, the most reasonable approach consists in averaging both reconstructions, weighted by their uncertainty. However, since we are lacking uncertainties that can be meaningfully compared, our current recommendation is an arithmetic mean of the two model datasets: (i) the empirical model NRLTSI2 and NRLSSI2 (Coddington et al., 2015), and (ii) the semi-empirical model SATIRE-TS (Yeo et al., 2014). Clearly, this solution can be improved in the future as better reconstructions and observational composites (including SOLID) will become available.

For historical data (Jan 1, 1850 – Dec 31, 2014) both models rely, as described above, on one or several of: the international sunspot number V1.0, sunspot area distribution (after 1882), solar photospheric magnetic field (after 1974), and the Mg II index (after 1978). Since NRLSSI2 and NRLTSI2 have yearly averages only before 1882, we reconstructed sub-yearly variations by using an ARMAX (AutoRegressive Moving Average with eXogeneous input) model (Box et al., 2015) with the sunspot number as input.

The EUV band (10-121 nm) is required for CMIP6 but is absent from NRLSSI and SATIRE. We added it with spectral bins from 10.5-114.5 nm by using a nonlinear regression from the SSI in the 115.5-188.5 nm band, trained with TIMED/SEE data from 2002 to 2009. In some climate models the EUV flux is parameterised as a function of the F10.7 index, which is the daily radio flux at 10.7 cm from Penticton Observatory, adjusted to 1 AU, and measured daily since 1947 (Tapping, 2013). For practical purposes, we also provide this index. Values prior to 1947 are obtained by multi-linear regression to the first 20 principal components of the SSI and application of minor non-linear adjustments. Let us note that while the F10.7 index is a good proxy for EUV variability on daily to yearly time scales, this may not be true anymore on multi-decadal time scales.

The dataset, together with a technical description, and a routine for how to read and integrate the SSI data to the radiation bands used in climate models can be found at http://solarisheppa.geomar.de/cmip6. In addition, a recommendation on how to implement the SSI changes in the models is provided in the Appendix A. A detailed description of the CMIP6 solar irradiance forcing in TSI and SSI, and a comparison to the CMIP5 recommendation is presented in the following.



**Total Solar Irradiance (TSI)**

Figure 1 presents time series of the TSI from the CMIP5, CMIP6, and the original NRLTSI2 and SATIRE-TS datasets, along with one observed composite from PMOD (version 42.64.1508) (https://www.pmodwrc.ch/pmod.php?topic=tsi/composite/SolarConstant). We stress that all the data are taken at their face value, using their latest version, without any adjustments

or scaling, except for NRLTSI1, whose value we uniformly reduced by 5 W/m$^2$ to account for the new recommendation for average TSI (see below).

    All TSI records agree well on daily to yearly time scales, and in some cases (e.g. NRLTSI1 and NRLTSI2) they match as well on multi-decadal time scales. The major difference arises in the long-term behaviour of SATIRE-TS and NRLTSI2, which impacts the CMIP6 composite, and leads to a weaker trend as compared to the CMIP5 recommendation (which was based on

NRLTSI1 only). In both models, the historical reconstructions are sensitive to the assumptions made when constraining them to direct (satellite era) observations that suffer from large uncertainties. This mainly explains why these models differ before 1990 by an offset. There is no consensus yet as to which one better represents long-term solar variability, and this is what motivated us to average them for making the CMIP6 composite.

    More subtle differences between the different TSI datasets arise in the satellite era, especially with the unusually deep

solar minimum that occurred in 2008-2010: the NRLTSI2 model has a weak negative trend between successive solar minima, whereas the SATIRE-TS reconstruction exhibits a larger one. The resulting trend in the CMIP6 composite is comparable to the observational TSI composite from PMOD (grey area). Figure 1 does not show any model uncertainties, because these are either absent or difficult to compare. We do provide uncertainties, however, for the observational PMOD composite, based on an instrument-independent approach that is described in Dudok de Wit et al. (2016). Note that both models are mostly within

the $\pm 1\sigma$ confidence interval, which highlights how delicate it is to constrain them by observational data.

    After CMIP5, the recommended value of the average TSI during solar minimum was reduced from $1365.4 \pm 1.3$ W/m$^2$ to a lower value of $1360.8 \pm 0.5$ W/m$^2$ after reexamination by Kopp and Lean (2011), later confirmed independently by Schmutz et al. (2013). Based on this, the International Astronomical Union recently recommended $1361.0 \pm 0.5$ W/m$^2$ as the nominal value of the TSI, averaged over solar cycle 23 (which lasted from 1996 to 2008) (Mamajek et al., 2015). Our CMIP6 composite

complies with this recommendation.

    To summarise for the TSI: the CMIP6 and CMIP5 recommendations are comparable on decadal and sub-decadal time scales. They differ, however, by a weaker secular trend in CMIP6. Between 1980 and 1880, the difference TSI(CMIP6)-TSI(CMIP5) progressively increases from 0.1 to 0.4 W/m$^2$ (after correcting the aforementioned 5 W/m$^2$ offset in CMIP5). This results in a weaker change in solar forcing, which will be detailed in Sec. 2.1.3.

To estimate the impact of these different trends on the radiative forcing (RF), we have conducted a high spectral resolution calculation using a single profile with a line-by-line radiative transfer code (libradtran) described in more detail below. This indicated an instantaneous change in downward solar flux of -0.16 W/m$^2$ over the 1986-2009 period for the combined CMIP6 dataset. A crude estimate of the global mean forcing from this change is -0.04 W/m$^2$, which is relatively small in comparison to other forcings over this period.





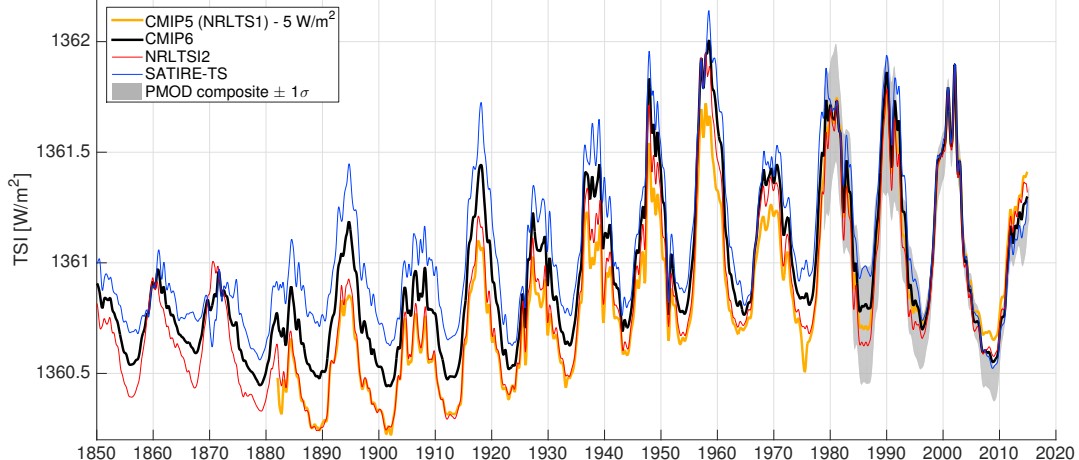

**Figure 1.** Comparison of several TSI reconstructions, showing 6-month running averages of: the NRLTSI1 record (reference for CMIP5), the CMIP6 composite, and the reconstructions from the NRLTSI2, and SATIRE-TS models. Also shown is the observational composite from PMOD (version 42.64.1508) with a $\pm 1\sigma$ confidence interval. A negative offset of -5 W/m$^2$ has been applied to the NRLTSI1 record to account for the change in average TSI that occurred between CMIP5 and CMIP6.

**Solar Spectral Irradiance (SSI)**

To investigate differences and similarities between the SSI datasets, following Ermolli et al. (2013), we concentrate on four specific wavelength ranges, 120–200 nm (UV1), 200–400 nm (UV2), 400–700 nm (VIS), and 700–1000 nm (NIR), with special emphasis on the CMIP5 (NRLSSI1) and the CMIP6 ((NRLSSI2+SATIRE-TS)/2) datasets. These ranges are relevant for climate studies, see for example Table 1 below. Figure 2 shows the SSI time series from 1880 through 2014. Note that we added vertical offsets by adjusting the mean values to facilitate their comparison, using the CMIP6 as a reference. We note that:

– The long-term increase from 1880 to 1980 is similar in all datasets (NRLSSI1 might appear to have larger long-term increase in the VIS but this is mostly caused by different solar cycle amplitudes). SATIRE-TS predicts a slightly larger increase in the UV2 (+0.03 W/m$^2$ with respect to NRLSSI2) from 1880 to 1930 compensated by a smaller increase in the VIS. NRLSSI2 has a larger but still small increase (+0.07 5W/m$^2$ with respect to SATIRE-TS) in the NIR from 1880 to present, which was also present in NRLSSI1 (CMIP5). The major difference come from SATIRE-TS after 1985 (see below).

– As already described for the TSI behaviour above (Fig. 1), SATIRE-TS predicts a significant downward trend of the baseline for the last three solar cycles, as can be seen by comparing the SSI at solar minima between cycles 21 & 22 (1985), 22 & 23 (1995), and 23 & 24 (2008). NRLSSI2 does not predict significant variations and therefore the



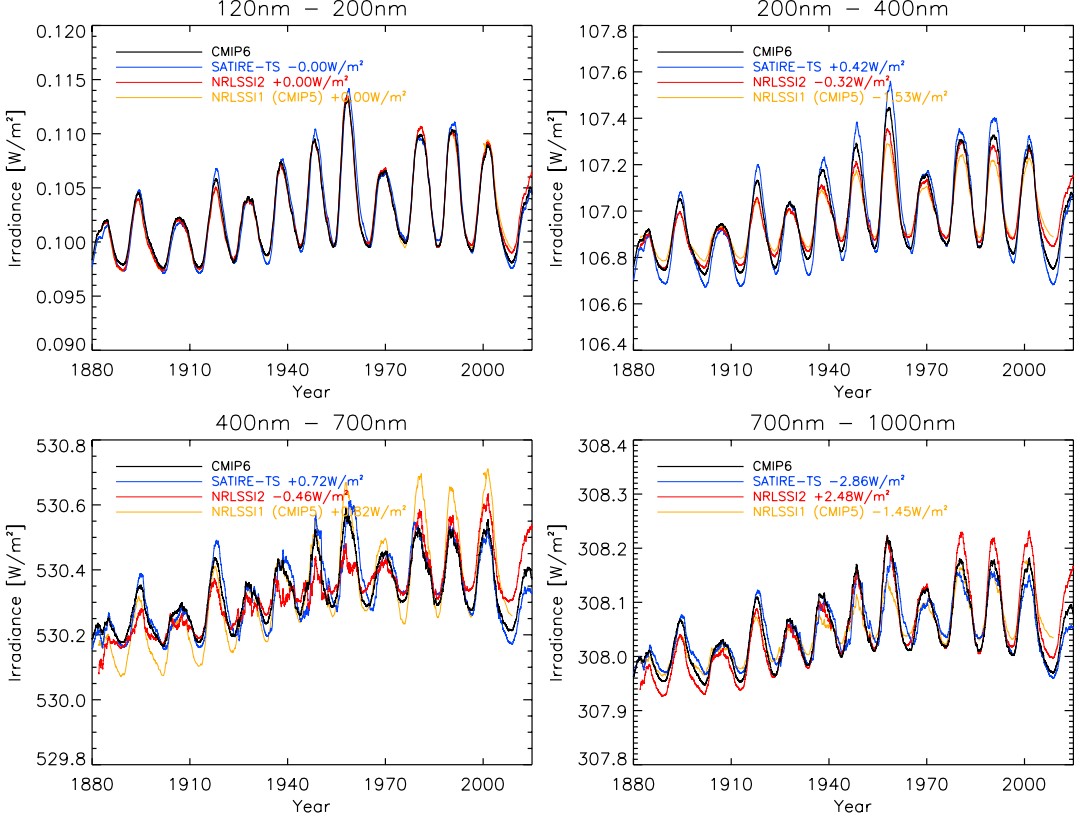

**Figure 2.** SSI time series from 1882 to 2014, integrated over following wavelength ranges: 120-200 nm (top left), 200-400 nm (top right), 400-700 nm (bottom left), and 700-1000 nm (bottom right). An offset, indicated in the legend, has been added to each time series, to ease visualisation. All time series are running averages over 2 years.

recommended CMIP6 time series has a slower downward trend than SATIRE-TS in the recent cycles. This trend was not apparent in the dataset recommended for CMIP5.

– The solar cycle variability in CMIP6 exceeds that of CMIP5, particularly in the UV2 and NIR ranges, while it is approximately the inverse in the VIS. The change in the NRLSSI model can be explained by the use of new, and higher-quality data from the SORCE mission on the rotational timescale in NRLSSI2, while NRLSSI1 was based on data from older satellite missions. In the UV2, SATIRE-TS predicts larger solar cycle amplitudes, which can be explained by a larger weight of the network at these wavelengths.

Figure 3 compares our CMIP6 dataset with the observational SOLID composite (see description above) and some direct SSI satellite observations. Generally speaking, the observations and observation-based composite agree very well with each other, and the CMIP6 dataset up to 200 nm. Larger cycle variations than in the CMIP6 SSI occur above about 200 nm in the observations. Such discrepancies are inherent to the observation of small variations over eleven years. In the VIS and NIR part



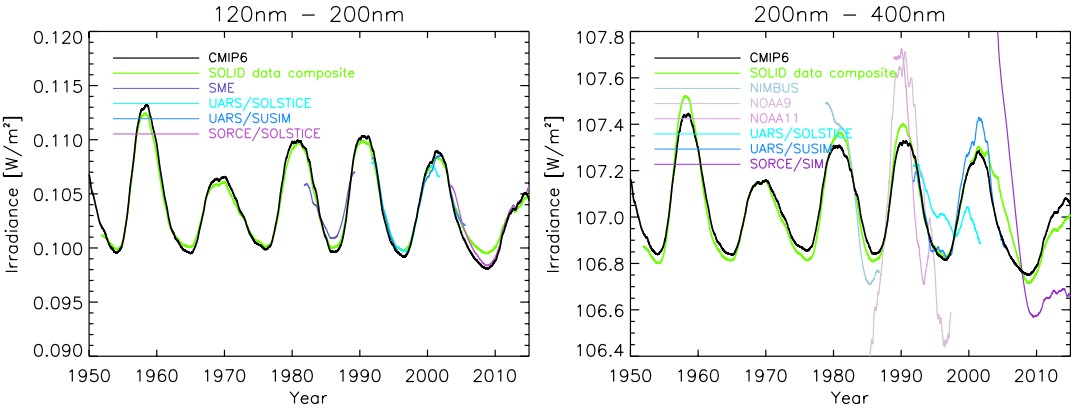

**Figure 3.** CMIP6 recommended SSI time series (black) from 1950 to 2015 together with the SOLID data composite (green) and relevant instrument observations for the following wavelength bins: 120nm–200nm (left), 200nm–400nm (right). The SOLID and instrument time series have been adjusted to match the average level of the CMIP6 time series. All time series are running averages over 2 years.

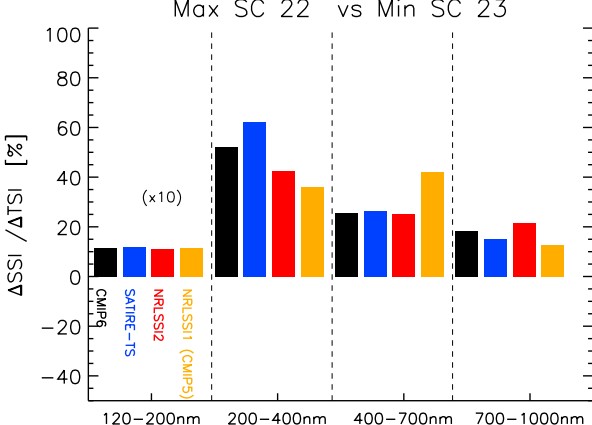

**Figure 4.** Contribution in percent of various wavelength ranges to the TSI variability between the maximum of cycle 22 and the minimum between cycles 22 and 23. Contributions between 120 nm and 200 nm have been multiplied by 10 for improved visibility. Maximum and minimum values have been taken on November 1989 and November 1994, respectively.

of the spectrum, the only available measurements are from the SORCE/SIM instrument, which has calibration issues (Lean and DeLand, 2012) and hence cannot be meaningfully compared to the modeled SSI datasets.

Figure 4 shows the contribution of the different wavelength ranges to TSI variations between solar maximum on November 1989 (solar cycle 22) and solar minimum on November 1994 (between cycles 22 and 23) for the different solar irradiance
5    models. A similar figure can be found in Ermolli et al. (2013) except that its dates belong to the next solar cycle, when SORCE/SIM was operating. Our dates coincide with the ones chosen in the CCM timeslice experiments, see Section 2.1.3.





SSI models agree very well for the 120-200 nm wavelength range. Discrepancies arise for wavelengths longward of 200 nm, as already discussed in Fig. 2. In the 200-400 nm range, the SATIRE-TS model shows the largest variability, followed by NRLSSI2, and NRLSSI1. This results in a CMIP6 variability that is larger than for CMIP5, 50% as compared to 35% (Fig. 4). In the VIS range this reverses, CMIP6 shows a smaller variability than CMIP5 (25% as compared to 40%). Remarkable is also

the very good agreement between NRLSSI2 and SATIRE-TS. In the NIR CMIP6 shows slightly larger variability than CMIP5 and remarkable here is the largest variability in NRLSSI2. The implications of these different spectral variabilities on the atmospheric heating and ozone chemistry and subsequent thermal and dynamical effects with respect to both, climatological differences between CMIP5 and CMIP6 as well as the solar cycle signals in CMIP5 and CMIP6 will be discussed in section 2.1.3.

### 2.1.3 Evaluation of SSI Datasets in Climate Models

Providing a first assessment of implications employing the SSI recommended for CMIP6 in comparison to CMIP5, we present results of two state-of-the-art chemistry climate models (CCM): – the Whole Atmosphere Community Climate Model (CESM1(WACCM); Marsh et al., 2013) – and the ECHAM/MESSy Atmospheric Chemistry model (EMAC; Jöckel et al., 2010, 2016). Additionally, we include results of single-profile radiative transfer calculations performed with the line-by-line

radiative transfer code *libradtran* (Mayer and Kylling, 2005). We use the latter to present estimates of direct SW radiative heating impacts neglecting the ozone chemistry feedback which is included in the CCM results.

**Chemistry-Climate Model Descriptions**

**WACCM:** The Whole Atmosphere Community Climate Model (version 4; Marsh et al., 2013) is an integrative part of the *Community Earth System Model* suite (version 1.0.6; Hurrell et al., 2013). CESM1(WACCM) is a "high-top" CCM covering

an altitude range from the surface to the lower thermosphere, i.e. up to $5 \times 10^{-6}$ hPa equivalent to approx. 140 km. It is an extension of the *Community Atmospheric Model* (CAM4; Neale et al., 2013) with all its physical parametrisations. For this study the model is integrated with a horizontal resolution of $1.9°$lat$\times 2.5°$lon and 66 levels in the vertical. CESM1(WACCM) contains a middle atmosphere chemistry module based on the Model for Ozone and Related Chemical Tracers (MOZART3; Kinnison et al., 2007). It contains all members of the $O_X$, $NO_X$, $HO_X$, $ClO_X$, and $BrO_X$ chemical groups as well as tropospheric

source species $N_2O$, $H_2O$, $CH_4$, CFCs and other halogen components (59 species and 217 gas-phase chemical reactions in total). Its photolysis scheme resolves 100 spectral bands in the UV and VIS range (121-750 nm) (see also Tab. 1). The SW radiation module is a combination of different parametrisations. Above approx. 70 km the spectral resolution is identical to the photolysis scheme (plus the parametrisation of Solomon and Qian, 2005, based on F10.7cm solar radio flux to account for EUV irradiances). Below approx. 60 km the SW radiation of CAM4 is retained, employing 19 spectral bands between 200

and 5,000 nm (Collins, 1998). For the transition zone (60-70 km) SW heating rates are calculated as weighted averages of the two approaches. Tab. 1 contains an overview of the SW radiation and photolysis schemes in comparison to EMAC, the second CCM utilized for this study. CESM1(WACCM) features relaxation of stratospheric equatorial winds to an observed or idealized Quasi-Biennial Oscillation (QBO; Matthes et al., 2010).



**EMAC:** The ECHAM/MESSy Atmospheric Chemistry (EMAC) model is a CCM that includes sub-models describing tropospheric and middle atmospheric processes and their interaction with oceans, land and human influences (Jöckel et al., 2010). It uses the second version of the Modular Earth Submodel System (MESSy2) to link multi-institutional computer codes. The core atmospheric model is the 5th generation European Centre Hamburg general circulation model (ECHAM5, Roeckner et al., 2006). For the present study we applied EMAC (ECHAM5 version 5.3.02, MESSy version 2.51, Jöckel et al., 2016) in the T42L47MA-resolution, i.e. with a spherical truncation of T42 (corresponding to a quadratic Gaussian grid of approx. 2.8 by 2.8 degrees in latitude and longitude) with 47 hybrid pressure levels up to 0.01 hPa (∼80 km). The applied model setup comprises, among others, the submodels: MECCA, JVAL, RAD/RAD-FUBRAD, and QBO. MECCA (Module Efficiently Calculating the Chemistry of the Atmosphere) (Sander et al., 2011a) provides the atmospheric chemistry model. JVAL (Sander et al., 2014) provides photolysis rate coefficients based on updated rate coefficients recommended by JPL (Sander et al., 2011b). RAD/RAD-FUBRAD (Dietmüller et al., 2016) provides the parameterisation of radiative transfer based on Fouquart and Bonnel (1980) and Roeckner et al. (2003) (RAD). For a better resolution of the UV-VIS spectral band RAD-FUBRAD is used for pressures lower than 70 hPa, increasing the spectral resolution in the UV-VIS from one band to 106 bands (Nissen et al., 2007; Kunze et al., 2014). Tab. 1 presents more details of the SW radiation and photolysis schemes in comparison to WACCM. The submodel QBO is used to relax the zonal wind near the equator towards the observed zonal wind in the lower stratosphere (Giorgetta and Bengtsson, 1999).

**CCM Experimental Design**

The CCM simulations with CESM1(WACCM) and EMAC are identically conducted in atmosphere-only timeslice configuration. This means, the external forcings such as the solar and the anthropogenic forcing are fixed for the whole simulation period, that is 45 model years plus spin-up (~5 years for EMAC, ~3 years for CESM1(WACCM)). Concentrations of greenhouse gases (GHGs) and ozone-depleting substances (ODS) are set to constant conditions representative for the year 2000. The lower-boundary forcing is specified by the mean annual cycle of SSTs and sea-ice of the decade 1995-2004 derived from the *HadISST1.1*-dataset (Rayner et al., 2003). All simulations are nudged towards an observed (EMAC) or idealized 28-months varying (CESM1(WACCM)) QBO. The only difference between the simulations is in the solar forcing. Four simulations for each of the following SSI datasets have been performed with EMAC and WACCM: CMIP6-SSI, its constituent datasets NRLSSI2 (Coddington et al., 2015) and SATIRE-TS (Krivova et al., 2010; Yeo et al., 2014), as well as NRLSSI1 (Lean, 2000). The latter was recommended as solar forcing for CMIP5 including a uniform scaling of the spectrum to match TSI measurements of the *Total Irradiance Monitor* (TIM) instrument. As one emphasis of this study is to highlight differences to the previous phase of CMIP we employed NRLSSI1 including this scaling and refer to it as NRLSSI1(CMIP5) in the following. Runs for each of the four dataset have been performed with both CCMs for a solar minimum timeslice and a solar maximum timeslice, respectively. For solar maximum timeslices, SSIs averaged over Nov. 1989 are used (maximum of solar cycle 22) while for the solar minimum timeslices averages over Nov. 1994 are chosen. The latter does not match the absolute minimum of solar cycle 21/22 (Jun. 1996). However, solar activity in Nov. 1994 was already close to the minimum. The differences

**Table 1.** Summary of spectral resolution of the SW radiation and photolysis schemes in EMAC and CESM1(WACCM). Boundaries of spectral intervals and further refinement in brackets when larger than one.

| Spectral region | Gases | CESM1(WACCM) | EMAC |
|---|---|---|---|
| SW radiation[*,+] | | | |
| Lyman-$\alpha$ | $O_2$ | | [121-122] |
| Schumann-Runge continuum | $O_2$ | | [125-175] (3) |
| Schumann-Runge bands | $O_2$ | | [175-205] |
| Herzberg cont./Hartley bands | $O_2$, $O_3$ | [200-245] | [206.5-243.5] (15) |
| Hartley bands | $O_3$ | [245-275] (2) | [243.5-277.5] (10) |
| Huggins bands | $O_3$ | [275-350] (4) | [277.5-362.5] (18) |
| UV-A/Chappuis bands | $O_3$ | [350-700] (2) | [362.5-690] (58) |
| Near Infrared/Infrared | $O_2$, $O_3$, $CO_2$, $H_2O$ | [700-5000] (10) | [690-4000] (3) |
| Photolysis | | | |
| Lyman-$\alpha$ | | [121-122] | [121-122] |
| Schumann-Runge continuum | | [122-178.6] (20) | |
| Schumann-Runge bands | | [178.6-200] (12) | [178.6-202] |
| Herzberg cont./Hartley bands | | [200-241] (15) | [202-241] |
| Hartley bands | | [241-291] (14) | [241-289.9] |
| Huggins bands | | [291-305.5] (4) | [289.9-305.5] |
| UV-B | | [305.5-314.5] (3) | [305.5-313.5] |
| UV-B/UV-A | | [314.5-337.5] (5) | [313.5-337.5] |
| UV-A/Chappuis bands | | [337.5-420] (17) | [337.5-422.5] |
| Chappuis bands | | [420-700] (9) | [422.5-682.5] |

[*]Note that given bands for CESM1(WACCM) apply below ∼65 km only. The resolution of the SW radiation code above ∼65 km corresponds to the resolution of the photolysis scheme. [+]Note that given bands from 121–690 nm for EMAC apply at pressures lower than 70 hPa only. At pressures larger than 70 hPa there is one band extending from 250–690 nm.

in solar activity between our solar minimum and solar maximum timeslices for the respective datasets are within a range of 0.988 W/m$^2$ for NRLSSI1(CMIP5) to 1.057 W/m$^2$ for NRLSSI2.

It should be noted that these experiments will illustrate only one part of solar influence on climate. Given the atmosphere-only set-up of the runs, oceanic absorption of (mainly visible) solar irradiance and subsequent heating and feedbacks to the atmosphere – the so-called bottom-up mechanism (see Gray et al., 2010, and references therein) – is not represented in our simulations. Therefore we focus only on stratospheric signals and "top-down" dynamically induced responses in the troposphere. A second constraint of this study's experimental set-up is the choice of one solar cycle. Solar activity and hence spectral ir-



radiance vary between different solar cycles. However, these differences are small compared to the total 11 year solar cycle amplitude and will probably not affect the main results of this study. It should also be noted that the timeslice simulations were designed as a sensitivity study to test the impact of the different solar input datasets. They do not represent the full feedbacks of transient CMIP6 simulations.

**5**   **Radiative Transfer Model libradtran**

Radiatiative transfer calculations were performed with the high resolution model libradtran (Mayer and Kylling, 2005), which is a library of radiative transfer equation solvers widely used for UV and heating rate calculations (www.libradtran.org). Libradtran was configured with the pseudo-spherical approximation of the DISORT solver, which accounts for the sphericity of the atmosphere, running in a six-streams mode. Calculations pertain to a cloud- and aerosol-free tropical atmosphere (0.56°N),

the surface reflectivity is set to a constant value of 0.1 and effects of Rayleigh scattering are enabled. The atmosphere is portioned into 80 layers extending from the surface to 80km. The model output is annual averages of spectral heating rates from 120 nm to 700 nm in 1 nm spectral resolution, calculated according to the recommendations for the Radiation Intercomparison of the Chemistry-Climate Model Validation Activity (CCMVal) (Forster et al., 2011). As for the CCM simulations described above, calculations of the heating rates were performed for CMIP6-SSI, SATIRE-TS, NRLSSI2 and NRLSSI1(CMIP5). The

same climatological ozone profile is specified for both solar maximum and minimum conditions in order to assess the direct effects in atmospheric heating by SSI variations only. As such, the line-by-line calculations do not take into account the positive ozone feedback with the solar cycle and SW heating rate changes are expected to be weaker compared to the signatures in the two CCM simulations.

**Methods**

The analyses presented in the following consist of differences between climatologies derived from the various simulations. Given the timeslice configuration of the CCM runs with all external forcings equal except for the SSI dataset, we assume that statistically significant differences of two climatologies are the result of the differing solar irradiance forcings. Confidence intervals (95%) as presented in Figs. 5 and 7 as well as statistical significances (p<5%) as marked in Figs. 9 and 10 are based on 1000-fold bootstrapping. Confidence intervals in Figs. 5 and 7 are only given for the CCM-results related to CMIP6 SSI.

**Climatological Differences to CMIP5**

Although all solar irradiance reconstructions subject to this analysis agree fairly well in TSI (see Fig. 1), they disagree significantly with respect to the spectral distribution of energy input, i.e. the shape of the solar spectrum. This is obvious from the offsets noted in Fig. 2 for the different spectral regions above 200 nm. Hence, we focus first on the climatological differences between the solar forcing in CMIP5 and CMIP6. We therefore compare the minimum timeslice simulations from the two

CCMs and libradtran in Fig. 5 with respect to the climatological annual mean SW heating rates, as well as the temperatures, and ozone concentrations between the two CCMs resulting from CMIP6-SSI, NRLSSI2, and SATIRE-TS, respectively, as dif-





ferences to equivalent simulations forced by NRLSSI1(CMIP5). The profiles represent the tropical (averaged over 25°S-25°N) stratosphere and mesosphere (100–0.01 hPa) for annual mean conditions for the CCMs and libradtran.

Employing CMIP6-SSI results into significantly decreased radiative heating of large parts of the mesosphere and stratosphere (above 10 hPa) compared to NRLSSI1(CMIP5). Whereas the largest differences can be found at the stratopause with approx.

-0.35 K/day according to both CCMs, and even more -0.42 K/day for libradtran (without any ozone chemistry feedback), libradtran and EMAC yield slightly increased SW heating rates below ~7 hPa and 10 hPa, respectively. This weaker SW heating in the new CMIP6 SSI datset in the upper stratosphere and the stronger heating in the lower stratosphere are confirmed by the wavelength dependent percentage changes between the CMIP6 and CMIP5 SSI datasets with respect to the radiation and photolysis schemes (Fig.6). Regardless of the number of bands in the radiation code, both models show a smaller percentage

difference of -5% below about 300nm and weaker or negligible differences above 300nm (Fig.6).

Significant differences in radiative heating throughout the stratosphere related to the three state-of-the art SSI reconstructions are produced only with radiation codes of high spectral resolution such as in libradtran or – to a lesser degree – in EMAC (for the middle to lower stratosphere). Comparisons between CMIP6-SSI and its constituents NRLSSI2 and SATIRE-TS in WACCM and EMAC lead to the conclusion that the choice of the CCM and its specific radiation and photolysis scheme is more important

than the choice of the SSI dataset with respect to SW heating rates. In addition the ozone chemistry damps the SW heating response in the CCMs as compared to libradtran which misses the ozone feedback. Less SW radiation below 300nm reduces ozone production (note also the reduced photolysis rates around 240nm in Fig.6) and hence less ozone is available to absorb SW radiation and results in a relative cooling of the upper stratosphere.

Corresponding to the SW heating rate differences, large parts of the stratosphere and mesosphere are significantly cooler (up

to -1.6K at the stratopause) in simulations using CMIP6-SSI compared to NRLSSI1(CMIP5) irradiances. Note that libradtran results are shown for the SW heating rate differences only, as temperature and ozone profiles are prescribed for the radiative transfer calculations. No significant differences in temperature are found when employing NRLSSI2 or SATIRE-TS instead of CMIP6-SSI in CESM1(WACCM) which has a coarser spectral resolution in the SW heating parameterization than EMAC (Tab.1 and Fig.6). EMAC instead simulates significantly lower (higher) temperatures in the stratosphere when using NRLSSI2

(SATIRE-TS) than CMIP6-SSI forcing and in general a warmer stratosphere (and cooler stratopause and mesosphere) than CESM1(WACCM).

The impact of CMIP6-SSI as compared to NRLSSI1(CMIP5) irradiance changes on ozone are more complicated. In the middle tropical stratosphere, ozone concentrations are significantly lower (peaking at ~7 hPa with approx. -3.2%). In contrast, ozone concentrations around the stratopause are significantly higher for CMIP6-SSI (+0.8% and +1.6% according

to EMAC and CESM1(WACCM), respectively) than under NRLSSI1(CMIP5) irradiances. Despite the considerable differences in spectral resolution of the photolysis schemes (Tab.1 and Fig.6)), for larger parts of the stratosphere below about 3hPa, CESM1(WACCM) and EMAC agree fairly well. For both models the SATIRE-TS irradiances show larger signals than NRLSSI2 irradiances with the signal for CMIP6 in between. The ozone signals start to differ at and above the stratopause, probably due to the more detailed photolysis code and the higher model top in CESM1(WACCM) as compared to EMAC. The





ozone signal is much more uncertain with respect to the different SSI forcings than the SW heating rate and the temperature signals.

In summary, the CMIP6-SSI irradiances lead to lower SW heating rates, lower temperatures as well as smaller ozone signals in the lower stratosphere and larger ozone signals in the upper stratosphere and lower mesosphere than the CMIP5-SSI irradiances. Differences between the three tested SSI datasets occur in the SW heating rates only with a very high spectral resolution of the radiation code (libradtran, EMAC) and more prominent for ozone in a similar way for both CCMs, i.e. stronger effects occur for SATIRE-TS than NRLSSI2. These direct radiative effects in the tropical stratosphere lead to a weakening of the meridional temperature gradient and hence to a statistically significant weakening of the stratospheric polar night jet in early winter (not shown).

**Impacts of Solar Cycle Variability**

The second question tackled by this evaluation is the atmospheric impact of the 11-year solar cycle using different SSI irradiance reconstructions. A special focus lies on the comparison of the new CMIP6 dataset with its predecessor NRLSSI1(CMIP5). Fig. 7 provides annual mean tropical (25°S-25°N) profiles analogue to Fig. 5 but now illustrating differences between perpetual solar maximum and perpetual solar minimum conditions according to simulations forced by the various SSI-datasets.

All models and SSI-forcings produce the well-known solar cycle impact of enhanced SW heating at solar maximum throughout the upper stratosphere and mesosphere. Differences to solar minimum forcing peak at the stratopause with approx. +0.19 to +0.23 K/day. Only the libradtran-calculations – that do not include any ozone-feedback – yield considerably weaker responses. According to libradtran and CESM1(WACCM), CMIP6-SSI produces slightly higher SW heating rates than NRLSSI1(CMIP5). However, for EMAC this is not the case. For both CCMs and libradtran, the usage of SATIRE-TS leads to strongest solar cycle induced SW heating rate signals, while NRLSSI2 is associated with the weakest response (though not significantly different from NRLSSI1(CMIP5) for EMAC and libradtran).

Temperatures in the tropical stratosphere and mesosphere are generally higher during solar maximum than during phases of low solar activity. A local maximum of temperature differences is found at the stratopause with positive differences of 0.8-1.0 K compared to solar minimum. According to both CCMs, CMIP6-SSI forcing yields slightly higher temperatures (up to +0.2 K in the mesosphere in CESM1(WACCM)) for the stratopause region and the (lower) mesosphere than NRLSSI1(CMIP5). However, most of these differences are statistically not significant. Comparing CMIP6-SSI-forced results with its components NRLSSI2 and SATIRE-TS yields heterogeneous results. According to EMAC, NRLSSI2 leads to a slightly weaker solar cycle response throughout the stratosphere, while the mesospheric response is stronger than SATIRE-TS and CMIP6-SSI. CESM1(WACCM)-results show that the stratospheric (up to approx. 2 hPa) solar cycle response to CMIP6-SSI-forcing in temperature is slightly weaker than in both, NRLSSI2- and SATIRE-TS-driven simulations. As opposed to that, simulations forced by SATIRE-TS and CMIP6-SSI yield very similar warming signals in the mesosphere while NRLSSI2 produces a (significantly) weaker response in the mesosphere.

The solar cycle signal in ozone is very consistent for most parts of the stratosphere and mesosphere with respect to the SSI datasets. More important for the solar ozone signals seems to be the choice of the CCM (with its specific photolysis scheme, see



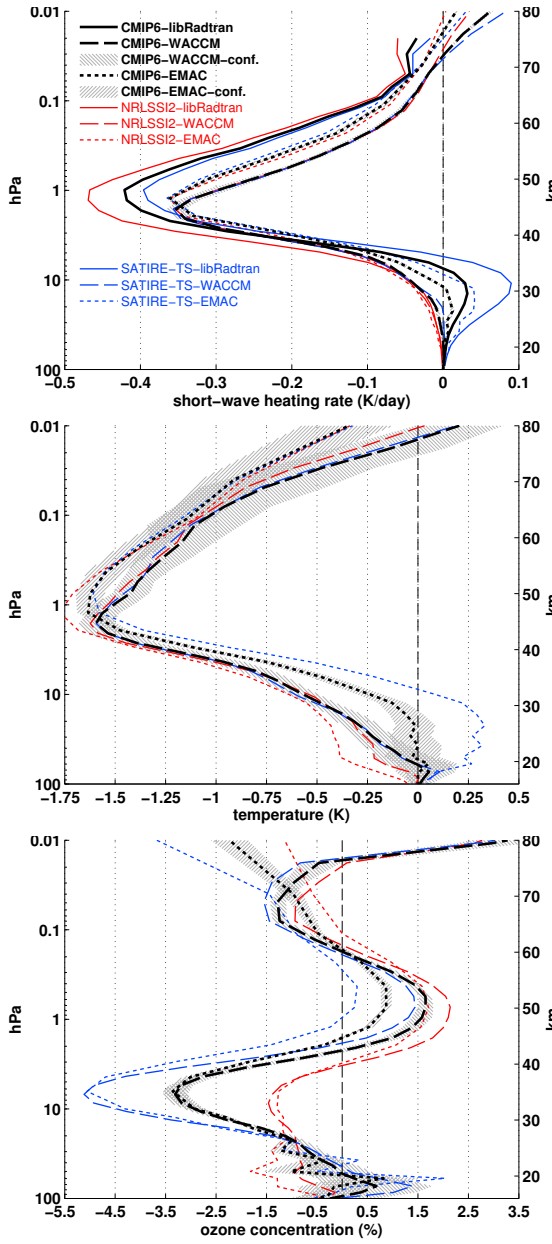

**Figure 5.** Impact of solar forcing according to CMIP6 (black) as well as constituent NRLSSI2 (red) and SATIRE-TS (blue) datasets on climatological (annual mean) profiles of SW heating rates (top), temperature (center), and ozone concentrations (bottom) averaged over the tropics (25°S–25°N) when compared to NRLSSI1(CMIP5) solar forcing; derived from simulations with CESM1(WACCM) (long-dashed), EMAC (short-dashed), and libradtran radiative transfer calculations (solid, only top panel) only shown for SW heating rates; 95% confidence intervals for CMIP6 simulations (hatched) estimated by bootstrapping.





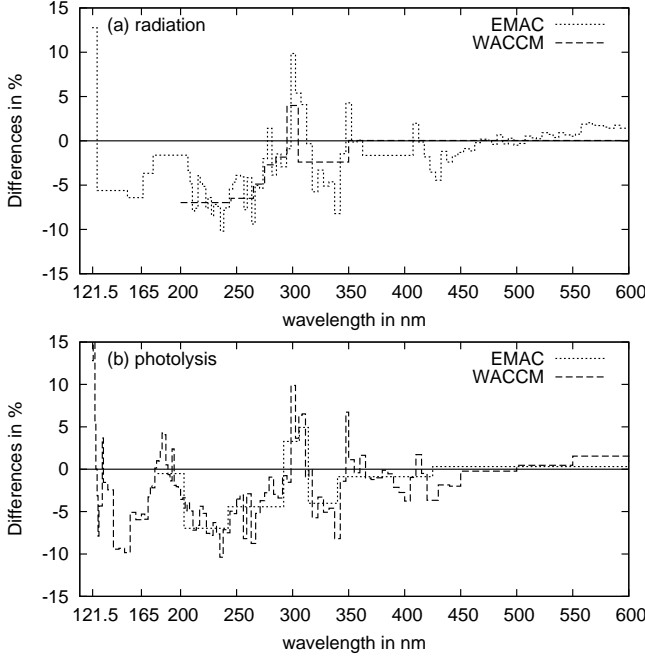

**Figure 6.** CMIP6 SSI differences in % for perpetual solar minimum conditions compared to CMIP5(NRLSSI1). Top: in the spectral resolution of the radiation schemes; bottom: in the spectral resolution of the photolysis schemes of EMAC (short-dashed) and CESM1(WACCM) (long-dashed).

also Fig. 8), especially for the lower stratosphere (10 hPa and below). All analyzed combinations of CCMs and forcing datasets agree very well on the (relative) peak of the ozone response (+2.3-2.5%) to the solar cycle at 3-5 hPa. In the lower mesosphere (0.2-1 hPa) CMIP6-SSI (and SATIRE-TS) lead to a significantly weaker solar cycle ozone response (+0.3-0.5% at 0.5 hPa) than NRLSSI1(CMIP5) (and NRLSSI2; +0.6-0.8% at 0.5 hPa). For the lower stratosphere (below 7 hPa) both CCMs agree that

SATIRE-TS leads to strongest solar cycle ozone signals, though still within the uncertainty associated with CMIP6-SSI-forced simulations. The comparison between CMIP6-SSI and NRLSSI1(CMIP5) yields no unequivocal result: CESM1(WACCM) exhibits a secondary maximum ozone response at approx. 70 hPa that is weaker with CMIP6-SSI than with NRLSSI1(CMIP5) while the opposite is seen in EMAC. Given the large uncertainty in the lower stratospheric solar ozone signal, we can only conclude that the signal is positive.

In summary, the CMIP6-SSI irradiance forcing leads to slightly enhanced solar cycle signals in SW heating rates, temperatures as well as ozone than the CMIP5-SSI irradiance forcing. In general, differences between the different SSI datasets are statistically not significant. Note that statistically significant irradiance differences between CMIP5 and CMIP6-SSI irradiances are particularly observed between 300 and 350nm, a wavelength region important for ozone destruction (below 320nm) consistently in both CCMs (Fig. 8). The direct radiative effects in the tropical stratosphere from the CMIP6-SSI dataset, i.e. en-

hanced solar cycle signals in SW heating rates, temperatures, and ozone in the tropical upper stratosphere lead to the expected





strengthening of the meridional temperature gradient and hence to a statistically significant stronger stratospheric polar night jet which propagates poleward and downward during winter from December through January (Fig. 9) and significantly affects the troposphere with a positive AO-like signal developing in late winter, i.e. January and February (Fig. 10). This signal is very similar and statistically significant for both CCMs, therefore the ensemble mean of both models is shown. Besides the radiative

impact of the solar cycle, also energetic particles have an impact on the atmosphere and will be discussed in the following.

## 2.2  Particle Forcing

Precipitating energetic particles ionize the neutral atmosphere leading to the formation of $NO_x$ ([N] + [NO] + [NO$_2$]) and $HO_x$ ([H] + [OH] + [HO$_2$]) (Porter et al., 1976; Rusch et al., 1981; Solomon et al., 1981) as well as of some more minor species (Verronen et al., 2008; Funke et al., 2008; Winkler et al., 2009; Verronen et al., 2011; Funke et al., 2011) due to both dissociation

and ionization of the most abundant species, as well as due to complex ion chemistry reaction chains. The formation of $NO_x$ and $HO_x$ radicals leads to catalytic ozone loss that further triggers changes of the thermal and dynamical structure of the middle atmosphere. Energetic Particle Precipitation (EPP) introduces thus chemical changes to the middle atmospheric composition and can therefore only be considered explicitly in climate simulations that employ interactive chemistry. In the following we provide recommendations for the consideration of EPP effects in CCMs separately for auroral and radiation belt

electrons (Sec. 2.2.1), for solar protons (Sec. 2.2.2), and for galactic cosmic rays (Sec. 2.2.3). In most cases, particle forcing can be expressed in terms of ion pair production rates. Recommendations for their implementation into chemistry schemes are provided in Sec. 2.2.4.

### 2.2.1  Geomagnetic Forcing (Auroral and Radiation Belt Electrons)

Energetic particles are trapped in the space around the Earth dominated by the geomagnetic field (known as the magnetosphere).

The loss of electrons into the atmosphere is termed "electron precipitation". Due to the Earth's magnetic field configuration electron precipitation occurs mainly in the polar auroral and sub-auroral regions, i.e., at geomagnetic latitudes that typically higher than 50 °. Enhanced loss fluxes are associated with geomagnetic storms, which can occur randomly, and also with periodicity's ranging from the ∼27 day solar rotation to the 11-year solar cycle, and even to multi-decadal timescales. The altitudes at which precipitating electrons deposit their momentum are dependent on their energy spectrum, with lower energy

particles impacting the atmosphere at higher altitudes than those with higher energies (e.g. Turunen et al., 2009). Auroral electrons, originating principally from the plasma sheet, have energies < 10 keV and affect the lower thermosphere (95–120 km). Processes that occur in the outer radiation belt typically generate mid-energy electron (MEE) precipitation within the energy range ∼10 keV to several MeV, affecting the atmosphere at altitudes of ∼50–100 km (Codrescu et al., 1997).

Odd nitrogen, produced by precipitating electrons, is long-lived during polar winter and can then be transported down from

its source region into the stratosphere, to altitudes well below 30 km. This has been postulated already by Solomon et al. (1982) and observed many times (Callis et al., 1996; Randall et al., 1998; Siskind, 2000; Funke et al., 2005; Randall et al., 2007). This so-called EPP "indirect effect" contributes significant amounts of $NO_y$ to the polar middle atmosphere during every winter in both hemispheres, however, with varying magnitude ranging from a few percent up to 40% (Randall et al., 2009; Funke et al.,





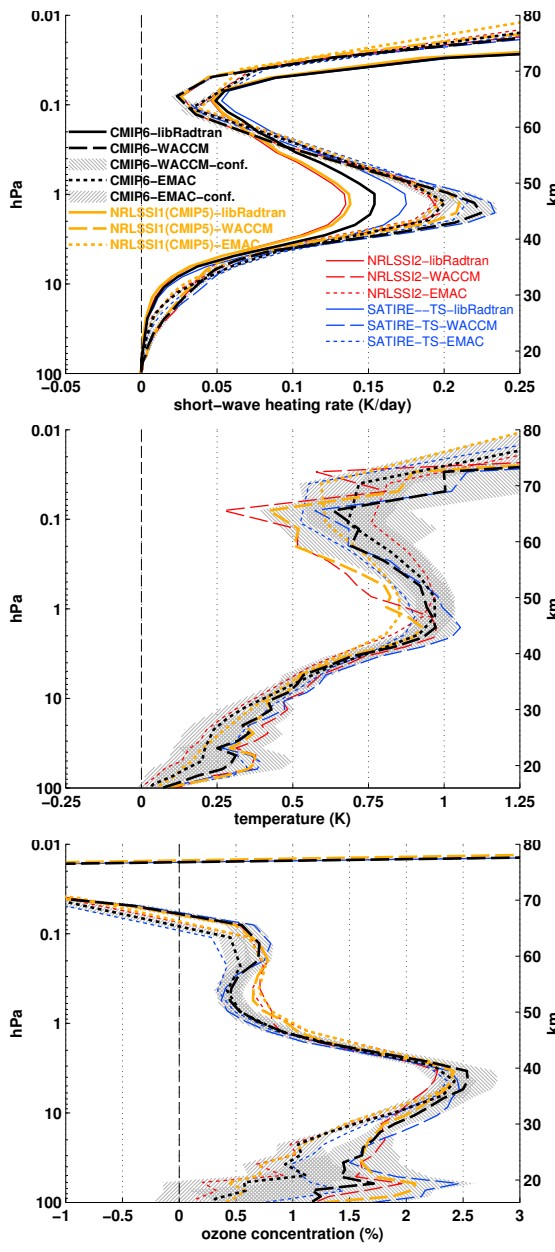

**Figure 7.** Impact of the 11-year solar cycle (differences between perpetual solar maximum and solar minimum experiments) on climatological (annual mean) profiles of shortwave heating rates (top), temperature (center), and ozone concentrations (bottom) averaged over the tropics (25°S–25°N) according to CMIP6 (black) and CMIP5 (yellow) solar forcing as well as NRLSSI2 (red) and SATIRE (blue) derived from simulations with CESM1(WACCM) (long-dashed), EMAC (short-dashed), and libradtran radiative transfer calculations (solid; only in the top panel); 95% confidence intervals for CMIP6 simulations (hatched) estimated by bootstrapping.





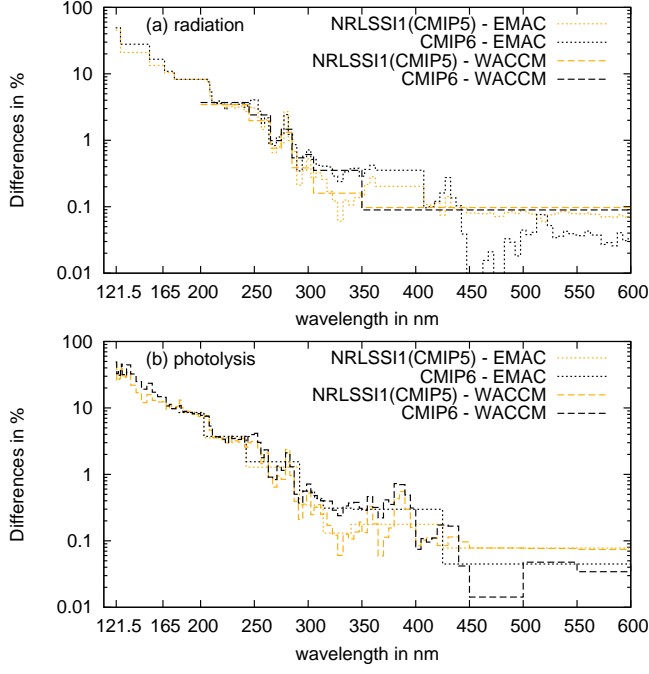

**Figure 8.** SSI differences in % for the solar amplitude between perpetual solar maximum and perpetual solar minimum conditions. Top: in the spectral resolution of the radiation schemes; bottom: in the spectral resolution of the photolysis schemes of EMAC (short-dashed) and CESM1(WACCM) (long-dashed).

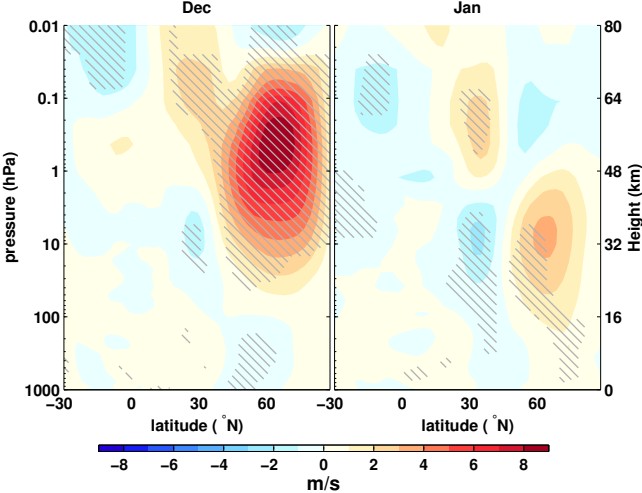

**Figure 9.** Zonal mean zonal wind response to the 11-year solar cycle according to CMIP6-SSI in December and January as "ensemble mean" of CESM1(WACCM) and EMAC simulations; hatched areas denote statistical significances (p<5%) of shown differences.



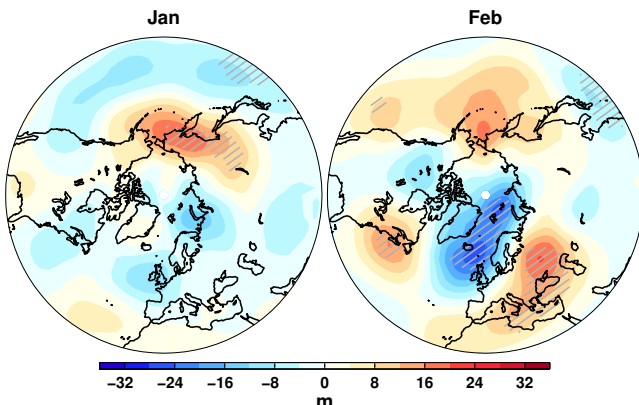

**Figure 10.** 500 hPa geopotential height response to the 11-year solar cycle according to CMIP6-SSI in January and February as "ensemble mean" of CESM1(WACCM) and EMAC simulations; hatched areas denote statistical significances (p<5%) of shown differences.

2014a). Its consideration in climate models with their upper lid in the mesosphere, thus not covering the entire EPP source region, requires the implementation of an upper boundary condition (UBC) that accounts for the transport of $NO_x$ into the model domain, as discussed below.

Stratospheric ozone loss due to electron-induced $NO_x$ production in the upper mesosphere /lower thermosphere and subse-
quent downward transport has been postulated by model experiments many times (Solomon et al., 1982; Schmidt et al., 2006; Marsh et al., 2007; Baumgaertner et al., 2009; Reddmann et al., 2010; Semeniuk et al., 2011; Rozanov et al., 2012). However, observational evidence for EPP-induced variations of stratospheric ozone linked to geomagnetic activity, characterised by a negative anomaly moving down with time during polar winter, have been given only very recently (Fytterer et al., 2015a; Damiani et al., 2016).

In addition, mesospheric ozone effects have been observed (Andersson et al., 2014a; Fytterer et al., 2015b) which are caused by $HO_x$ increases during MEE precipitation (Verronen et al., 2011). Although the $HO_x$-driven response is short-lived, the frequency of MEE events is large enough to cause solar cycle variability in ozone (Andersson et al., 2014a). $HO_x$ response is seen at magnetic latitudes connected to the outer radiation belts, with e.g. the yearly amount of $HO_x$ varying with the observed magnitude of precipitation (Andersson et al., 2014b). The consideration of the effects of MEE on atmospheric species other
than $NO_x$, $HO_x$, and ozone have not been investigated in detail to date, but they can be expected to be qualitatively similar to those caused by solar proton events (SPEs) (Verronen and Lehmann, 2013).

The impact of magnetospheric particles on the atmosphere is strongly linked to the strength of geomagnetic activity; this has been shown both for the direct production of NO in the thermosphere (Marsh et al., 2004; Hendrickx et al., 2015) and mesosphere (Sinnhuber et al., 2016), for mesospheric OH production Fytterer et al. (2015b), and for the EPP indirect effect
(Sinnhuber et al., 2011; Funke et al., 2014a). Geomagnetic activity can be constrained over centennial time scales by means of proxy data provided by geomagnetic indices. Since our forcing dataset for magnetospheric particle precipitation relies on these indices, their reconstruction and homogenisation is discussed first.



### Reconstruction of Geomagnetic Indices

Geomagnetic indices provide a measure of the level of geomagnetic activity resulting from the response of the magnetosphere-ionosphere system to variability in the solar and near-Earth solar wind forcings. Many geomagnetic indices have been constructed and different indices are sensitive to different aspects of magnetospheric and ionospheric dynamics (Mayaud, 1980).

The Kp and Ap geomagnetic indices (Bartels, 1949) are directly related by a quasi-logarithmic conversion; they are proxies for the global level of geomagnetic activity, and are used as inputs to parameterisations of magnetospheric particle precipitation. For the historical solar forcing data, daily values of the Kp and Ap indices from 1850 to 2014 are required. However, these indices, as provided by the International Service of Geomagnetic Indices (http://isgi.unistra.fr/), have only been produced from 1932 onwards. It is not possible to directly and consistently extend the Kp and Ap indices prior to 1932, as they use data from 13 geomagnetic observatories around the globe, and these data are unavailable further back in time. So, before 1932 the Kp and Ap indices must be estimated from other geomagnetic indices. The aa index (Mayaud, 1972) is the most appropriate choice, as it was constructed to be as similar as possible to the Ap index on annual timescales (Lockwood et al., 2013). However, the original aa-index only extends back to 1868 (also available from http://isgi.unistra.fr/), and so an extension (Nevanlinna, 2004) to the aa-index is also employed, extending it back to 1844 by use of the Ak indices from the Helsinki geomagnetic observatory, spanning 1844–1912. In addition, we implement a correction to the aa-index to account for a change in the derivation of the index in 1957, see (Lockwood et al., 2014). On larger than annual timescales, the response of the aa and Ap indices is similar, and the indices are positively linearly correlated. However, on daily timescales the relationship between aa and Ap is not linear, and also displays a regular annual variation. Therefore, to estimate the daily Ap indices during the period 1868-1931, we used piecewise polynomial fits between the daily Ap and aa values for the period 1932 — present, for each calendar month. These fits were then extrapolated to estimate the Ap values between 1868 and 1931 from the aa values. This process was repeated to estimate the relationship between the Ak indices provided by Nevanlinna (2004), and the Ap values estimated from the aa index. The piecewise polynomial fits for each calendar month were calculated using the overlap period between the Ak and estimated Ap records, 1868–1912. These were then extrapolated to estimate Ap in the period 1850–1867. Figure 11 shows the time series of the reconstructed Ap index and the aa and Ak indices used for extension back to 1850.

The daily Kp index for the period 1868–1931 was estimated by using the monthly piecewise polynomial aa-Ap fits to estimate the 3-hourly ap-index values from the aa index values. These 3-hour ap values were then converted to the corresponding Kp indices, from which the daily mean was calculated. Since only daily Ak data is available, such an approach is not possible for the period 1850–1867, and so here the daily estimates of Ap, derived from Ak, are directly converted into daily Kp. The quasi-logarithmic nature of the conversion between the hourly Kp and ap indices, means that calculating daily values of Kp in this manner results in lower values than the standard method of averaging the eight 3-hourly values in a day, resulting in a slight bias in the Kp estimates. A statistical correction for this bias was employed by estimating the bias using the difference between the aa-derived Kp and the Ak-derived Kp for the period 1868–1912. The estimated bias was then subtracted from the Ak-derived Kp estimates for the period 1850–1867.





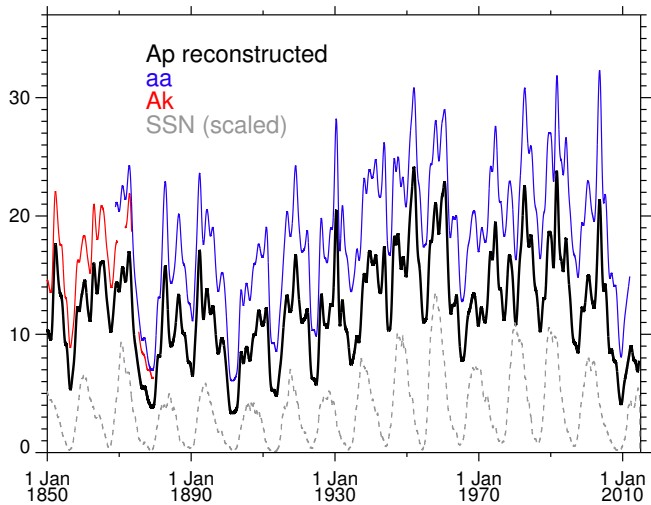

**Figure 11.** Time series of the reconstructed Ap index (black), together with the aa (blue) and Ak (red) indices used for its reconstruction, with comparison to the sunspot number variability (SSN scaled by a factor of 0.67, grey dashed). All the data have been smoothed with a 365-day running mean. Note that the reconstructed Ap includes the original Ap data from the International Service of Geomagnetic Indices since 1932.

**Auroral Electrons**

Lower thermospheric nitric oxide production by auroral electron precipitation can only be considered explicitly in CCMs extending up to 120 km or higher. There were only a few Earth system models of this characteristic in CMIP5 and it is expected that the number of such models will not increase significantly within CMIP6. Most of the models falling into this category use parameterizations for the calculation of auroral ionisation rates or NO productions in the polar cusp and polar cap (Schmidt et al., 2006; Marsh et al., 2007). Those parameterisations are typically driven by geomagnetic indices and we hence recommend the use of the extended Ap or Kp time series described above.

Figure 12 demonstrates the improvement in 2004–2009 wintertime polar $NO_y$ modelling when production due to electron precipitation is included. The simulations are from the SD-WACCM model version 4 (Marsh et al., 2013) nudged to the NASA Global Modeling and Assimilation Office Modern-Era Retrospective Analysis for Research and Applications (MERRA) (Rienecker et al., 2011) dynamics, and they are compared to observations from the ACE-FTS instrument (Jones et al., 2011). The auroral electron contribution was calculated with a Kp-based parameterisation and was further controlled through eddy diffusion affecting the $NO_x$ descent from lower thermosphere. MEE ionisation and $NO_x$ production was calculated using electron flux observations from the NOAA SEM-2 medium energy proton and electron detector (MEPED) instrument onboard the POES spacecraft (Evans and Greer, 2000), using methods described in more detail in the following MEE section. Enhancing the transport of auroral $NO_x$ from the lower thermosphere and including the mesospheric $NO_x$ production by MEE clearly improves the wintertime $NO_y$ near the stratopause. Around 0.1 hPa, modelled $NO_y$ increases by 100% in both hemispheres, which leads to better agreement with ACE-FTS. Both auroral electrons and MEE have a clear impact, although the auroral

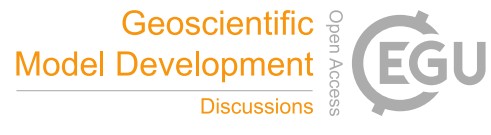

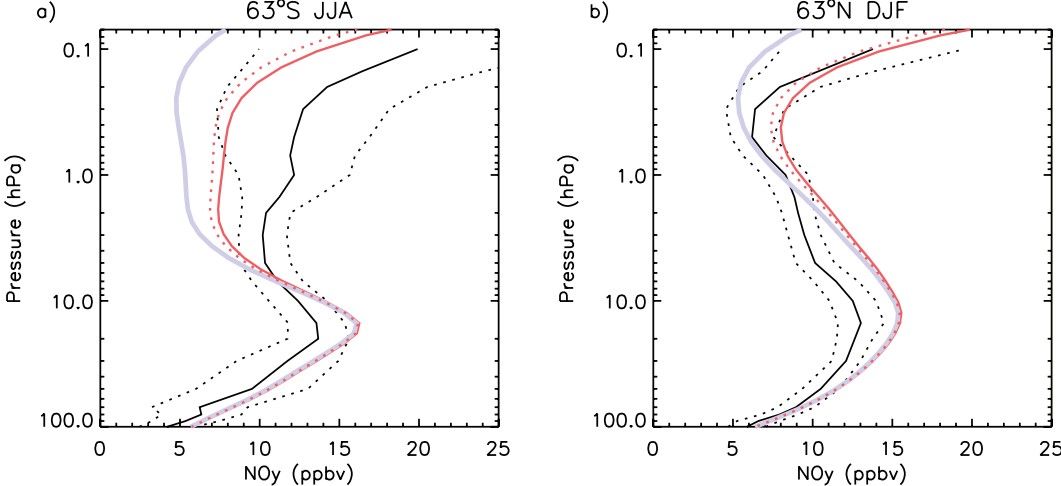

**Figure 12.** Comparison of 2004–2009 wintertime polar $NO_y$ climatology between ACE-FTS observations and SD-WACCM simulations. Solid black line is ACE, black dots are the average standard deviation of the monthly means. Grey line is WACCM with weak transport of auroral $NO_y$ from the lower thermosphere and no mesospheric production by medium energy electrons (MEE), dot-red line is stronger $NO_x$ transport but no production by MEE, red line is stronger $NO_x$ transport and production by MEE.

contribution is larger. However, it should be noted that 2004–2009 was a period of weak MEE in general, and during other periods of stronger MEE the contributions become more equal such that the effect on model $NO_y$ is stronger (not shown).

**Mid-Energy Electrons (MEE) from the Radiation Belts**

Highly energetic particles trapped in the radiation belts mainly consist of electrons and protons, forming inner and outer belts

separated by a "slot" region (Van Allen and Frank, 1959). The outer radiation belt (located 3.5–8 Earth radii from the Earth's centre) is highly dynamic, with electron fluxes changing by several orders of magnitude on timescales of hours to days (e.g. Morley et al., 2010). These changes are caused by the acceleration and loss of energetic electrons, through enhancements in radial diffusion and wave-particle interactions, during and after geomagnetic storms (e.g. Reeves et al., 2003). Storm-driven dynamic variations in the underlying cold plasma density influence the effectiveness of such processes in different regions of

the inner magnetosphere (e.g. Summers et al., 2007).

In order to characterise the electron precipitation into the atmosphere since 1850 it is necessary to develop a model that uses in-situ satellite observations from the modern era. The most comprehensive, long-duration, and appropriate set of observations are provided by the NOAA SEM-2 MEPED instrument onboard the POES spacecraft (Evans and Greer, 2000; Rodger et al., 2010a). The MEPED instrument covers an energy range from 50 eV to 2700 keV. In this study we are primarily concerned with

measurements made with the three medium energy integral electron detectors, i.e., >30, >100, and >300 keV, as the lower "auroral" energy range has been well characterised in previous work. The SEM-2 instrument has been flown on low-Earth orbiting (∼800 km) Sun-synchronous satellites since 1998, with up to 6 instruments operating simultaneously on occasion.



Electron precipitation fluxes from the outer radiation belt are measured with the 0 °detectors, which are mounted approximately parallel to the Earth-centre-to-satellite vector.

Improved calibration of the SEM-2 detectors has been undertaken by Yando et al. (2011) using modelling techniques contained in the GEANT-4 code to determine the detector geometric conversion factor, or detector efficiency (following the original work described in Evans and Greer, 2000). Further treatment of the data requires correction for the false counts caused by incident proton fluxes, which we undertake using the technique described in Lam et al. (2010). These calibration and corrections have been tested through comparison with other satellite (e.g. Whittaker et al., 2014b) and ground-based observations (e.g. Rodger et al., 2013; Neal et al., 2015). We convert the satellite position into the geomagnetic latitude parameter L (McIlwain, 1961) using the International Geomagnetic Reference Field IGRF (see Appendix B), and bin the precipitating flux data into zonal means with 0.25 L resolution from L = 2–10 (40–75 °geomagnetic latitude).

Using observed electron flux data in 2002–2012, a precipitation model for radiation belt electrons was created by van de Kamp et al. (2016). The precipitation model was fit to the corrected observations of the MEPED/POES detectors following the approach outlined in Whittaker et al. (2014a). In the CMIP6 application of this model, the Ap index is used as the driving input parameter. Ap defines the level of magnetospheric disturbance and the location of the plasmapause, both of which are needed to calculate precipitating electron fluxes at different magnetic latitudes. Thus, the reconstructed Ap record, as described earlier, can be readily used to create a continuous electron precipitation time series for the whole CMIP6 period. As output, the model provides daily spectral parameters of precipitation: integrated flux at energies above 30 keV and a power-law spectral gradient. A test of high-energy resolved precipitating electron flux measurements made by the DEMETER satellite found that the power-law fit consistently provides the best representation of the flux (Whittaker et al., 2013). The model output has been shown to compare well with the spectral parameters derived from POES satellite data (van de Kamp et al., 2016).

An atmospheric ionisation data set has been calculated based on the Ap-based precipitation model, using a computationally fast ionisation parameterisation (Fang et al., 2010) and atmospheric composition from the NRLMSISE-00 model (Picone et al., 2002). This calculation considered MEE (30–1000 keV) with maximum energy deposition at altitudes between about 60 and 90 km (van de Kamp et al., 2016). Note that the ionization parameterisation does not consider the contribution of Bremsstrahlung which could be significant only at altitudes below 50 km (Frahm et al., 1997).

Figure 13 shows examples of solar cycle variability of the modeled atmospheric MEE ionization rates at ≈80 km altitude. At 68° magnetic latitude (L shell 7.25), MEE precipitation is mostly driven by magnetic substorms and the solar cycle variability is relatively weak, except in around 2009 and the mid 1960s when extended periods of very low geomagnetic activity occurred. At 64° (L shell 5.25), precipitation is driven by high-speed solar wind streams. A more clear solar cycle variability can be seen with maximum ionization lagging the sunspot maximum by 1–2 years. At 56° (L shell 3.25), precipitation is mainly driven by coronal mass ejections which lead to more of an event-type behavior. Relatively infrequent ionization peaks are contrasted with long periods of very low ionization. Similar behavior is seen at other altitudes as well (not shown).

In the following, we demonstrate with examples the MEE impact in WACCM simulations. The purpose is to present a proof of concept, i.e., show that the MEE ionization data set can be used in chemistry-climate modeling, and is producing the expected direct effect in the mesosphere. We simulated the 2002–2012 period, including the Ap-driven MEE ionisation rates,





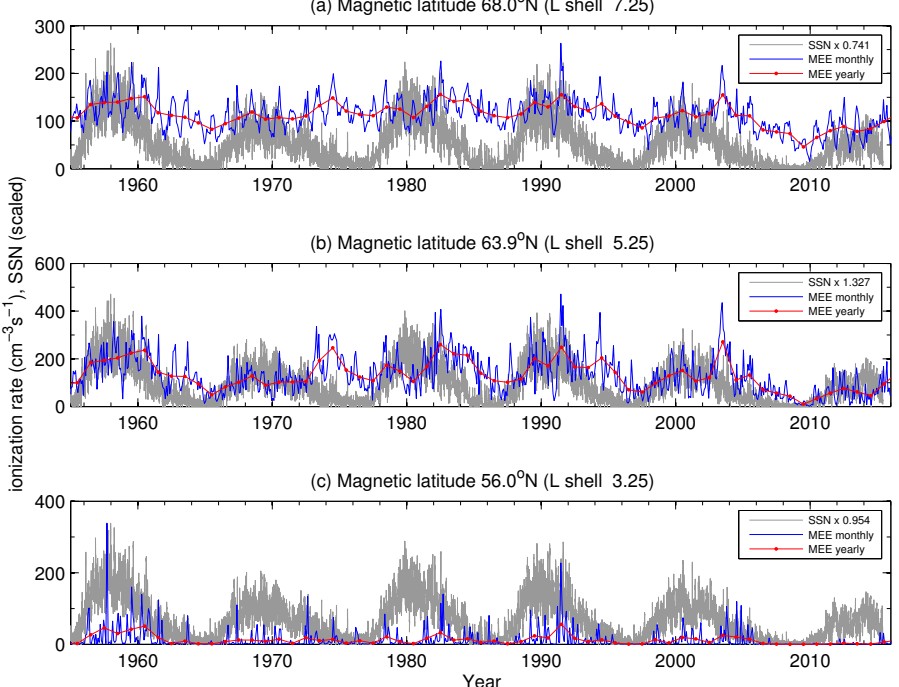

**Figure 13.** Examples of solar cycle variability of modeled, Ap-driven MEE ionization at ≈80 km altitude, with comparison to the sunspot number variability (SSN, scaled).

and analysed mesospheric OH and ozone responses at 0.040–0.015 hPa (approx. 70–80 km in altitude). This altitude region was selected because of the clear and direct MEE impact seen in satellite observations (e.g. Andersson et al., 2014a, b; Fytterer et al., 2015b; Sinnhuber et al., 2016). WACCM version 4 (see above) was used with $1.9° \times 2.5°$ horizontal resolution extending from the surface to $5.9 \times 10^{-6}$ hPa (≈140 km geometric height) in the specified dynamics mode, nudged to MERRA reanalysis

at every dynamics time step below about 50 km.

Figures 14a and 14b show global differences in yearly median OH mixing ratios due to MEE. Distinct features on the map are the stripes of enhanced values at magnetic latitudes between $55°$ and $75°$ (both hemispheres) which connect through the magnetic field to the outer radiation belt. The impact decreases from 2005 to 2009 due to the decline in geomagnetic activity and MEE precipitation (as shown in Figure 13). These features are of expected quality and magnitude, and similar to those

based on Microwave Limb Sounder (MLS) data analysis (Andersson et al., 2014b).

Figures 14c and 14d show relative differences in wintertime (MJJA) mean ozone due to MEE in the Southern Hemisphere. As expected, ozone is affected at high polar latitudes. In 2009, when MEE precipitation was weak, a maximum of 5–10% decrease is seen near the south pole relative to a reference WACCM simulation. In 2005, with much stronger MEE precipitation, the effect reaches up to 10–20% and covers the whole polar cap above about 60 °latitude. The magnitude of the 2005 response,

tens of percent, is comparable to that seen in MLS observations (Andersson et al., 2014a).





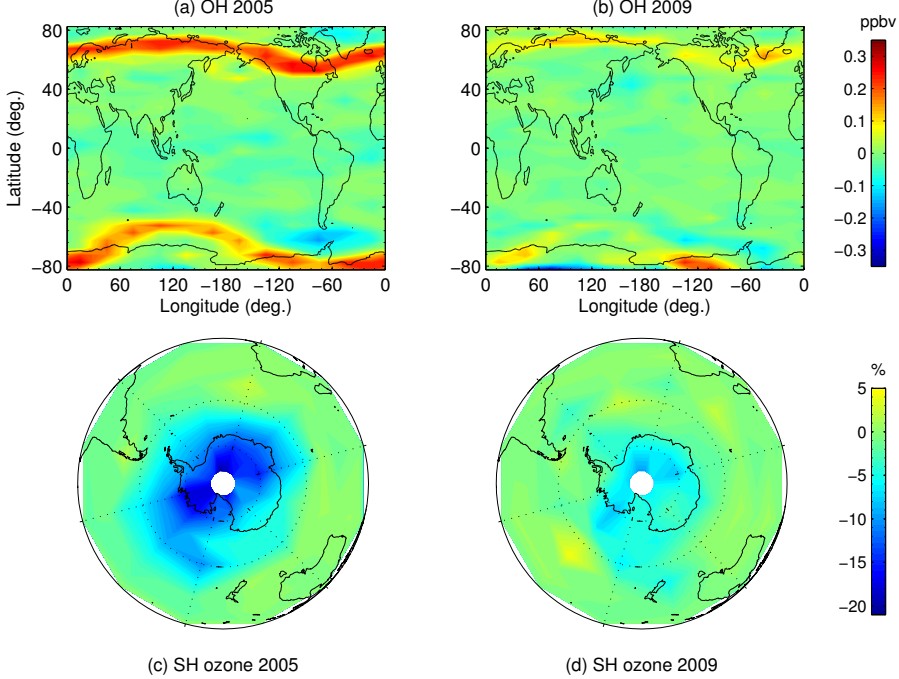

**Figure 14.** (a) and (b) Difference in yearly median OH mixing ratios at about 70–80 km between SD-WACCM runs with and without MEE ionization. (c) and (d) Relative differences in Southern Hemispheric wintertime mean $O_3$ at about 70–80 km between SD-WACCM runs with and without MEE ionization.

## The EPP Indirect Effect: Odd Nitrogen Upper Boundary Condition

Those models with their upper lid in the mesosphere, i.e., which do not represent the entire EPP source region, require an odd nitrogen upper boundary condition (UBC), accounting for EPP productions higher up, in order to allow for simulating the introduced EPP indirect effect in the model domain. Odd nitrogen UBCs have been previously used in CCMs. In some model studies, the UBC was taken directly from $NO_x$ observations (e.g., Reddmann et al., 2010; Salmi et al., 2011), which, however, implies the restriction to the relatively short time period spanned by the observations. In other cases, a simple parameterisation in dependence of the seasonally averaged Ap index (Baumgaertner et al., 2009) was employed (e.g., Baumgaertner et al., 2011; Rozanov et al., 2012), enabling extended simulations over multi-decadal time periods. We recommend the use of the UBC model described in Funke et al. (2016) which is designed for the latter application and represents an improved parameterisation due to its more detailed representation of geomagnetic modulations, latitudinal distribution, and seasonal evolution. This semi-empirical model for computing time-dependent global zonal mean $NO_y$ concentrations (in units of $cm^{-3}$) or EPP-$NO_y$ molecular fluxes (in units of $cm^{-2}\,s^{-1}$) at pressure levels within 1–0.01 hPa and is available at http://solarisheppa.geomar.de/solarisheppa/cmip6.





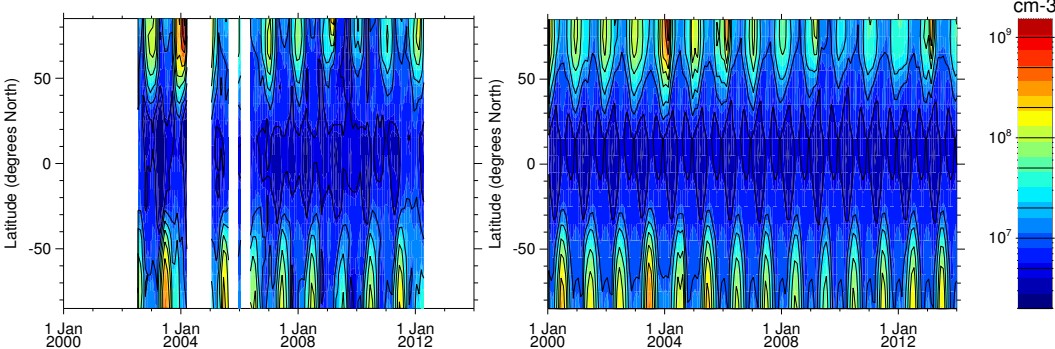

**Figure 15.** Latitude-time sections of $NO_y$ concentrations observed by MIPAS (left) and from the UBC model (right) at 0.1 hPa.

The UBC model has been trained with the EPP-$NO_y$ record inferred from Michelson Interferometer for Passive Atmospheric Sounding (MIPAS) observations (Funke et al., 2014a). Inter-annual variations of the EPP indirect effect at a given time of the winter are related to variations of the EPP source strength, the latter being considered to depend linearly on the Ap index. A finite impulse response approach is employed to describe the impact of vertical transport on this modulation. Interannual vari-

ations of the EPP-$NO_y$ seasonal dependence, driven by variations of chemical losses and transport patterns, are not considered in the standard mode of the UBC model. Optionally, episodes of accelerated descent associated with Elevated Stratopause (ES) events in Arctic winters can be considered by means of a dedicated parameterisation, taking into account the dependence of the EPP-$NO_y$ amounts and fluxes on the event timing (Holt et al., 2013). Although its application is recommended in principle, we note that it requires the implementation of the UBC model into the climate model system since ES events cannot be predicted

in free-running model simulations. Further, the ES detection criterion might need to be tuned for each individual model system.

We recommend to prescribe $NO_y$ concentrations, as this has already been tested successfully in a CCM. As an example, Fig. 15 shows the $NO_y$ concentrations from the UBC model at 0.1 hPa in comparison with the MIPAS observations. Care has to be taken when balancing $[NO_y] = [NO] + [NO_2] + [NO_3] + [HNO_3] + 2[N_2O_5] + [ClONO_2]$ in order to avoid model artifacts at the upper boundary (primarily triggered by the loss reaction of $NO_2$ with atomic oxygen). The simplest way to

achieve this is to set $[NO] = [NO_y]$ while forcing the concentrations of all other $NO_y$ species to be zero. Note that below the vertical domain where $NO_y$ is prescribed, MEE ionization still might occur and its consideration (as described before) is recommended. However, its consideration should be strictly limited to this vertical range since at and above the UBC, MEE is already implicitly accounted for by the prescribed $NO_y$ from the observation-based UBC model.

The UBC was tested in the EMAC CCM version 2.50 (see also Sec. 2.1.2 and Jöckel et al., 2010) with a T42L90 resolution.

$NO_y$ concentrations were prescribed as NO in the uppermost four model boxes at pressure levels from 0.09 to 0.01 hPa. $NO_y$. There, $NO_2$ was set to zero to suppress artificial $NO_2$ buildup. The model was run from 1999 to 2010 in the specified dynamics mode, nudged to ERA-Interim reanalysis data (Dee et al., 2011) below 1 hPa. A special treatment of ES events was disabled in the UBC model and SPEs were not considered. A comparison of polar $NO_y$ from EMAC with MIPAS observations is shown





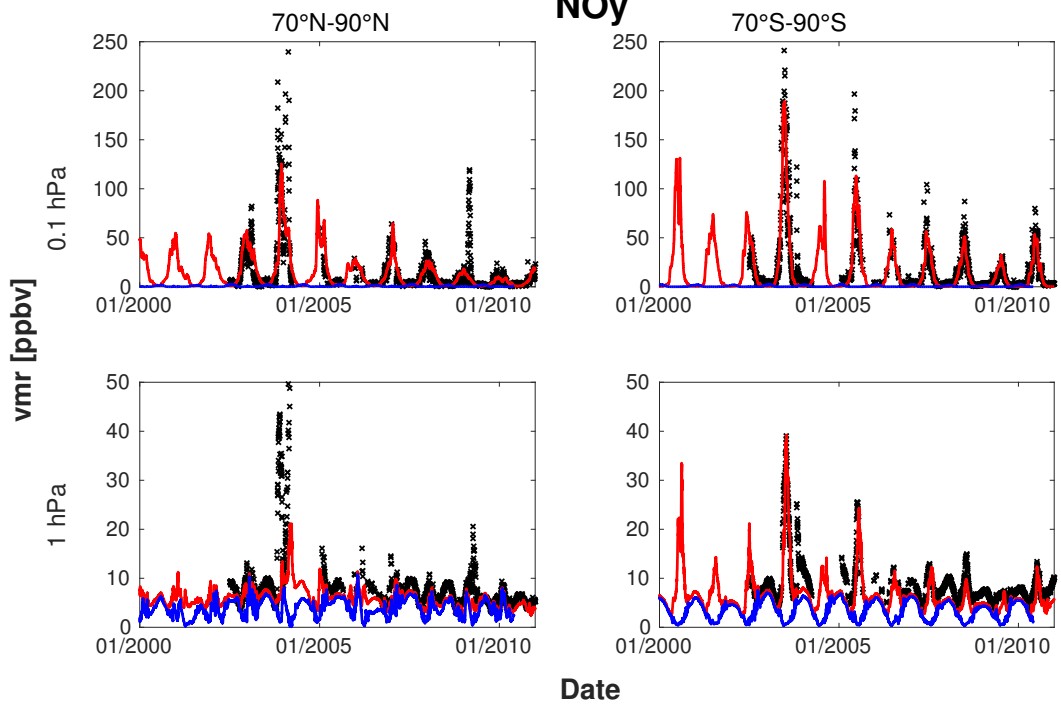

**Figure 16.** Comparison of $NO_y$ from MIPAS observations and different EMAC model runs at 70–90°S (left) and 70–90°N (right) for 0.01 hPa (upper panel) and 0.1 hPa (lower panel), from 2000–2010. Black crosses: MIPAS observations. Red line: EMAC with the MIPAS-derived UBC for $NO_y$ (see text); blue line: EMAC without UBC for $NO_y$.

in Fig. 16 for 0.1 hPa (just below the prescription altitudes) and 1 hPa. A very good agreement between model predictions and observations is found at 0.1 hPa in both hemispheres, with the exception of periods of large SPEs (October/November 2003) in both hemispheres and ES events (January 2004 and February 2009) in the Northern Hemisphere. At 1 hPa, the agreement is still very good during winter, but EMAC underestimates the summer maximum of $NO_y$ slightly. This is also observed in the

5  base model run without employing the UBC (see Fig. 16).

The interannual variation of ozone in the stratosphere and lower mesosphere has been investigated in this model in a similar way as for a three-satellite composite (Fytterer et al., 2015a). The ozone difference between Austral winters with high and low geomagnetic activity during 2005–2010 is shown in Fig. 17 for 27-day running means relative to the mean of all years. This period has been chosen because of its low SSI variability. Cross-correlations between SSI and particle impact are thus

10  minimized. EMAC results are in excellent agreement with the observations as provided in Fytterer et al. (2015a, Figure 5), showing a clear negative ozone anomaly of 5-10% moving down from the upper stratosphere to below 10 hPa (∼30 km) from July to October. Below, a positive anomaly of smaller amplitude is observed both in EMAC and the three-satellite composite



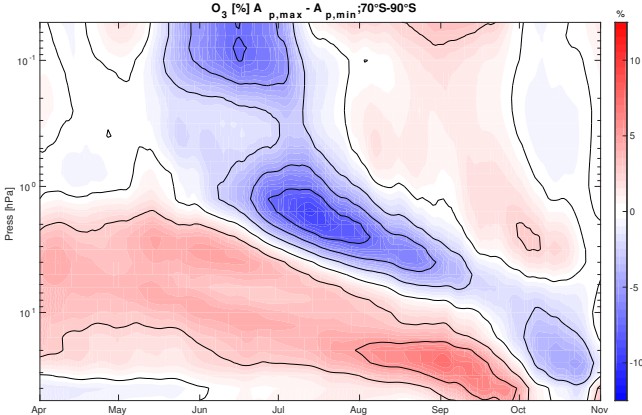

**Figure 17.** Ozone interannual variation due to geomagnetic forcing in 2005-2011 from EMAC model run using the MIPAS-derived UBC for $NO_y$. Shown are 27-day running means of the mean of the three years with highest – mean of the three years with lowest geomagnetic activity averaged over 70-90°S. EMAC results are in excellent agreement with $O_3$ observations using a three-satellite composite for the same period of time (see Figure 5 in Fytterer et al., 2015a).

which might be due to a combination of self-healing, dynamical feedbacks, and chemical feedbacks ($NO_x$-induced chlorine buffering in the processed "ozone hole" area).

### 2.2.2 Solar Protons

Solar eruptive events sometimes result in large fluxes of high-energy solar protons at the Earth, especially near the maximum
and declining periods of activity of a solar cycle. This disturbed time, wherein the solar proton flux is generally elevated for a few days, is known as a solar proton event (SPE). Solar protons are guided by the Earth's magnetic field and impact both the northern and southern polar cap regions (>60° geomagnetic latitude, e.g., see Jackman and McPeters, 2004). These protons can impact the neutral middle atmosphere (stratosphere and mesosphere) and produce both hydrogen radicals and reactive nitrogen constituents.

The ozone response due to very large SPEs is fairly rapid and substantial and has been observed during and after numerous events to date (e.g., Weeks et al., 1972; Heath et al., 1977; McPeters et al., 1981; Thomas et al., 1983; Solomon et al., 1982; McPeters and Jackman, 1985; Jackman et al., 1990, 1995, 2001, 2005b, 2008, 2011, 2014; López-Puertas et al., 2005a; Rohen et al., 2005; Seppälä et al., 2006; Krivolutsky et al., 2008; Funke et al., 2011; von Clarmann et al., 2013). Ozone within the polar caps (60-90°S or 60-90°N geomagnetic) is generally depleted to some extent in the mesosphere and upper stratosphere
(e.g., Jackman et al., 2005b) within hours of the start of the SPE and can last for months beyond the event at lower altitudes in the stratosphere.

Decreases in mesospheric and upper stratospheric ozone are mostly caused by SPE-induced $HO_x$ increases, which were predicted to occur over 42 years ago (e.g., see Swider and Keneshea, 1973). Direct measurements of SPE-caused OH and



$HO_2$ enhancements have confirmed these early predictions (e.g., Verronen et al., 2006; Damiani et al., 2008; Jackman et al., 2011, 2014). Other observations of increased $H_2O_2$ (Jackman et al., 2011) and of chlorine-containing constituents HOCl (an increase, see von Clarmann et al., 2005; Jackman et al., 2008; Damiani et al., 2008, 2012; Funke et al., 2011) and HCl (a decrease, see Winkler et al., 2009; Damiani et al., 2012) support the SPE-caused $HO_x$ enhancement theory. Since $HO_x$

constituents have relatively short lifetimes (hours), these SPE-enhanced species have only a short-term impact on ozone.

The SPE-induced $NO_y$ enhancements, on the other hand, cause a much lengthier reduction in ozone, given their much longer atmospheric lifetime ($\sim$months) in the stratosphere. SPE-caused $NO_x$ increases have been shown in several studies (e.g., McPeters, 1986; Zadorozhny et al., 1992, 1994; Randall et al., 2001; López-Puertas et al., 2005a; Jackman et al., 1995, 2005b, 2008, 2011, 2014; Funke et al., 2011; von Clarmann et al., 2013; Friederich et al., 2013). Other $NO_y$ constituents like

$HNO_3$, $HNO_4$, $N_2O_5$, and $ClONO_2$ (e.g., López-Puertas et al., 2005b; Jackman et al., 2008; Funke et al., 2011; Damiani et al., 2012; von Clarmann et al., 2013) as well as the total $NO_y$ family (e.g., Funke et al., 2011, 2014a, b) have also been shown to increase as a result of large SPEs. Additionally, $N_2O$ has been measured to increase as a result of large SPEs (Funke et al., 2008; von Clarmann et al., 2013).

Solar proton fluxes have been measured by a number of satellites in interplanetary space or in orbit around the Earth.

The National Aeronautics and Space Administration (NASA) Interplanetary Monitoring Platform (IMP) series of satellites provided measurements of proton fluxes from 1963–1993. IMPs 1-7 were used for the fluxes from 1963-1973 (Jackman et al., 1990) and IMP 8 was used for the fluxes from 1974–1993 (Vitt and Jackman, 1996). The National Oceanic and Atmospheric Administration (NOAA) Geostationary Operational Environmental Satellites (GOES) were used for proton fluxes from 1994–2014 (e.g., Jackman et al., 2005a, 2014).

Other precipitating particles are associated with SPEs, besides protons. These include alpha particles, which comprise, on average (but this value may vary from event to event) about 10% of the positively charged solar particles, other ions, which account for less than 1% of the remainder, and electrons (e.g., Mewaldt et al., 2005). Only solar protons are included in energy deposition computations given in this paper. Please note that other charged particles could add modestly to this energy deposition in the middle atmosphere during SPEs.

The proton fluxes of energies 1-300 MeV were used to compute daily average ion pair production profiles using an energy deposition scheme first discussed in Jackman et al. (1980). The scheme includes the deposition of energy by the protons and assumes 35 eV are required to produce one ion pair (Porter et al., 1976). Note that this approach misses development of the atmospheric cascade (Sec. 2.2.3). This process, crucial for GCRs, is minor for SPEs in the upper atmosphere but may contribute modestly to the energy deposition in the lower stratosphere.

The dataset for daily average ion pair production rates at 60–90° geomagnetic latitudes from SPEs was computed over a 52 year time period (1963–2014), when proton flux measurements from satellites were available. A longer-term dataset for these SPE-caused ion pair production rates was created for the 1850–1962 time period using activity levels of the measured sunspots over the solar cycles. SPEs are much more frequent during years of maximum solar activity and vice versa. This longer-term dataset was reconstructed for years 1850–1962 in a random way using solar activity levels combined with the



52-year calculated SPE-caused ion pair production. Thus, an historical record of atmospheric forcing by SPEs in the form of a daily average ion pair production rate is available over the entire period 1850–2014 for use in global models.

### 2.2.3 Galactic Cosmic Rays

The Earth's atmosphere is continuously irradiated by Galactic cosmic rays (GCR), which consist mostly of protons and $\alpha$-particles with a small amount of heavier fully ionized species up to iron and beyond. These cosmic rays originate from galactic (mostly supernova shocks) and exotic extra-galactic sources and may have an energy up to $10^{20}$ eV but the bulk energy is in the range of several GeV/nucleon. While the GCR flux can be assumed (at time scales shorter than thousands of year) constant and isotropic in the interstellar space, it is subject to strong modulations within the heliosphere (the region of about 200 AU across hydromagnetically controlled by the solar wind and the heliospheric magnetic field). This modulation is driven by solar magnetic activity – the stronger the solar activity, the lower is the GCR flux near the Earth. This flux is often described by the so-called force-field model (Caballero-Lopez and Moraal, 2004) parameterized via the time-variable modulation potential $\phi$ and the fixed shape of the local interstellar spectrum (see, e.g., Usoskin et al., 2005, for more details). Typically, the value of the modulation potential is defined by fitting data from the world-wide network of ground-based neutron monitors calibrated to fragmentary space-borne measurements of GCR energy spectra. These data are available since 1951 or, with caveats of using the ground-based ionization chambers, since 1936 (Usoskin et al., 2011).

Before impinging on the Earth's atmosphere, GCR are additionally deflected by the geomagnetic field so that there is no shielding in the polar regions, but particles must have rigidity (the ratio of a particles momentum and charge) exceeding 15–18 GV to be able to penetrate in equatorial regions. Since fast transient solar energetic particle events often occur at the background of enhanced geomagnetic disturbances, straight-forward computation of the particle trajectories in a realistic geomagnetic field is needed (e.g., Smart et al., 2000). However, for slowly changing GCR variability, this shielding is often parameterized in the form of the effective geomagnetic rigidity cutoff, so that particles with rigidity/energy exceeding the cutoff can penetrate to the atmosphere at a given location while less energetic particles are fully rejected (Cooke et al., 1991). When energetic cosmic rays enter the atmosphere, they initiate a nucleonic-muon-electromagnetic cascade in the atmosphere, ionizing ambient air. As a sub-product of this cascade cosmogenic isotopes such as $^{14}$C, $^{10}$Be and others can be produced. These cosmogenic isotopes are long-lived and can be used for the reconstruction of solar activity over several thousand years (see Sec. 3).

Between the surface and 25—30 km cosmic rays are the main source of atmospheric ionization (Mironova et al., 2015) causing the production of $NO_x$ and $HO_x$. The influence of GCRs on atmospheric chemistry has been investigated in several model studies (Krivolutsky et al., 2002; Calisto et al., 2011; Rozanov et al., 2012; Mironova et al., 2015; Jackman et al., 2015). GCR-induced ozone reductions of more than 10% in the tropopause region and up to a few percent in the polar lower stratosphere have been reported. The potential impact on surface climate has been studied by Calisto et al. (2011) and Rozanov et al. (2012).

The process of development of the atmospheric cascade, initiated by energetic cosmic rays, is complicated and needs to be modelled using direct Monte-Carlo simulations of all the processes involved in the development of the cascade, including all types of interactions, scattering and decay of various species. We note that older models based on empirical parameterisations

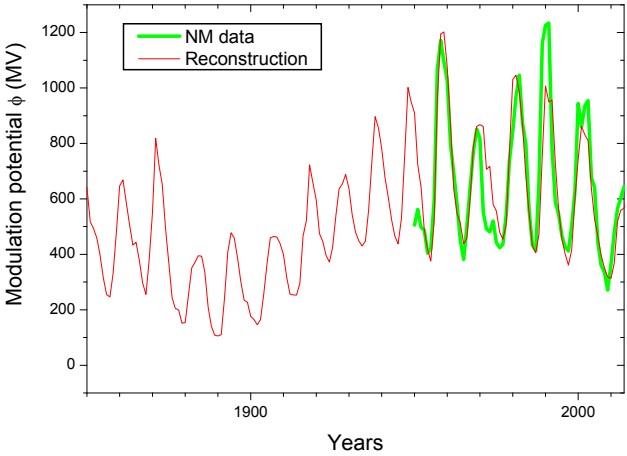

**Figure 18.** Time series of the reconstructed heliospheric modulation potential $\phi$ including solar cycle variations. The thick green line is the modulation potential reconstructed for the period 1951–2014 using data from the worldwide neutron monitor (NM) network (Usoskin et al., 2011).

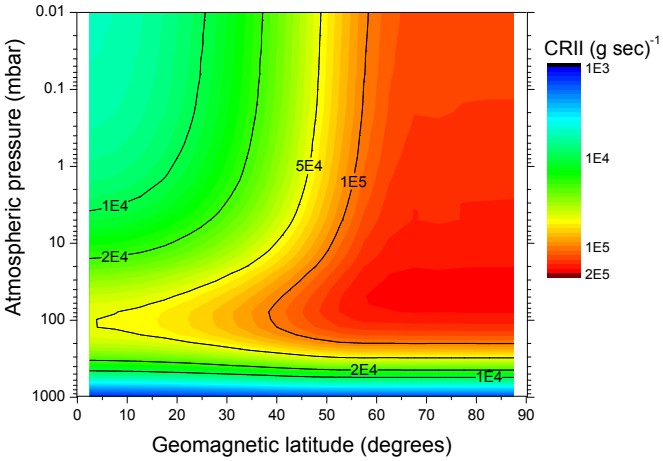

**Figure 19.** Calculated annual mean ion pair production rate for the year 2014 as a function of barometric pressure and geomagnetic latitude. Computations were done using the CRAC:CRII model.

or on solution of Boltzmann-type equations may introduce significant biases in the results, especially in the lower atmosphere. Accordingly, we use a full Monte-Carlo model CRAC:CRII (Usoskin and Kovaltsov, 2006; Usoskin et al., 2010) based on the CORSIKA Monte-Carlo package. A similar result can be obtained with the PLANETOCOSMIC (Desorgher et al., 2005) based on the GEANT package. The agreement between the two models has been verified (Usoskin et al., 2009) to be within 10%.



GCR ion pair production rates are provided as a function of the barometric pressure and geomagnetic latitude and were calculated from the modulation potential values $\phi$ of the 9400-year long record by Steinhilber et al. (2012). Since this dataset has a 22-year time resolution, it has been interpolated to interannual time scales to resolve individual solar cycles, based on the sunspot numbers (see Fig. 18). One can see that this agrees well with the values of $\phi$ reconstructed using data from the worldwide network of neutron monitors (NM) (Usoskin et al., 2011). An example of the calculated ionisation rate is shown in Fig. 19. The ionisation maximizes in polar regions at heights of 15–20 km, while in the equatorial region the maximum of ionisation occurs at about 12 km (note that in case of using ionization per $cm^3$, the ionisation maximizes at about 10 km in the equatorial region and 12 km over the poles).

### 2.2.4 Implementation of Chemical Changes Induced by Particle-Induced Ionisation

MEE, SPE, and GCR-induced atmospheric ionisation is expressed in the CMIP6 forcing dataset in terms of ion pair production rates (IPR).Note that IPR data are provided in units of ion pairs per gram per second as a function of the barometric pressure. These units are natural for the ionisation processes and are mostly independent of the atmospheric conditions. Conversion into units of $cm^{-3}\,s^{-1}$ (by multiplying with mass density) should be done on the model grid ideally at each time step, but at least once per day. Recommendations for the projection of ion pair production rates onto geographic coordinates can be found in Appendix B.

Particle-induced ionisation causes, along with the generation of the ion pairs, the production of $NO_x$ and $HO_x$. As a basic approach, we recommend to consider these $NO_x$ and $HO_x$ productions in CCMs with interactive chemistry by using the parameterisations provided by Porter et al. (1976) and Solomon et al. (1981), respectively. More detailed information about these approaches is provided in Appendix C and D. Recommendations for the implementation of EPP effects on minor species are given in Appendix E.

## 3 Future Scenarios (2015-2300)

One of the key questions in the CMIP6 project is our ability to assess future climate changes given climate variability, predictability and uncertainties in scenarios. In CMIP5, climate projections were based on a stationary Sun scenario, obtained by simply repeating solar cycle 23, which ran from April 1996 till June 2008 (Lean and Rind, 2009). In CMIP6, we include a more realistic solar forcing, and provide two different scenarios:

- a reference (REF) scenario with the most likely level of solar activity;

- an extreme (EXT) scenario with an exceptionally low level of solar activity, corresponding to the lower 5th percentile of all forecasts. This extreme scenario is meant to be used for sensitivity studies.

We ignore scenarios with high levels of solar activity because the Sun just left such an episode (called grand solar maximum), and several studies suggest that it is very unlikely to return to one in the next 300 years (Abreu et al., 2008; Barnard et al., 2011; Steinhilber et al., 2012).



The main challenge consists in forecasting solar activity up to 2300. Ever since the solar cycle was first observed, people have been trying to predict what future cycles may look like. Prediction methods were empirical, and at best could give some clue of what the amplitude of the next cycle could be (Petrovay, 2010). This situation prevailed until the early 21st century, when physical models of the magnetic dynamo that drives solar activity started unveiling a more realistic picture (Charbonneau,

2010). Many were confident that in a near future one would be able to predict the solar cycle several decades ahead. The unusually long solar cycle number 23 that ended in 2009, and the weak one (nr. 24) that followed came as a surprise, and manifested our evident lack of understanding of the solar cycle. As of today, even predicting the cycle amplitude one cycle ahead remains a major challenge (Pesnell, 2012).

In this context, predicting solar activity several tens of cycles ahead may seem like a hopeless task. However, erratic as the

solar cycle may be, solar activity on multi-decadal time scales (i.e. averaged over solar cycles) does show some regularity, which may be used to predict it, see for example (Hanslmeier and Brajša, 2010) and references therein. The solar cycle is driven by the solar dynamo, by which the dynamical interactions of flows and magnetic fields in the solar convection zone lead to periodic reversals of polarity of the solar magnetic field (Charbonneau, 2010). One of its consequences is the emergence of regions with enhanced magnetic field, namely sunspots, whose number are the most widely known proxy for solar activity. During

that emergence process, the predominantly toroïdal magnetic field generates a dipole moment, which in turns generates a new toroïdal magnetic component through rotational shearing. Because these inductive processes are operating in the turbulent environment of the solar convection zone, memoryless stochastic forcing of the dynamo is certainly presentat some level. Nonetheless, memory effects associated with these periodic reversals play a major role in determining solar variability on multi-decadal time scales, and to some degree are decoupled from the short-term variability. This is our prime motivation for

considering predictions on multi-decadal time scales.

There are two approaches for constraining such predictions. One is to learn from solar dynamo models, and the other is to infer from past variations of solar activity. Recent years have witnessed significant advances in solar dynamo modeling, and the development of several physical models (Charbonneau, 2014). Most models exhibit some persistence in the solar cycle-averaged level of activity, with a memory of up to a few cycles. However, among models that do succeed in producing deep

activity minima similar to the Maunder minimum, most show onsets occuring surprisingly fast, typically within one or two cycles.

The best gauge of past solar variability is the production rate of the [14]C and [10]Be cosmogenic isotopes, as already described in Sec. 2.2.3. The level of activity is usually expressed in terms of the modulation potential $\Phi$ (Usoskin, 2008; Beer et al., 2012), which is intimately related to the open solar magnetic flux. There exist today different records of cosmogenic isotopes,

which are gradually improving as new observations are being added, and underlying assumptions, such as the strength of the geomagnetic dipole, are better constrained. Here, we consider the 9400-year long record by Steinhilber et al. (2012), which is a composite of [14]C and [10]Be data, and is available from http://www.ncdc.noaa.gov/paleo/forcing.html. The record is sampled every 22 years, and runs from 7439 BC till 1977 AD. For making better predictions, we want our historic observations to end as close as possible to the present. Therefore we extended the record from 1977 to 1999, using the geomagnetic reconstruction

of the open solar flux Lockwood et al. (2014). The geomagnetic reconstruction provides annual values of the open solar flux





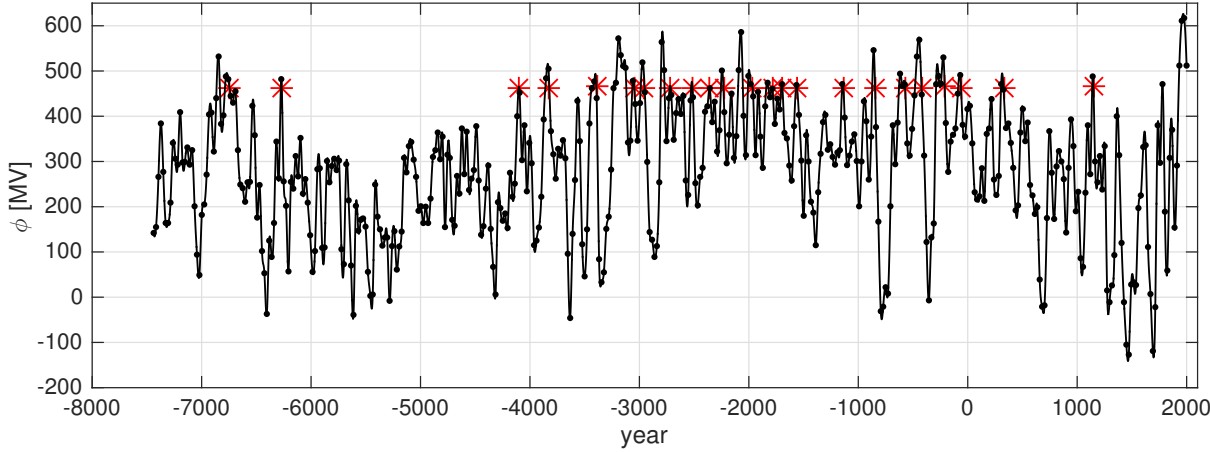

**Figure 20.** Modulation potential record $\Phi$ [MV] used for forecasting solar activity. What matters is the relative variation in $\Phi$, which reflects that in the TSI: large values of $\Phi$ correspond to grand solar maxima, whereas low values correspond to grand solar minima. Red stars refer to the events used in the analogue forecast, see Sec. 3.1.1.

back to 1845, and we extrapolate the linear regression between the 22-year boxcar smoothed geomagnetic reconstruction and the cosmogenic reconstruction to provide an estimate of the cosmogenic $\Phi$ in 1999.

Figure 20 displays the complete modulation potential record, which exhibits occasional periods of low solar activity (i.e. grand solar minima) separated by periods during which the fluctuations seem more erratic. It is noteworthy that the Sun was
more active during the recent decades (the Modern grand maximum) than during most of the other periods.

In the following, we shall consider three different (and to some degree, complementary) approaches for predicting $\Phi$ up to 300 years ahead, and use their weighted average as the most likely value. To convert these 22-year averages of $\Phi$ into quantities that are relevant for climate forcing, we first convert the 22-year averaged modulation potential into an average sunspot number. Historic solar cycles that have the same average sunspot number are subsequently stitched together to obtain a
record with daily-resolved sunspot numbers. Using the latter, we estimate the SSI, and particle forcing, as described in Sec. 3.5.

According to solar dynamo models, the solar-cycle averaged modulation potential (and the sunspot number) cannot be predicted more than a few decades ahead. The observed modulation potential has an autocorrelation function that decays exponentially with a characteristic time of $48 \pm 5$ years. This quantity can be interpreted as the time beyond which memory is lost. As we shall see below, our three prediction methods too exhibit prediction horizons of approximately 60 years. However,
one of them (the deterministic harmonic model) maintains substantial predictive capacity on much longer time scales. To the best of our knowledge, no existing method has been able to meaningfully predict solar activity more than 60 years ahead. For that reason, we shall from now on speak in terms of *forecast* rather than *prediction*, and concentrate on *scenarios* of solar activity.





## 3.1 Forecast Methods

Here we construct the reference (REF) and extreme (EXT) solar activity scenarios by applying three prediction methods to the heliospheric modulation potential time series produced by Steinhilber et al. (2012). Other prediction techniques exist which could have also been used, but we choose to use only these three techniques as we consider them to reflect a fair range

of possibilities for the future evolution of the heliospheric modulation potential, and it would be impractical to include an exhaustive set of techniques. Below we describe each of the prediction methods, before detailing how the two scenarios were constructed.

### 3.1.1 Analogue Forecast

The analogue forecast (AF) is calculated with a simple technique, known across disciplines by various names including com-

positing, superposed-epoch analysis, conditional sampling and Chree analysis. In a data sequence that exhibits a low amplitude response to a specific trigger event, the response may be obscured by sources of random variability. The AF technique aims to reveal the response to a specific trigger event by averaging the responses to many occurrences of the trigger event, such that over many events random variability will be suppressed and the response will emerge (Laken and Čalogović, 2013). Barnard et al. (2011) used this technique with the Steinhilber et al. (2012) $\Phi$ record to estimate the possible future $\Phi$ evolution given

the expected decline from the grand solar maximum that persisted through the late 20th century. Here we perform an updated version of the procedure employed by Barnard et al. (2011).

Defining grand solar maxima in the $\Phi$ record as any period above the $90^{th}$ percentile of the $\Phi$ distribution (462 MV), identifies 23 grand solar maxima in the $\Phi$ record prior to the most recent one. Here the declines from the grand solar maxima are used as the event triggers from which the AF is calculated, and these times are marked on the $\Phi$ time series shown in Figure

by red stars. Figure 20 also shows that the most recent values in the $\Phi$ record have not yet fallen below grand solar maxima threshold. Therefore, as the end date of the most recent grand solar maxima is not known, it must be estimated, to provide a date from which the AF applies. The grand solar maximum end date was estimated to be 2004, by extrapolating the regression of the 22-yr smoothed Lockwood et al. (2014) annual geomagnetic reconstruction of the open solar flux onto the Steinhilber et al. (2012) $\Phi$ record. So the forecast was applied from 2004 onwards and interpolated onto the dates required to continue

22-year sample sequence defined by the $\Phi$ record.

### 3.1.2 Autoregressive Model

Autoregressive (AR) models are widely used in time series forecasting (Box et al., 2015). These models assume that variations can be described by means of a linear stochastic difference equation, so that future values are expressed as a linear combination of present and past values. In our context, we have

$$\hat{\Phi}_{k+h} = a_1 \Phi_{k-1} + a_2 \Phi_{k-2} + \ldots + a_p \Phi_{k-p} \,, \tag{1}$$



where $\Phi_k$ is the heliospheric modulation potential (after subtracting its time average) at the $k$'th time step, and $\hat{\Phi}_{k+h}$ is its value predicted $h \geq 0$ time steps ahead. Since $\Phi_k$ is measured with a cadence of 22 years, each value of $k$ corresponds to a 22-year time step. AR models are capable of describing a variety of dynamical behavior, including oscillations, red noise, etc. The main free parameter is the model order $p$, for which there exist several selection criteria (Ljung, 1997). In our case, we

obtain $p = 20$. According to this value, our forecasts are based on observations that go back at most 440 years into the past.

Because AR models are linear, they cannot properly describe nonlinear dynamical effects such as the occasional occurrence of grand solar minima, which appear as a different mode of solar activity (Usoskin et al., 2014). To partly overcome this limitation, we train the model by considering time intervals whose conditions are similar to those prevailing at the end of the 20th century. More specifically, we train the model by using only observations that belong to either of the 23 time intervals

$[t_{GSM} - 1100\text{years}, t_{GSM} + 1100\text{years}]$ that are centered on the same occurrences $t_{GSM}$ of the 23 grand solar maxima as in the analogue forecast (see Sec, 3.1.1, and Fig. 20). We exclude observations that follow $t_{GSM}$ by up to 300 years in order to give us a means for testing the prediction on a time interval that is (mostly) independent of the one the model has been trained on. The only exception in this list is the last grand solar maximum of the late 20th century, for which we do not have future observations available. The 2200-year duration of the time interval is the shortest one below which the performance of the AR

model starts degrading.

Using AR models, we now forecast the heliospheric potential 22, 44, . . . , 308 years ahead by training a different model for each value of the forecast horizon $h$ in Eq. 1. The forecast error, which is the usual metric for describing forecast performance, is classically defined as

$$s(h) = \sqrt{\left\langle \left( \hat{\Phi}_{k+h} - \Phi_{k+h} \right)^2 \right\rangle} \, , \tag{2}$$

where the ensemble average $\langle \ldots \rangle$ runs over all 23 grand solar maxima. Clearly, the AR model can be improved in several ways. One of them consists in modelling the full record of the heliospheric potential, and use threshold AR models to account for mode changes. These issues will be addressed in a forthcoming publication.

### 3.1.3  Harmonic Model

Several studies have reported the existence of periodicities in cosmogenic solar proxies, with outstanding periods of approxi-

mately 87 years (known as the Gleissberg cycle), 208 years (de Vries cycle), 350 years, and more (McCracken et al., 2013). The origin of these elusive periodicities has been hotly debated, and is beyond the scope of our study. Steinhilber and Beer (2013) successfully used them to model solar activity on multi-decadal time scales, and produced a 500 year forecast of the heliospheric potential. We consider the same approach, and thus assume that the dynamical evolution of the heliospheric potential obeys a deterministic model

$$\hat{\Phi}_k = b_0 + \sum_{i=1}^{N} \left( b_i \sin(2\pi t_k / T_i) + c_i \cos(2\pi t_k / T_i) \right) \, . \tag{3}$$

We parameterize and train this harmonic model in a way that is similar to the preceding AR model. First, we select 2600-year intervals that are centered on the timings $t_{GSM}$ of each of the 23 grand solar maxima, and exclude the 300 years that





follow each $t_{GSM}$. In contrast to the AR model, however, we estimate the model coefficients separately for each interval in order to account for possible phase drifts. To select the periods $T$, and reduce their number, we start from an initial set of $N = 19$ periods of less than 2200 years, taken either from McCracken et al. (2013), or obtained from spectral analysis. We then estimate the forecast error after discarding one period at a time, only keeping those that do not lead to a significant increase
of the forecast error. Finally, we end up with a set of 12 periods of {88, 105, 130, 150, 197, 208, 233, 285, 353, 509, 718, 974} years. Likewise, the forecast error is used to fix the 2600-year duration of the intervals. Longer intervals give a better statistic, but result in a poorer fit because of possible phase drifts in short-period oscillations.

### 3.2   Summary of Forecasts

Figure 21A shows the results of the AF, AR and HM forecasts, as well as the observed $\Phi$ record from 1845-1999. All three
methods forecast a decrease in solar activity out to approximately 2100. In the HM model, oscillations with largest amplitudes occur, on average, at 88, 208, and 285 years, and so periodicities are clearly present in the HM forecast. In contrast, forecasts obtained from the AF and AR models tend to converge toward a climatological mean.

### 3.3   Prediction Errors

The error of each prediction method was assessed with a bootstrap approach. Defining grand solar maxima as any period in the
$\Phi$ record larger than the $90^{th}$ percentile of the $\Phi$ distribution, there are 23 other grand solar maxima in the $\Phi$ record prior to the one that persisted through the late 20th century. For each prediction method, hindcasts were made for the 308 years following the decline from each prior grand solar maximum. For each method and each grand solar maximum, the models were trained analogously to the descriptions above, such that no $\Phi$ data from within the prediction window is used to generate each hindcast. The typical error in each prediction method as a function of prediction horizon was then calculated as the root mean square of
the error of the 23 hindcasts at each prediction horizon.

Although not used in the scenario construction, a simple persistence forecast and the corresponding error was also calculated, to serve as a benchmark to compare the AF, AR and HM methods against. The typical prediction error as a function of prediction horizon for the AF, AR, HM and persistence (PS) methods is shown in Figure 21C. The AF, AR and HM methods have similar error levels and each quickly outperforms the simple persistence model. For most of the prediction window, the AR method
shows the lowest error, although the error in the HM decreases near a prediction horizon of 220 years, arguably due to the strength of the de Vries cycle, a 208 year periodicity observed in the power spectrum of the $\Phi$ record, and an important component of the HM model.

### 3.4   Scenario Construction

The REF scenario was calculated as the weighted average of the AF, AR and HM predictions for the current grand solar
maximum, where the prediction errors shown in Figure 21C were used as the weightings. A different approach was used to calculate the EXT scenario. Here, the AF, AR and HM methods were used to generate hindcasts of the 23 prior grand solar





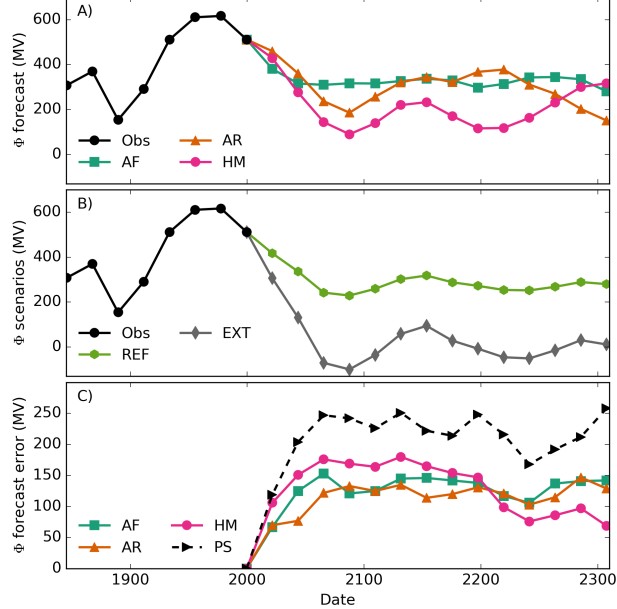

**Figure 21.** A) Observations of $\Phi$ (Obs) from 1850 until 1999, and the three forecasts from 1999 until 2300, from the analogue forecast (AF), auto-regressive model (AR) and harmonic analysis (HM) methods. B) The CMIP6 reference scenario (REF) and extreme scenario (EXT). C) The forecast error for the AF, AR, HM and PS methods, estimated by employing a bootstrap hindcast approach, calculating the root-mean-square of the hindcast errors for 23 prior grand solar maxima in the $\phi$ record.

maxima, also for a 308 year prediction window. The extreme scenario was then calculated as the 5th percentile of the $3 \times 23$ hindcasts at each prediction horizon. The REF and EXT scenarios are shown in Figure 21B.

### 3.5 Future Solar Cycle Definition and Scaling Procedure

Future cycles are constructed from historical cycles by projecting them into the future. The average solar activity level of the projected historical cycles was thereby scaled in accordance to the predicted activity level of the scenarios. Solar activity variations on time scales shorter than a solar cycle are hence preserved. This strategy ensures consistency between the different types of radiative and particle forcing on all time scales also in the future. The historical cycles used for projection into the future are listed in Table 4 of Appendix F.

We assume a linear dependence of the 22-year average sunspot number $<\text{SSN}>_{22}$ on $\Phi$ for the scaling of future solar cycles:

$$< \text{SSN} >_{22} = 0.084\,\Phi + 20.6. \tag{4}$$

The coefficients of Eq. 4 have been obtained from a regression fit, based on SSN and $\Phi$ in the time period 1768–2010 (see Fig. 22). We use international sunspot number version 1.0, because most SSI models rely on that version (see Sec. 2.1.1).





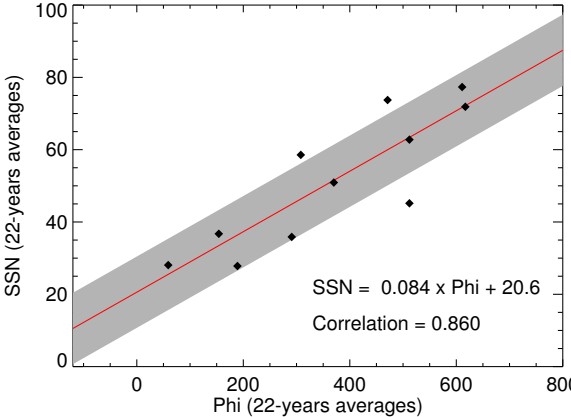

**Figure 22.** Regression of 22-year averaged SSN to the modulation potential $\Phi$. The grey-shaded area represents the $1\sigma$ uncertainty range of the fit. Regression coefficients and the correlation coefficient are also indicated.

The resulting SSN time series of both future scenarios have then be used to calculate the SSI with the SATIRE-TS and NRLSSI2 models with annual time resolution. As for the historical CMIP6 dataset, we took for each scenario the arithmetic mean of the two model results. SSI variations on shorter time scales are taken from the corresponding past solar cycles, and are scaled to a comparable cycle-average level of activity by means of a dedicated scaling procedure, as described in Appendix G.

F10.7 radio flux data has been constructed from the resulting future SSI record as described in Sec. 2.1.2.

A similar approach has also been chosen for the future particle forcing. Magnetospheric particle forcing (Sec. 2.2.1) relies on the geomagnetic indices Ap and Kp, being closely related to sunspot number on decadal time scales (e.g., Cliver et al., 1998). The scaling of these indices in past solar cycles into the future on basis of <SSN> is described in Appendix H. The 2015–2300 Ap time series we obtained have then been used to calculate MEE ionization rates for the REF and EXT scenarios.

Similarly, odd nitrogen upper boundary conditions for the consideration of the EPP indirect effect in climate models with their upper lid in the mesosphere can be computed on basis of the future Ap index with the recommended UBC model (Funke et al., 2016). Future GCR-induced ionization (see Sec. 2.2.3) is calculated from the $\Phi$ of the respective scenarios and interpolated to interannual timescales by using the future SSN time series. The proton forcing of past solar cycles (Sec. 2.2.2) has also been projected into the future, however, no scaling of the proton ionization in dependence of the future cycles' activity level has

been made. This is primarily motivated by the lack of knowledge on long-term variations of proton fluxes, related to the short availability of observational records (since 1962).

## 3.6 Solar Forcing in Future Scenarios

As mentioned before, we provide two scenarios of future solar activity: the reference one is based on the most likely evolution of solar activity from 2015 to 2300, while the extreme one corresponds to the lower 5th percentile of all forecasts. We first

forecast the modulation potential $\Phi$ with the three approaches described in Sec. 3.1.1 to 3.1.3, and define then the reference



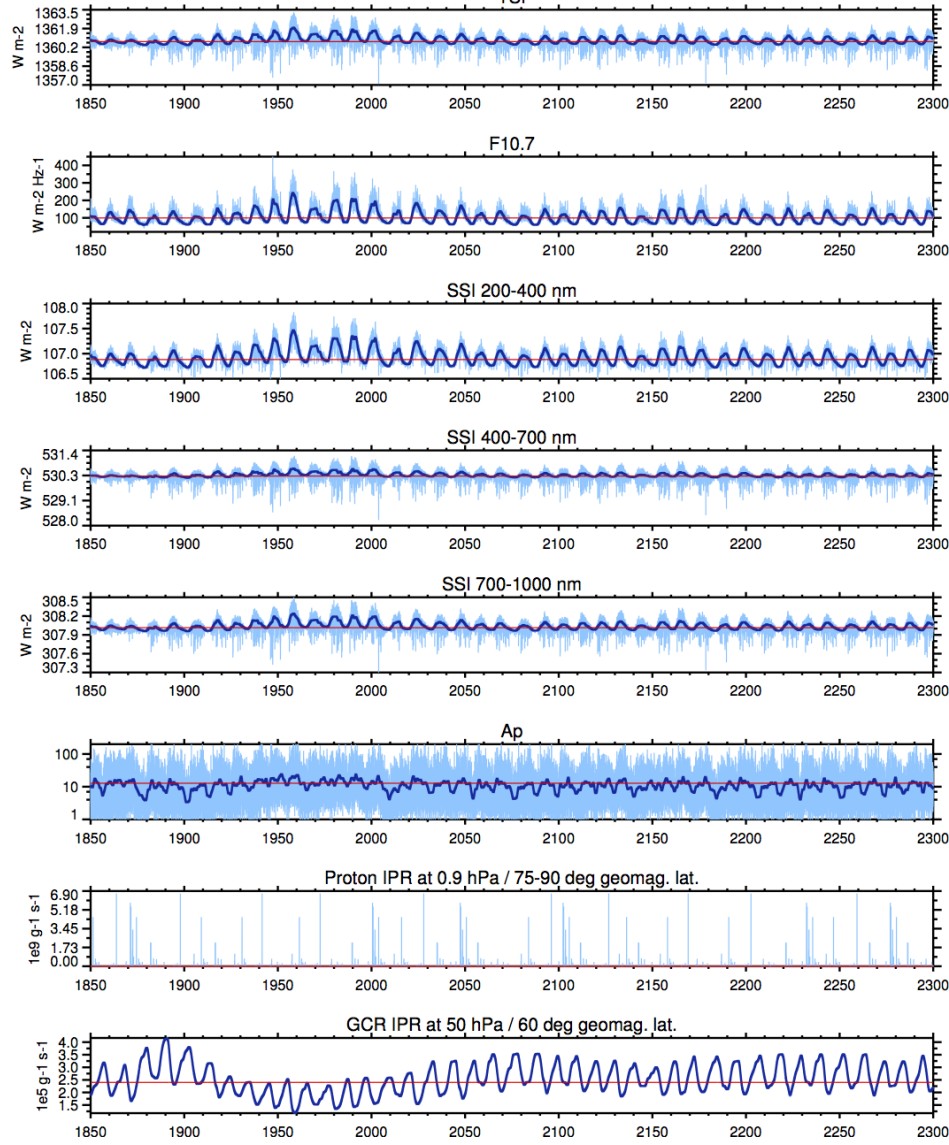

**Figure 23.** CMIP6 Reference (REF) scenario forcing shown for (from top to bottom) TSI, F10.7, SSI at 200–400 nm, SSI at 400–700 nm, SSI at 700-1000 nm, Ap, proton IPR at 1 hPa and 70° geomagnetic latitude, and GCR IPR 50 hPa and 60° geomagnetic latitude. Annually smoothed values are shown by dark blue lines. Constant values of the PI control forcing (see Sec. 4) are shown with red lines as reference.

scenario as their average, weighted by their inverse forecast error (Eq. 2). Note that the analogue forecast, and to a lesser degree, the AR forecast tend to converge toward a climatological mean, whereas the harmonic forecast keeps on oscillating. Because of that, our forecasts are likely to exhibit somewhat less variability than the observed Φ.





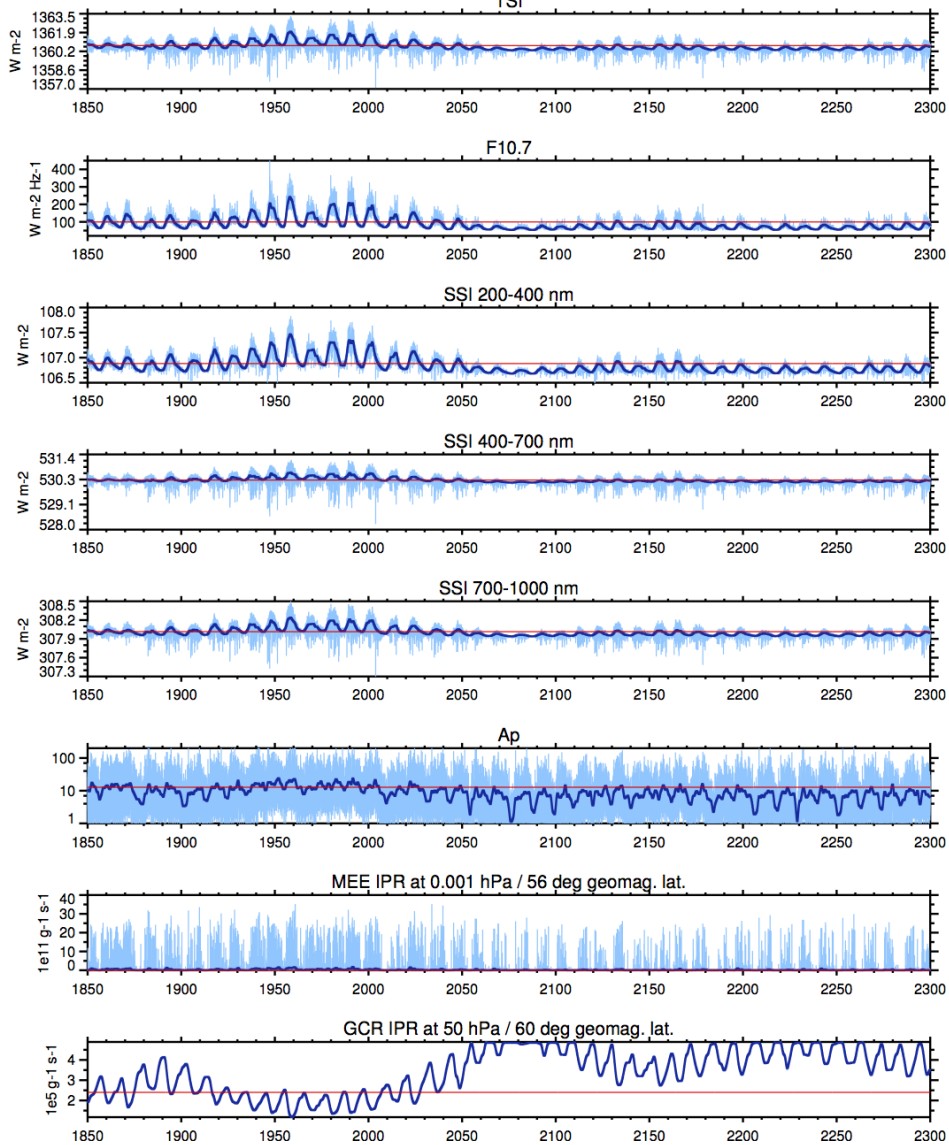

**Figure 24.** CMIP6 Deep minimum (EXT) scenario forcing shown for (from top to bottom) TSI, F10.7, SSI at 200–400 nm, SSI at 400–700 nm, SSI at 700-1000 nm, Ap, MEE IPR at 0.001 hPa and 56° geomagnetic latitude, and GCR IPR 50 hPa and 60° geomagnetic latitude. Annually smoothed values are shown by dark blue lines. Constant values of the PI control forcing (see Sec. 4) are shown with red lines as reference.

Figures 23 and 24 present an overview of the entire daily CMIP6 solar forcing file from 1850 through 2300, respectively for the reference and extreme scenarios. Both show the the TSI, the F10.7cm solar radio flux, which is a good proxy for Lyman-$\alpha$ line, and three different SSI wavelength ranges in the UV, VIS, and NIR. Also shown are the Ap index as a proxy for auroral



electron precipitation, and the ionization rates due to solar protons and galactic cosmic rays. In Fig.24, MEE instead of proton ionization rates are shown, as the latter are identical in both scenarios.

Both scenarios start with a phase of low solar activity, which extends from approximately 2050 to 2110: in the reference scenario, the deepest level is comparable to the Gleissberg minimum that occcured in the late 19th century, whereas in the
extreme scenario, it is considerably deeper, and reaches a Maunder-type minimum. The extreme scenario lingers in that state, whereas the reference one recovers to a climatological mean that is comparable to levels observed during the 1st half of the 20th century. Let us stress that none of the forecasts exhibits a grand solar maximum similar to the one that just ended.

As explained in Appendix F, our scenarios are built out of past solar cycles; therefore, both the solar cycles and their daily variations are consistent with the average level of heliospheric potential. In this sense, the future scenarios for CMIP6 are much
more realistic than the stationary Sun scenario that went into CMIP5.

## 4  Pre-industrial (PI) Control Forcing

For the PI control experiment, we recommend to use one constant (solar cycle averaged) value for the TSI and SSI spectrum representative for 1850 conditions (Fig.25). The average in TSI, SSI, Ap, Kp, F10.7, as well as the ion-pair production rate by GCRs covers the time period from 1.1.1850 to 28.1.1873, which is two full solar cycles. For the ion-pair production rates
by SPEs and MEEs median values representative for the background are provided in order to avoid the occurrence of large sporadic events in the PI control experiment.

As usual the PI control run is supposed to provide an estimate of the unforced climate system to understand internal model variability. It is also used for detection and attribution studies to disentangle contributions from different natural and anthropogenic forcings (some of which include a long-term trend, such as GHGs, aerosols, solar forcing).
For those groups that are interested, we also provide a 1000-year solar forcing time series with 11-year solar cycle variability included but without long-term trend (Fig.25). This time series still has slightly different solar cycle amplitudes and also preserves the variable phase of the solar cycle, however, the solar cycle mean activity level is held constant as compared to the reference scenario in Fig.23. By running a second PI control experiment with solar cycle variability, this provides one additional periodic forcing on top of the seasonal cycle. Since the PI control is also used to determine model variability at decadal time
scales, including a solar cycle would certainly change the mean climate and the variance of the control experiment as compared to the "standard" control experiment with constant 1850 solar forcing. However, not including the solar cycle variability may underestimate the variance of the climate system and may lead to climate system biases. Ideally the groups would do two PI control experiments: one with and one without solar cycle variability.

The variable PI control forcing has been generated by scaling the REF forcing dataset to a constant solar cycle mean activity
level representative for the time period from 1.1.1850 to 28.1.1873. The scaling procedure for SSI and F10.7 is described in Appendix G. The scaling of geomagnetic Ap and Kp indices is described in Appendix H. MEE-induced ion-pair production rates for the variable PI control forcing have then been calculated from the scaled Ap data. GCR-induced ion-pair production rates have been calculated using a constant value of $\Phi$ representative for the 1850–1873 period. Solar cycle variations have

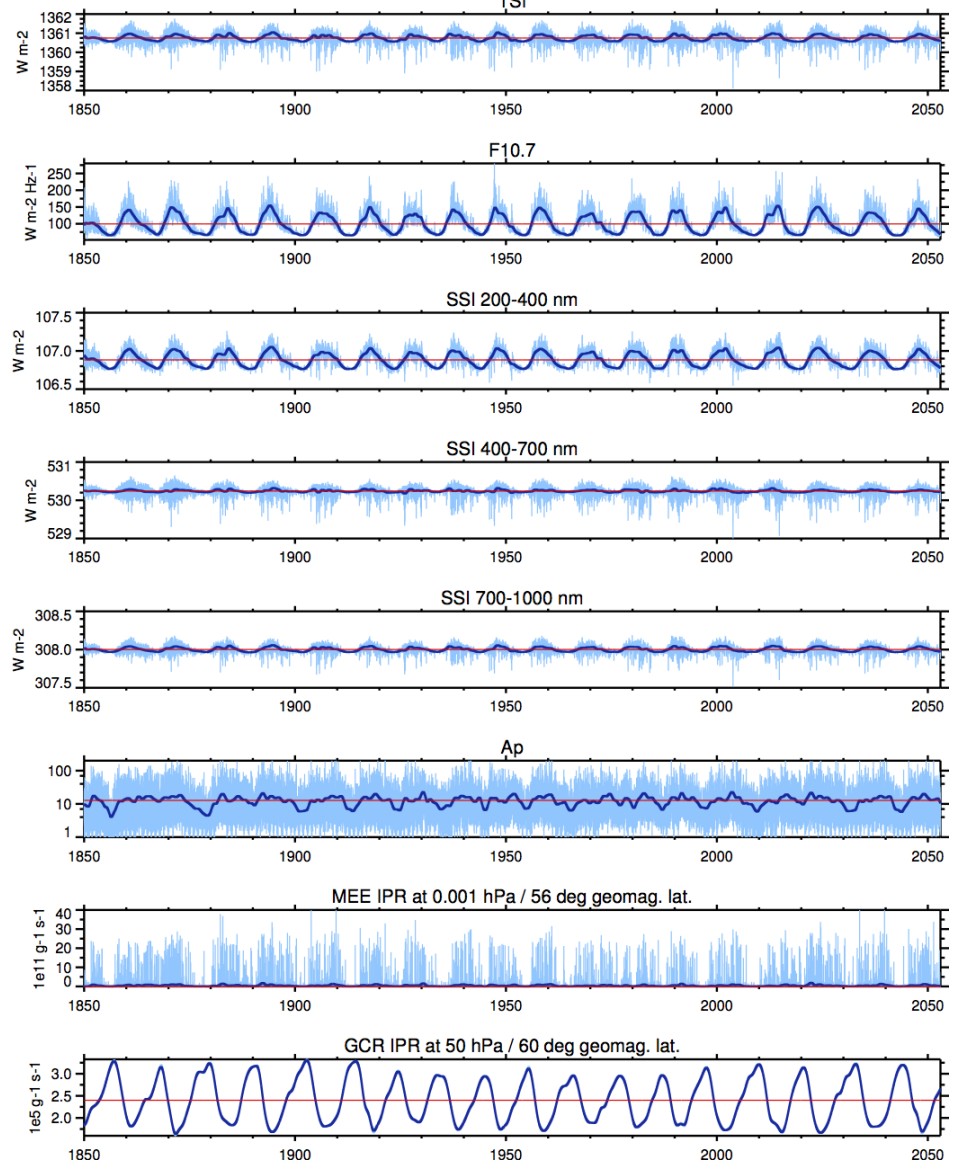

**Figure 25.** CMIP6 Variable (light blue) and constant PI-control (red) forcing shown for (from top to bottom) TSI, F10.7, SSI at 200–400 nm, SSI at 400–700 nm, SSI at 700-1000 nm, Ap, MEE IPR at 0.001 hPa and 56° geomagnetic latitude, and GCR IPR 50 hPa and 60° geomagnetic latitude. Annually smoothed values are shown by dark blue lines.

been added by using the future SSN time series, however, scaled to the 1850–1873 mean activity level. The variable PI control proton forcing is identical to the REF forcing since it does not include any long-term trend. Note that the temporal averages of SSI, TSI, F10.7, Ap, Kp, as well as GCR-induced ion-pair production rates are fully consistent wit the values provided in the



constant PI control forcing dataset. This, however, is not the case for the proton and MEE forcings, which, in the latter case, do not account for large, sporadic events.

The variable PI control dataset (see Fig. 25) covers the time period from 1.1.1850 until 9.9.2053 (end of solar cycle 27). The dataset can be extended to cover 1000 years by multiple repetition of the solar cycle sequence 12–27. The first 450 years of the resulting forcing time series are consistent in solar cycle phase and short-term fluctuations with the REF and EXT datasets. Solar forcing only experiments based on variable PI control, REF, and EXT forcing data would therefore be ideally suited to address the impact of long-term solar activity variations on the climate system.

## 5  Solar Signal in Stratospheric Ozone

The climate response to solar variability depends not only on the 'direct' impact of changes in TSI and SSI on atmospheric and surface heating rates, but also on the 'indirect' effects on stratospheric and mesospheric ozone abundances (e.g. Haigh, 1994). The associated solar-ozone response can contribute more than 50% of the total stratospheric heating response to solar variability in some regions (Shibata and Kodera, 2005; Gray et al., 2009). It is therefore important to include the solar-ozone response in model simulations to realistically capture the impacts of solar variability on climate.

In reality, the 'direct' and 'indirect' parts of the heating are highly coupled, since they reflect the same fundamental process (i.e. absorption of incoming solar photons by molecules). In (chemistry) climate models, the effects of these processes on atmospheric heating rates and temperatures is captured through the radiation scheme as a result of variations in the ozone field and the specified values of TSI and SSI (see Tab. 1). The ozone field in a model can be produced by an interactive photochemical scheme, as presented above for CESM1(WACCM) and EMAC, or it can be externally prescribed in models that do not have a chemistry scheme. Models with a chemistry scheme must adequately represent SSI variability in their photolysis schemes (e.g. in the UV part of the spectrum) to simulate a realistic solar-ozone response. Several CMIP5 models included stratospheric chemical schemes (Hood et al., 2015), and it seems likely that more models will have this in CMIP6. However, there are still likely to be CMIP6 models that do not include chemistry, but which resolve the stratosphere and specify SSI, so have some of the major ingredients for simulating a top-down pathway for solar-climate coupling (Mitchell et al., 2015b). For these models, the simulated climate response to solar variability will partly depend on how the solar-ozone response is represented in their prescribed ozone field.

CMIP5 models without chemistry were recommended to use the SPARC/AC&C Ozone Database (Cionni et al., 2011). The historical part of this dataset for the stratosphere provided monthly and zonal mean ozone concentrations based on a multiple regression analysis of data from the Stratospheric Aerosol and Gas Experiment (SAGE) satellite instruments. The regression coefficients for various key drivers (e.g. ODS, GHG, solar forcing) were used to reconstruct ozone values back to 1850 as a function of latitude and pressure. The historical part of the CMIP5 ozone dataset therefore implicitly included a solar-ozone response derived from satellite observations. However, the SAGE data only cover around two solar cycles and there are significant uncertainties associated with the solar-ozone response in these data that have been recently documented (Maycock



et al., 2016a). It is therefore desirable to update the representation of the solar-ozone response in the upcoming WCRP/SPARC Chemistry Climate Model Initiative (CCMI) Ozone Database being developed for CMIP6 (Hegglin, 2016).

To determine a "best approach" for including the solar-ozone response in the CCMI Ozone Database, Maycock et al. (2016a, b) conducted a detailed analysis of current satellite ozone datasets and CCM simulations. The emphasis of these studies was on
utilising the available data sources to define and solar-ozone response that meets the necessary minimum criteria for the CCMI Ozone Database (Hegglin, 2016). The main properties of the database are that it provides monthly mean ozone mixing ratios on pressure levels covering 1000-0.01 hPa and as a function of latitude for the period 1850-2100.

Maycock et al. (2016a) found that the solar-ozone response in SAGE II mixing ratio data, which has been used extensively for stratospheric ozone studies, shows a strong dependency on the independent temperature record used to convert retrievals from
their native number density on altitude coordinates to mixing ratios on pressure levels. Given the current large uncertainties in the historical evolution of stratospheric temperatures in reanalysis datasets (Mitchell et al., 2015a), Maycock et al. (2016a) concluded that the SAGE II number density data likely provide the most reliable estimate of the solar-ozone response from SAGE data at the current time. However, the relatively sparse spatial and temporal sampling of SAGE means that these data can only provide an annual mean solar-ozone response for the tropics and midlatitudes.

The Solar Backscatter Ultraviolet (SBUV) dataset is another long-term ozone record that has been used extensively for stratospheric ozone studies (e.g Tummon et al., 2015). As a nadir-viewing instrument, the SBUV data provide relatively good spatial coverage, which allows for an assessment of the solar-ozone response on seasonal timescales; however, it possesses much poorer vertical resolution below ∼15 hPa compared to SAGE II. Maycock et al. (2016a) identified some differences between the solar-ozone response in two versions of the recent SBUV VN8.6 data, which must be related to data selection,
calibration and merging procedures. However, the differences in the upper stratospheric solar-ozone response were smaller than between the different versions of SAGE II mixing ratio data. An analysis of the solar-ozone response on monthly timescales in SBUV data suggested substantial sub-annual variations in the magnitude of the solar-ozone response, particularly in the extratropics in the winter hemisphere. These shorter timescale variations in the solar-ozone response may be important for the climate response to solar variability (Hood et al., 2015), but were absent in the CMIP5 Ozone Database; it is therefore desirable
to incorporate them into the CCMI CMIP6 Ozone Database.

Since the core of the CCMI CMIP6 Ozone Database is based on CCMI simulations evaluated against observations (Hegglin, 2016), a similar approach has been adopted for defining the solar-ozone response in this database. Maycock et al. (2016b) examined the solar-ozone response in seven CCMI models. These models provided simulations of the recent past (1960-2009) that include all known external forcing agents (SSI, ODS, GHGs, volcanic aerosols, observed SSTs and sea ice). The
annual mean solar-ozone responses in the CCMs were compared to a subset of observational datasets that were determined by Maycock et al. (2016a) to currently provide the most reliable estimate of the solar-ozone response. Three of the seven CCMs were assessed to show key areas where their responses disagreed with the observations after the uncertainties in the modelled and observed solar-ozone responses were accounted for. The remaining four models were combined to create a CCMI multi-model mean (MMM) monthly and zonal mean fractional solar-ozone response that includes global coverage
from 1000-0.01 hPa. Tropospheric points were masked out using a monthly mean tropopause climatology. Data between the



uppermost CCMI data output level (0.1 hPa) at the uppermost level of the CCMI Ozone Database (0.01 hPa) were filled at each latitude using an exponentially decaying extrapolation with decreasing pressure of the ozone coefficients at 0.1 hPa.

As described above, variations in EPP also affect stratospheric and mesospheric ozone abundances. These effects will be implicitly captured in CCMs with the capability of prescribing EPP and/or their effects on chemical processes (e.g. NOx). The

approach of Maycock et al. (2016b) uses a multiple regression onto the F10.7cm solar flux to extract the solar-ozone response from CCMs. Two of the four CCMI models included in their composite EPP effects (CESM1-WACCM and SOCOL). Thus, although the analysis conducted by Maycock et al. (2016b) did not explicitly account for a solar-ozone response associated with EPP effects, it may implicitly include some component of this if the various indices for EPP (e.g. Ap) are correlated with the F10.7cm flux.

The monthly mean fractional solar-ozone response per 130 units of the F10.7cm solar flux defined by Maycock et al. (2016b) is shown in Figure 26. These show a solar-ozone response of ∼2% in the tropical mid-stratosphere which peaks at ∼5 hPa. The greatest fractional solar-ozone response occurs in the high latitudes, particularly in the winter hemispheres, and in the lowermost stratosphere. Since the solar-ozone response has been defined as a function of the F10.7cm flux alone, a sequence of spatially and temporally evolving ozone anomalies can readily be constructed for all relevant CMIP6 simulations (historical,

future, PI-control) using the time series of CMIP6 recommended F10.7cm solar fluxes described above. These anomalies are being incorporated into the CCMI CMIP6 Ozone Database (Hegglin, 2016). A stratosphere-resolving climate model without chemistry that adopts both the CMIP6 recommended SSI forcing and the CCMI Ozone Database will therefore include a consistent (i.e. in phase) representation of the impact of solar variability on atmospheric heating rates. If a climate model uses the recommended CMIP6 solar forcing dataset, but prescribes an alternative ozone dataset that includes a substantially different

solar-ozone response, it would be expected to show a different solar-climate response compared to other models (Hood et al., 2015). If such cases arise, a minimum requirement is that the temporal evolution of the solar-ozone response should match that of the CMIP6 solar forcing dataset. If this is not the case, then the simulated solar-climate response in a model will not be realistic. It is therefore important to know the methods adopted for implementing stratospheric ozone and the solar-ozone response in CMIP6 models without chemistry.

The ozone anomalies in Figure 26 have been implemented and tested in the Hadley Centre Global Environmental Model 3 (HadGEM3) climate model (Maycock et al., 2016b) using timeslice experiments for solar maximum and minimum conditions similar to those for (CESM1)WACCM and EMAC described above, but using the CMIP5 recommended SSI dataset (Lean, 2000). Equivalent experiments using the solar-ozone response from the CMIP5 ozone database have been conducted for comparison. The net (irradiance + ozone) tropical mean temperature response between 11 year solar maximum and minimum is

0.8 K at 1 hPa using the CMIP6 solar-ozone response from Figure 26. This can be compared to a peak tropical mean solar cycle temperature response of ∼1.2 K for the same model using the CMIP5 SPARC/AC&C Ozone Database. The new recommended solar-ozone response for CMIP6 therefore results in a smaller amplitude solar cycle signal in upper stratospheric temperature by around 30%.





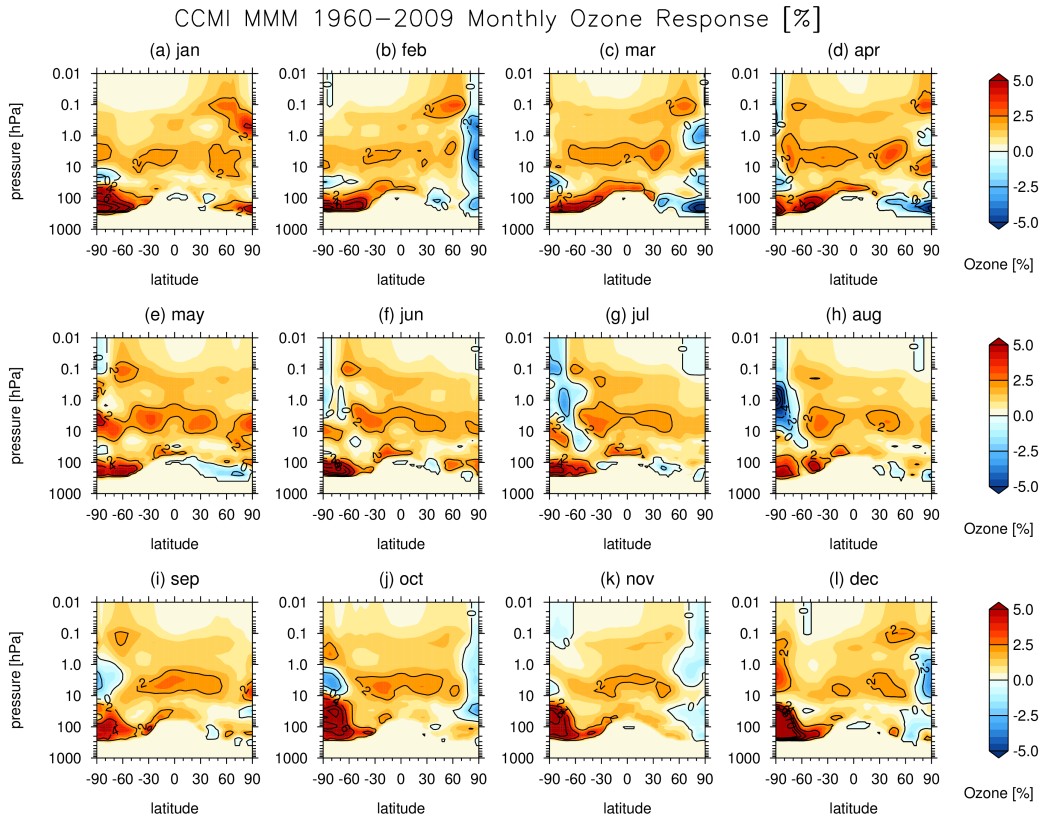

**Figure 26.** The CMIP6 recommended monthly percent (%) solar-ozone coefficients based on 4 CCMI models (CESM1-WACCM, LMDZre-pro, MRI-ESM1, SOCOL3). Figure reproduced from Maycock et al. (2016b).

## 6 Conclusions

This paper provides a comprehensive description of the solar forcing for CMIP6. The dataset consists of time series from 1850 through 2300 of radiative (TSI, SSI, F10.7cm) and particle (Ap, Kp, ionization rates due to SPEs, MEEs, and GCRs) forcings. This is the first time that solar-driven particle forcing has been included as part of the CMIP recommendation and represents a new capability for CMIP6. TSI and SSI time series are defined as averages of two (semi-) empirical solar irradiance models, namely the NRLTSI2/NRLSSI2 and SATIRE-TS. Since this represents a change from the CMIP5 recommended NRLTSI1 and NRLSSI1 dataset, the paper puts special emphasis on the comparison between the radiative properties of the CMIP5 and CMIP6 solar forcing recommendation. Solar forcing is provided in daily as well as monthly resolution separately for the reference period of the historical simulation, i.e. 1850–2014, for the future, i.e. 2015–2300, including an additional extreme Maunder Minimum-like sensitivity scenario, as well as for a constant and a time-varying PI-Control forcing. The particle



forcing is only included in the daily resolution files. The dataset as well as a metadata description and a number of tools to convert and implement the solar forcing data can be found here: http://solarisheppa.geomar.de/cmip6. In the following we highlight the most important points of the CMIP6 solar forcing dataset in comparison to CMIP5 that provide the reader with an overview without reading the paper in detail.

## 5 Radiative Forcing

- A new and lower TSI value is recommended: the contemporary solar cycle-average is now $1361.0 \pm 0.5$ W/m$^2$ (Mamajek et al., 2015).

- During the last three solar cycles in the satellite era there is a slight negative TSI trend in CMIP6 which is statistically indistinguishable from available observations. This trend leads to a radiative forcing on a global scale of -0.04 W/m$^2$ which is small in comparison with other forcings.

- The new CMIP6 SSI dataset is the arithmetic mean of the empirical NRLSSI2 and the semi-emipirical SATIRE-TS irradiance models and covers wavelengths from 10–10,000 nm.

- The CMIP6 SSI dataset agrees very well with available satellite measurements in the contribution of solar cycle variability to TSI in the 120–200 nm wavelength range. In the 200–400 nm range, which is also important for ozone photochemistry, CMIP6 shows a larger solar cycle variability contribution to TSI than CMIP5 (50% as compared to 35%). However, there is a lack of accurate satellite measurements to validate variations in this spectral region. In the VIS part of the spectrum, CMIP6 shows smaller solar cycle variability than CMIP5 (25% as compared to 40%). In the NIR, CMIP6 shows slightly larger variability than CMIP5. The implications of the differences in the spectral characteristics of SSI between CMIP5 and CMIP6 on climatological and solar cycle variability in atmospheric heating rates and ozone photochemistry have been tested using two state-of-the art CCMs, EMAC and CESM1(WACCM), and a line-by-line radiative transfer model, libradtran.

- When comparing the annual mean climatological differences under perpetual solar minimum conditions, the CMIP6-SSI irradiances lead to lower SW heating rates (-0.35 K/day at the stratopause), cooler stratospheric temperatures (-1.5 K in the upper stratosphere), lower ozone abundances in the lower stratosphere (-3%), and higher ozone abundances (+1.5%) in the upper stratosphere and lower mesosphere, as compared to the CMIP5-SSI irradiances. These radiative effects lead to a weakening of the meridional temperature gradient between the tropics and high latitudes and hence to a statistically significant weakening of the stratospheric polar night jet in early winter.

- The changes in solar irradiances between solar cycle maximum and minimum in the CMIP6-SSI dataset result in increases in SW heating rates (+0.2 K/day at the stratopause), temperatures ($\sim$1 K at the stratopause), and ozone (+2.5% in the upper stratosphere) in the tropical upper stratosphere. These direct radiative effects lead to a strengthening of the meridional temperature gradient between the tropics and high latitudes and a statistically significant strengthening of the stratospheric polar night jet in early winter, which propagates poleward and downward during mid-winter and affects





tropospheric weather, with a positive Arctic Oscillation signal in late winter. This regional surface climate response is similar and statistically significant in both CCMs. The CMIP6-SSI irradiances lead to slightly enhanced solar cycle signals in SW heating rates, temperatures and ozone, as compared to the CMIP5-SSI. However, the differences between the solar cycle signals for the two SSI datasets are generally not statistically significant and the solar signal is smaller than the before mentioned climatological differences between CMIP6 and CMIP5.

**Particle Forcing**

– The reconstruction of geomagnetic Ap and Kp indices backwards in time (starting in 1850) allowed to generate a consistent historical dataset of geomagnetic particle forcing for the consideration of the atmospheric impact of precipitating auroral and radiation belt electrons. Regarding the latter, we employed a novel precipitation model for mid-energy electrons (MEE), based on the Ap index. Computed MEE ionisation rates have been successfully tested in the WACCM model. For the consideration of polar winter descent of EPP-generated $NO_x$ in climate models with their upper lid in the mesosphere (i.e., below the EPP source region), recommendations for the implementation of an odd nitrogen upper boundary condition (UBC) are provided. The UBC has been successfully tested with the EMAC model by comparison to observations. Inclusion of the recommended CMIP magnetospheric particle forcing in climate model simulations improves significantly the agreement with observed $NO_x$, $HO_x$, and ozone distributions in the polar stratosphere and mesosphere.

– Solar proton and Galactic Cosmic Ray forcings have been built from well-established datasets which have been used in many atmospheric model studies. Observed proton fluxes, however, are only available since 1963. Before, the proton forcing included in our dataset is fictitious, although resembling the expected overall strength and distribution along the solar cycles.

– In most cases, particle forcing can be expressed in terms of ion pair production rates. We have provided detailed recommendations for their implementation into chemistry schemes.

– CMIP6 model simulations utilizing the recommended particle forcing for the historic (1850–2014) period will allow to address the potential long-term effect of particles as planned to be assessed in upcoming coordinated WCRP/SPARC SOLARIS-HEPPA studies.

**Future Forcing**

In CMIP5, future solar irradiances assumed no long-term changes in the Sun and were obtained by simply repeating solar cycle 23 into the future. In CMIP6, we include a more realistic evolution for future solar forcing based on the weighted average of three statistical forecast models; this shows a moderate decrease to a Gleissberg-type level of solar activity until 2100 for the reference (REF) scenario. We ignore scenarios with high levels of solar activity because the Sun just left such an episode (called grand solar maximum), and several studies suggest that it is very unlikely to return to it in the next 300 years. In addition, we



provide an extreme (EXT) scenario to be used for sensitivity studies, which includes an evolution to an exceptionally low level of solar activity similar to that estimated for the Maunder Minimum.

**PI-Control Forcing**

For the PI control experiment, we recommend to use one constant (solar cycle averaged) value for the TSI and SSI spectrum representative for 1850 conditions. The average values in TSI, SSI, Ap, Kp, F10.7, as well as the ion-pair production rate by GCRs are derived from the time period 1.1.1850 to 28.1.1873, which is two full solar cycles. For the ion-pair production rates by SPEs and MEEs, median values representative of background conditions are provided in order to avoid the occurrence of large sporadic events in the PI control experiment.

We also provide a second PI control forcing time series with a time-varying solar cycle component, but without long-term trend. This time series contains some variation in solar cycle amplitude, and also preserves the variable phase of the solar cycle, however, the mean level of solar activity is held constant at the same values as for the reference PI control experiment. The PI control experiment with solar cycle variability included may reproduce decadal scale climate variability better. Ideally the groups will run two PI control experiments: one with and one without solar cycle variability.

**Solar Ozone Forcing**

For climate models that do not calculate ozone interactively, monthly solar cycle induced ozone anomalies have been calculated from an ensemble of CCMI models and will be used in conjunction with the CMIP6 recommended F10.7cm solar flux to construct a sequence of spatially and temporally varying ozone anomalies that will be incorporated into the CCMI Ozone Database for CMIP6 (Hegglin, 2016). Please note that particle-induced ozone anomalies are not yet explicitly considered in this ozone database. CMIP6 models that include both CMIP6-SSI and solar induced-ozone variations are expected to show a better representation of solar climate variability compared to models that exclude the solar-ozone response.

## 7 Data Availability

The CMIP6 solar forcing dataset described in this paper as well as the metadata description will be published at http://solarisheppa.geomar.de/cmip6 and linked to the Earth System Grid Federation (ESGF) with version control and digital objective identifiers (DOIs) assigned. An overview of the CMIP6 Special Issue can be found in Eyring et al. (2015).

## Appendix A: Recommendations for Model Implementation of SSI

The SSI dataset recommended for CMIP6 covers the solar spectrum from 10 to 100,000 nm. It is provided as irradiance averages for 3890 spectral bins (in W/(m$^2$ nm)). Sampling and equivalently bin width range from 1 nm (UV and VIS) to 50 nm (NIR). Tab. 2 contains more details regarding the resolution changing with wavelength.

**Table 2.** Sampling and bin width of CMIP6-SSI

| spectral range | sampling/bin width |
|---|---|
| 10-750 nm | 1 nm |
| 750-5,000 nm | 5 nm |
| 5,000-10,000 nm | 10 nm |
| 10,000-100,000 nm | 50 nm |

Most climate models prescribe either TSI or SSI in their respective radiation schemes. If the model's radiation code is able to handle spectrally resolved solar irradiance changes, SSI needs be integrated over the specific wavelength bands to generate Top of the Atmosphere (TOA) fluxes. Using CMIP6-SSI this is done for a given spectral band by simply summing up the irradiances of all (partially) contained bins, each multiplied by the bin width (subtracting potential bin parts that reach beyond the boundaries of the target band). A sample routine for the integration can be found here: http://solarisheppa.geomar.de/cmip6. Climate models that calculate ozone interactively also have to integrate SSI to the respective wavelength bands in their photolysis code. An example of the numbers of bands in the radiation and photolysis schemes of two state-of-the-art CCMs, WACCM and EMAC, is shown in Section 2.1.3.

**Appendix B: Recommendations for Geographic Projection of IPR Data**

In order to characterize the electron precipitation into the atmosphere since 1850 we must take into account the offset between geographic and magnetic field coordinates, and how the relationship between them changes with time. We recommend the following approach. For years 1850–1900, the gufm1 model may be used (Jackson et al., 2000). Note that this model would allow calculations earlier in time (1590). From 1900–2015 magnetic field conversions should use the current International Geomagnetic Reference Field (IGRF), which at the time of writing is IGRF-12 (Thébault, 2015). It is highly likely that the magnetic field will continue to evolve in the future, and as such we do not recommend fixing the field in any set configuration based on a specific year. Physics-based simulations are now providing representations of future geomagnetic field changes. For the years 2015–2115, Gauss coefficients based on the predicted evolution of the geodynamo can be used from the modelling of Aubert (2015). For years after 2115, secular variation values from 2115 (also from the Aubert (2015) model) are used to extrapolate forward in time, though obviously with increasing uncertainty. Matlab modelling code implementing the above recommendations is available on the SOLARIS-HEPPA CMIP6 website, which allows users to calculate the geomagnetic latitude for any given date, geographic location and altitude for the period 1590 onwards.

**Appendix C: NOx Production by Particle-Induced Ionization**

Following Porter et al. (1976) it is assumed that 1.25 N atoms are produced per ion pair. This study also further divided the proton impact of N atom production between the ground state N($^4$S) ( 45% or 0.55 per ion pair) and the excited state N($^2$D)



( 55% or 0.7 per ion pair). Ground state [$N(^4S)$] nitrogen atoms can create other $NO_y$ constituents, such as NO, through

$$N(^4S) + O_2 \rightarrow NO + O, \qquad \text{(CR1)}$$

$$N(^4S) + O_2 \rightarrow NO + O_2 \qquad \text{(CR2)}$$

or can lead to $NO_y$ destruction through

$$N(^4S) + NO \rightarrow N_2 + O. \qquad \text{(CR3)}$$

Generally, excited states of atomic nitrogen, such as $N(^2D)$ , result in the production of NO through

$$N(^2D) + O_2 \rightarrow NO + O \qquad \text{(CR4)}$$

(e.g., Rusch et al., 1981; Rees, 1989) and do not cause significant destruction of $NO_y$. If a model does not include the excited state of atomic nitrogen in their computations, the $NO_y$ production from EPP can still be included by assuming that its production is instantaneously converted into NO, resulting in a $N(^4S)$ production of 0.55 per ion pair and a NO production of 0.7 per ion pair.

### Appendix D: HOx Production by Particle-Induced Ionization

The production of $HO_x$ relies on complicated ion chemistry that takes place after the initial formation of ion pairs (Swider and Keneshea, 1973; Frederick, 1976; Solomon et al., 1981; Sinnhuber et al., 2012). Solomon et al. (1981) computed $HO_x$ production rates as a function of altitude and ion pair production rate. Each ion pair typically results in the production of around two $HO_x$ constituents in the stratosphere and lower mesosphere. Sinnhuber et al. (2012) have shown that $HO_x$ is formed as H and OH in nearly equal amounts, with small differences of less than 10% due to different ion reaction chains. In the middle and upper mesosphere, one ion pair is computed to produce less than two $HO_x$ constituents per ion pair because water vapor decreases sharply with altitude there, and is no longer available as a source of $HO_x$. For models which do not include D-region ion chemistry, we recommend to use the parameterisation of Solomon et al. (1981), which is summarized following Jackman et al. (2005b) in Table 3. If the partitioning between $HO_x$ species is considered in the model, H and OH should be formed in equal amounts. Below 40 km altitude and for ionization rates less than $10^2 \mathrm{cm}^{-3}\mathrm{s}^{-1}$ below 70 km altitude, two $HO_x$ can be formed per ion pair. Above 90 km altitude, $HO_x$-production can be set to zero; between 70 and 90 km, values need to be extrapolated for ion pair production rates smaller than $10^2 \mathrm{cm}^{-3}\mathrm{s}^{-1}$ and larger than $10^4 \mathrm{cm}^{-3}\mathrm{s}^{-1}$, taking care not to exceed zero and two.

### Appendix E: Minor Constituent Changes due to Particle-Induced Ionization

If available, the use of more comprehensive parameterizations for productions of individual $HO_x$ (OH and H) and $NO_y$ ($N(^4S)$, $N(^2D)$, NO, $NO_2$, $NO_3$, $N_2O_5$, $HNO_2$, and $HNO_3$) compounds (e.g., Verronen and Lehmann, 2013; Nieder et al., 2014) is



**Table 3.** $HO_x$ production per ion pair as a function of altitude and ion pair production rate (IPR). Table adapted from Jackman et al. (2005b).

| Altitude [km] | $HO_x$ production per ion pair | | (no units) |
| --- | --- | --- | --- |
| | IPR [$cm^{-3}s^{-1}$] | | |
| | $10^2$ | $10^3$ | $10^4$ |
| 40 | 2.00 | 2.00 | 1.99 |
| 45 | 2.00 | 1.99 | 1.99 |
| 50 | 1.99 | 1.99 | 1.98 |
| 55 | 1.99 | 1.98 | 1.97 |
| 60 | 1.98 | 1.97 | 1.94 |
| 65 | 1.98 | 1.94 | 1.87 |
| 70 | 1.94 | 1.87 | 1.77 |
| 75 | 1.84 | 1.73 | 1.60 |
| 80 | 1.40 | 1.20 | 0.95 |
| 85 | 0.15 | 0.10 | 0.00 |
| 90 | 0.00 | 0.00 | 0.00 |

encouraged. Similarly, if atmospheric models include detailed cluster ion chemistry of the lower ionosphere (D region), then the ionization rates should be used to drive the production rates of the primary ions ($N_2^+$, $N^+$, $O_2^+$, $O^+$) and neutrals (N, O) produced in particle impact ionization/dissociation (Sinnhuber et al., 2012). Since such a comprehensive treatment of EPP effects on minor species may introduce more sensitive composition changes via chemical feedbacks, it would be important to
5  document the adopted approaches.





## Appendix F: Projection of Historical Solar Cycles in Future Scenarios

**Table 4.** Historical solar cycles used for construction of future cycles (starting on 2015-01-01).

| Current cycle nb. | Historic cycle nb. | Start current cycle yyyy-mm-dd | Start hist. cycle yyyy-mm-dd |
|---|---|---|---|
| 24 | 12 | 2015-01-01 | 1883-02-01 |
| 25 | 13 | 2020-02-02 | 1890-01-28 |
| 26 | 14 | 2031-12-18 | 1901-12-14 |
| 27 | 15 | 2043-06-19 | 1913-06-15 |
| 28 | 12 | 2053-09-10 | 1878-12-13 |
| 29 | 13 | 2064-10-26 | 1890-01-28 |
| 30 | 14 | 2076-09-10 | 1901-12-14 |
| 31 | 15 | 2088-03-12 | 1913-06-15 |
| 32 | 16 | 2098-06-04 | 1923-09-07 |
| 33 | 17 | 2108-07-05 | 1933-10-07 |
| 34 | 18 | 2118-11-21 | 1944-02-23 |
| 35 | 19 | 2129-01-16 | 1954-04-20 |
| 36 | 20 | 2139-07-02 | 1964-10-03 |
| 37 | 21 | 2150-12-04 | 1976-03-07 |
| 38 | 22 | 2161-04-21 | 1986-07-24 |
| 39 | 23 | 2171-05-21 | 1996-08-22 |
| 40 | 24 | 2183-08-19 | 2008-11-20 |
|  | 12 | 2189-11-07 | 1883-02-01 |
| 41 | 13 | 2194-10-31 | 1890-01-28 |
| 42 | 14 | 2206-09-16 | 1901-12-14 |
| 43 | 15 | 2218-03-18 | 1913-06-15 |
| 44 | 12 | 2228-06-09 | 1878-12-13 |
| 45 | 13 | 2239-07-26 | 1890-01-28 |
| 46 | 14 | 2251-06-10 | 1901-12-14 |
| 47 | 15 | 2262-12-10 | 1913-06-15 |
| 48 | 16 | 2273-03-03 | 1923-09-07 |
| 49 | 17 | 2283-04-03 | 1933-10-07 |
| 50 | 18 | 2293-08-19 | 1944-02-23 |



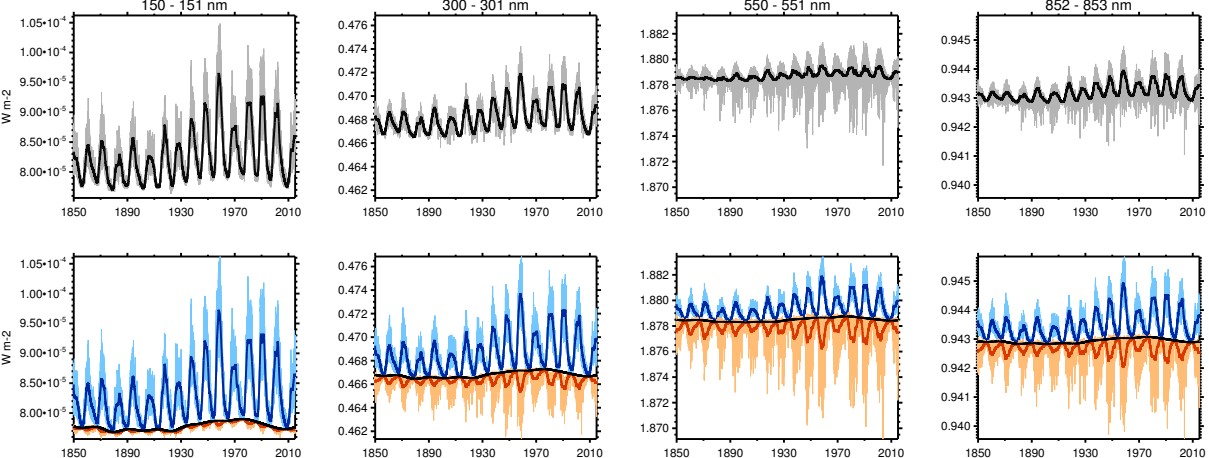

**Figure 27.** Decomposition of SSI in the scaling procedure (shown for wavelength bins centered at 150.5, 300.5, 550.5, and 852.5 nm, from left to right). Upper panels: daily (grey) and annually (black) resolved SSI. Lower panels: Individual components after decomposition: background $SSI^{bg}$ (black solid); facular brightening related $SSI^{+}$ annual, $A^{+}$, (dark-blue) and sub-annual, $D^{+}$, (light-blue); sunspot darkening related $SSI^{-}$ annual, $A^{-}$, (dark-red) and sub-annual, $D^{-}$, (light-red).

## Appendix G:  Scaling of SSI in Future Scenarios and Variable PI-Control Forcing

SSI variability is closely linked to solar magnetic activity variations, and hence sunspot number. However, the form of this relationship may differ significantly at different wavelengths, and time scales (i.e., decadal, annual, sub-annual). Indeed, the contribution to the SSI from different solar features such as faculae, sunspots, the network and ephemeral regions, show

different temporal responses (e.g. Vieira and Solanki, 2010). As a consequence, a simple, wavelength-independent scaling of historic SSI sequences for projection into the future or for the generation of the variable PI control forcing would lead to unrealistic results

    Instead, we first decompose the SSI time series at individual wavelength bins $\lambda$ into components corresponding to the background variability $SSI^{bg}(\lambda)$ (i.e., long-term variations of the SSI at solar minima), to facular brightening-related variability

$SSI^{+}(\lambda)$, and to sunspot darkening related variability $SSI^{-}(\lambda)$. The latter two components are further decomposed into annual ($A^{+}$ and $A^{-}$) and sub-annual ($D^{+}$ and $D^{-}$) contributions (see Fig. 27).

    For the projection of past solar cycles into the future only the $D^{+}$ and $D^{-}$ components need to be scaled since the annually resolved SSI is already provided by the SSI models. These components are shown in Fig. 28 for selected wavelength bins. The distributions of $D^{+}$ values within a given solar cycle are rather symmetric around zero and show a close to normal dis-

tribution. Variability differences between solar cycles are therefore well represented by the corresponding standard deviations $SD(D^{+})_{SC}$ of individual cycles. This quantity also shows good correlation with <SSN> and we therefore use it to construct time-resolved scaling functions. The distributions of $D^{-}$ values are largely skewed towards negative values and do not exhibit the characteristics of a normal distribution. This behavior is expected because of the more intermittent response of SSI to



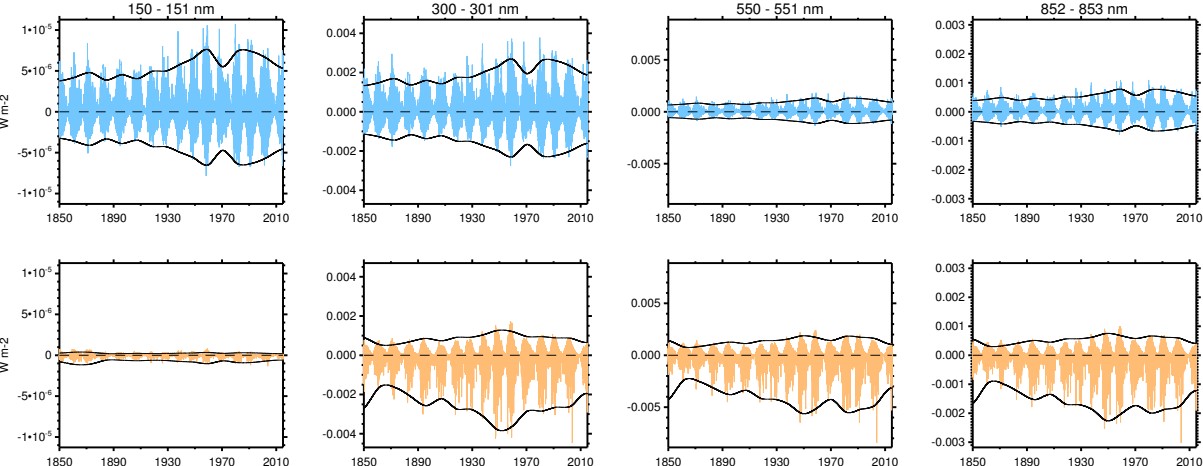

**Figure 28.** $D^+$ (top, light blue) and $D^-$ (bottom, light red) components of SSI at wavelength bins centered at 150.5, 300.5, 550.5, and 852.5 nm (from left to right). The corresponding scaling functions $SD(D^+)$ (top) and $MAD(D^-)$ (bottom), multiplied by 3.5 and -3 in the case of $SD(D^+)$ (5 and -15 in the case of $MAD(D^-)$) are shown by black solid lines.

sunspot darkening, as compared to facular brightening. Variability differences between solar cycles are therefore best represented by the corresponding median absolute deviations $MAD(D^-)_{SC}$, which are therefore used to construct the time-resolved scaling functions for the $D^-$ components.

The coefficients for the scaling of SSI variability with <SSN> have been obtained from linear regression fits of $SD(D^+)_{SC}$

5 and $MAD(D^-)_{SC}$ to <SSN> for each individual wavelength bin (see Fig. 29). In all fits, a nonzero offset is obtained, with particularly large values in the case of $MAD(D^-)_{SC}$. Sub-annual SSI variations are expected to be very low in the absence of sunspots, with variations mainly coming from the solar network (Bolduc et al., 2014). For that reason, we apply a non-linear correction in the scaling for low <SSN> values in order to obtain realistic results for solar cycles with very low activity in the EXT scenario. This has been achieved by multiplying a <SSN>-dependent exponential correction $f$ to the obtained offsets:

$$10 \quad f = \frac{1 - \exp(-0.1<SSN>)}{1 + \exp(-0.1<SSN>)} \tag{G1}$$

For the construction of the variable PI control forcing, all SSI components are scaled individually to 1850–1873 average conditions at each wavelength bin. The scaling has been performed for the $D^+$ and $D^-$ components based on $SD(D^+)$ and $MAD(D^-)$, respectively, as in the the future scenario construction. For $A^+$ and $A^-$, we use the corresponding solar cycle averages to construct time-resolved scaling functions. The background contributions $SSI^{bg}$ were set to constant values corre-

15 sponding to the 1850–1873 averages. The same procedure was also applied to the F10.7 radio flux.



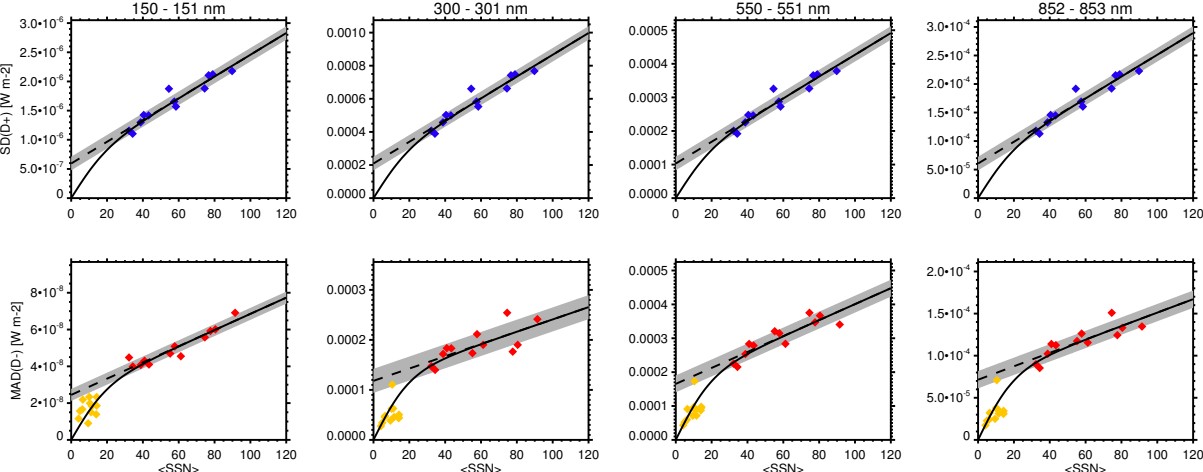

**Figure 29.** Regression of SD(D$^+$)$_{SC}$ (top, blue symbols) and MAD(D$^-$)$_{SC}$ (bottom, red symbols) to <SSN> at wavelength bins centered at 150.5, 300.5, 550.5, and 852.5 nm (from left to right). The resulting linear fit is shown by dashed black lines, the grey shaded areas reflect the RMS errors. The solid black lines show the functional dependence used in the scaling after application of a non-linear correction for low <SSN> values. MAD(D$^-$) values calculated from the low activity part of the solar cycles provide an estimate for very weak solar cycles and are shown with orange symbols (only lower panels).

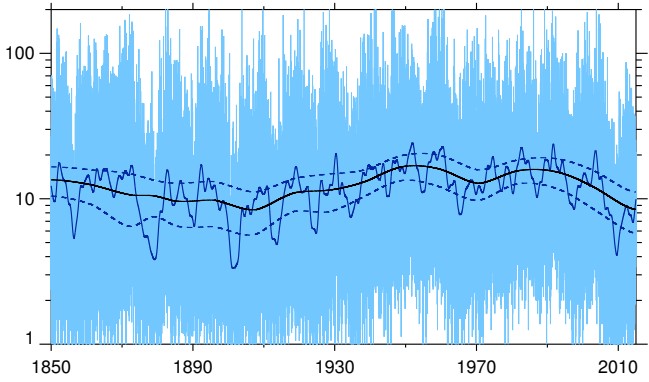

**Figure 30.** Decomposition of the reconstructed Ap index in the scaling procedure. Light blue: daily resolved Ap; black: background component Ap$^{bg}$; solid dark blue: annual component Ap$^a$; dashed dark blue: scaling function SD(Ap$^a$), multiplied by 1 and -1, used to scale the annual component.

## Appendix H: Scaling of Geomagnetic Indices in Future Scenarios and Variable PI-Control Forcing

The geomagnetic activity level is strongly linked to solar activity by the solar wind – magnetosphere interaction. The relationship of geomagnetic activity (and hence geomagnetic indices) and SSN depends strongly on the considered time scales.





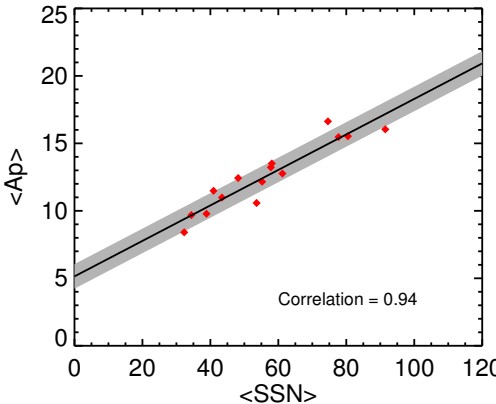 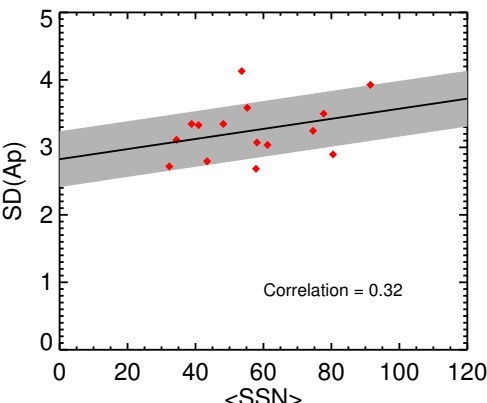

**Figure 31.** Regression of $Ap^{bg}$ (red symbols, left) and $SD(Ap^a)_{SC}$ (red symbols, right) to <SSI>. The resulting linear fit is shown by black lines, the grey shaded areas reflect the RMS errors.

Therefore, we decompose the Ap index in components corresponding to different time scales in order to scale them individually for projection of historic Ap sequences into the future or for the generation of the variable PI control forcing (see Fig. 30). The magnitude of sub-annual Ap variations is large and ruled by the mid-term geomagnetic activity level. Since there is strong evidence for Ap to be described by a multiplicative process (Watkins et al., 2005), we decompose Ap as follows:

$$\text{Ap}(t) = \left( \text{Ap}^{bg}(t) + \text{Ap}^a(t) \right) D(t), \tag{H1}$$

where $Ap^{bg}$ is the background component obtained from the Ap solar cycle averages, $Ap^a$ the annually averaged component, and $D$ a multiplicative daily component, the latter characterized by a nearly constant magnitude of variability on decadal to secular time scales. The $Ap^a$ variability shows only a weak dependence on the long-term geomagnetic activity level. We use its standard deviation from individual solar cycles for construction of a time-dependent scaling function $SD(Ap^a)$.

For the projection of past solar cycles into the future, the components $Ap^{bg}$ and $Ap^a$ need to be scaled in relation to <SSN>. Linear regression fits of $Ap^{bg}$ and $SD(Ap^a)$ to <SSN> are shown in Fig. 31. The correlation of $Ap^{bg}$ and <SSN> is very tight with a correlation coefficient of 0.94. As expected, this is not the case for $SD(Ap^a)$. The pronounced offsets of the regression fits at <SSN>=0 suggest residual geomagnetic activity for very low solar activity (i.e., Maunder Minimum) conditions in agreement with previous studies (e.g., Cliver et al., 1998).

For the construction of the variable PI control forcing, the Ap index is scaled to 1850–1873 average conditions. The scaling has been performed for the $Ap^a$ component on basis of $SD(Ap^a)$. The background contributions $Ap^{bg}$ was set to a constant value corresponding to the 1850–1873 average.

In both future scenario and variable PI control forcing constructions, Ap has been converted into Kp using a statistical correction to account for biases related to the conversion from hourly to daily indices as described in Sec. 2.2.1.





*Author contributions.*   This paper was initiated, coordinated, and edited by K. Matthes and B. Funke. M. Andersson made WACCM simulations for Sec. 2.2 and made Figure 14. L. Barnard contributed text to Sections 2.2 and 3.1, and made Fig.21. M. Clilverd and C. Rodgers led the processing and analysis of the SEM-2 MEPED precipitating electron flux data and wrote part of the text in Sec. 2.2. T. Dudok deWit wrote parts of Sections 1, 2.1.1, 3, and made Figures 1 and 20. B. Funke developed and conducted the extrapolation and scaling of geomagnetic indices, F10.7, and SSI data, made Figures 11, 15, 22–25, 27-31, and wrote parts of Sections 1-4, 6, C-H. M. Haberreiter contributed to writing and interpretations in Sec. 2.1.1. A. Hendry developed the code for geographic-to-geomagnetic conversions with support from C. Rodger and M. Andersson. C. Jackman provided the solar proton IPR dataset and wrote Sec. 2.2.2. M. Kretzschmar contributed to writing and interpretations in Sec. 2.1.1 and made Figures 2-4. T. Kruschke set up and conducted the CESM1(WACCM) simulations for Sec. 2.1.3, made Figs. 5, 7, 9-10, and contributed to the text in Sections 2.1.3 and A. M. Kunze performed the EMAC model runs for Sec. 2.1.3, made Figures 6 and 8, and contributed to the text in Sec. 2.1.3. U. Langematz contributed to the design, analysis and interpretation of the CCM simulations in Sec. 2.1.3 and assisted in editing the manuscript. D. Marsh made WACCM simulations for Sec. 2.2 and provided Fig. 12. K. Matthes wrote the abstract and parts of Sections 1-6, A, and coordinated the design, analysis and interpretation of the CCM and libradtran simulations in Sec. 2.1.3. A. Maycock made Fig. 26 and wrote Sec. 5. S. Misios performed the libradtran simulations and contributed to the text in Sec. 2.1.3. M. Shangguan performed the radiative forcing calculations with libradtran in Sec. 2.1.2. M. Sinnhuber contributed to the analysis of the EMAC tests of the NOy UBC and wrote parts of the text in Sections 1 and 2.2. K. Tourpali contributed to the design and analysis of the libradtran simulations and to the text in Sec. 2.1.3. I. Usoskin provided the GCR ionisation data, wrote Sec. 2.2.3, and made Figures 18 and 19. M. Van de Kamp developed the MEE precipitation model and calculated the spectral parameters with support and coordination from A. Seppälä, P. T. Verronen, M. Clilverd and C. Rodger. P.T. Verronen calculated the MEE ionisation rates, provided Fig. 13, and wrote the MEE text in Sec. 2.2. S. Versick made Figures 16 and 17, did the EMAC model setup, performed the model runs, and contributed to the analysis of the EMAC tests with the NOy UBC. L. Barnard, J. Beer, P. Charbonneau, T. Dudok de Wit, B. Funke, K. Matthes, A. Maycock, I. Usoskin, and A. Scaife designed the future solar activity scenarios and assisted in editing the manuscript. The final solar forcing dataset for CMIP6 available at http://solarisheppa.geomar.de/cmip6 was generated by B. Funke and T. Kruschke.

*Acknowledgements.*   We are very grateful to the SATIRE and NRL teams for providing their respective solar irradiance reconstructions. Without theses input datasets the CMIP6 solar forcing recommendation as well as this paper wouldn't have been possible. We also thank the researchers and engineers of NOAA's Space Environment Center for the provision of the data and the operation of the SEM-2 instrument carried onboard the POES spacecraft. This work has been conducted in the frame of the WCRP/SPARC SOLARIS-HEPPA activity, and an early part inside the EU COST Action ES1005 (TOSCA) which some authors were involved in. This work contributes to the ROSMIC activity within the SCOSTEP VarSITI programme. Part of the work described in this article emanates from two teams at the International Space Science Institute (ISSI) in Bern, i.e. *Quantifying Hemispheric Differences in Particle Forcing Effects on Stratospheric Ozone* (Leader: D. R. Marsh) and *Scenarios of Future Solar Activity for Climate Modelling* (Leader: T. Dudok de Wit), which we gratefully acknowledge for hospitality. MK, TD, MH, KT and SM acknowledge that the research leading to the results has received funding from the European Community's Seventh Framework Programme (FP7 2012) under grant agreement no 313188 (SOLID, http://projects.pmodwrc.ch/solid). PTV, MEA, AS, and MvdK were supported by the Academy of Finland projects #276926 (SECTIC: Sun-Earth Connection Through Ion Chemistry) and #258165/#265005 (CLASP: Climate and Solar Particle Forcing). KM, TK, MK, UL, SV and MS gratefully acknowledge funding by the German Ministry of Research (BMBF) within the nationally funded project ROMIC-SOLIC (grant number 01LG1219). KM also acknowledges support from the Helmholtz-University Young Investigators Group NATHAN funded by the Helmholtz-Association through the President's Initiative and Networking Fund and the GEOMAR Helmholtz Centre for Ocean Research Kiel. MS gratefully acknowledges funding by



the Helmholtz Association of German Research Centres (HGF), grant 608 VH-NG-624. BF was supported by the Spanish MCINN under grant ESP2014-54362-P and EC FEDER funds. LB thanks the Science and Technology Facilities Council (STFC) for support under grant ST/M000885/1. AS was supported by the joint DECC/Defra Met Office Hadley Centre Climate Programme (GA01101) and the EU SPECS project (GA308378). UL and MK thank the North-German Supercomputing Alliance (HLRN) for support and computer time. KM and TK

5    thank the computing centre at Kiel University for support and computer time.



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
