# Peer review of "Solar Forcing for CMIP6 (v3.2)"

_Geoscientific Model Development, 2016_

## Short Comment (SC1) · 21 Jun 2016

This is an important contribution which provides crucial datasets for future modelling. Nevertheless, the manuscript does not address a key issue which needs to be resolved before a new generation of climate models commences.

According to the IPCC AR5 report, chapter 5.3.5. of the Physical Science Basis, climate models still struggle to reproduce key climatic events such as the warm phase of the Medieval Climate Anomaly. This major deficiency and challenge has been recently confirmed by e.g. Wilson et al. 2016 and Luterbacher et al. 2016. http://www.sciencedirect.com/science/article/pii/S0277379115301888?np=y http://iopscience.iop.org/article/10.1088/1748-9326/11/2/024001

The simulations are essentially running too cold and do not reach the high temprature levels during the Medieval Climate Anomaly which are reconstructed based on palaeoproxies. This may indicate that the radiative forcing assigned to solar activity changes in the climate models may actually be too low and needs a upward adjustment. It is clear that climate models and the value for solar radiative forcing need to first proof themselves in the hindcast before qualifying for future climate predictions.

The current manuscript, unfortunatey, fails to address this important issue by starting key datasets only at 1850, i.e. the end of the Little Ice Age. In order to compare apples to apples it is necessary, however, to compare the current warm phase with the previous warm phase, i.e. the Medieval Climate Anomaly / Medieval Warm Period.

On page 48, lines 20/21 the authors write:

"For those groups that are interested, we also provide a 1000-year solar forcing time series with 11-year solar cycle variability included but without long-term trend".

Notably, this long-term trend is effectively the key to the problem, therefore it is counter-prooductive that this trend is cut out. Millennial-scale solar cycles of Eddy and Hallstatt nature deserve much greater importance, especially as Holocene climate fluctuations have been documented (e.g. Bond cycles) to operate on similar time scales.

I encourage the authors to openly address the hindcast and solar radiative forcing challenge. Long-term trends in solar activity need to be added for the past 1000-2000 years, in order to enable modellers to tackle the problem.

---

## Referee Comment (RC1) · M. Snow (Referee) · 6 Jul 2016

This manuscript gives a very understandable overview of how the recommended solar forcing is determined. The topic is very technical, but the authors have done a great job in making the content accessible to a wide range of scientists. My comments are primarily on the radiative sections, and all of them are minor.

P3 L29: The text mentions uncertainties in the irradiance measurements, and there is no question that these uncertainties can be sometimes larger than solar variability at some wavelengths. There is a brief discussion that uncertainties in models are sometimes difficult to assess. Measurements only go back a few solar cycles, and both irradiance models are based on interpretation of those measurements. Extrapolating these proxy models to the past and future requires several assumptions about proxy relationships remaining invariant. For example, the numerical relationship between sunspot number (or area) and the sunspot blocking function in NRLSSI2 could

change if the sunspot contrast evolves over time. So it is not automatically true that we understand proxy relationships well enough over long timescales. Model development will continue to improve as we continue to make better measurements. State of the art model reconstructions as described in this manuscript are the best we currently have, but their uncertainties are also still significant.

P9 L14: Averaging two quantities that disagree produces a result that is also not likely to be correct. Calling this "the most reasonable approach" is perhaps controversial. Maybe calling it "a reasonable approach" would be more appropriate.

P9 L30: The comment that F10.7 was a good proxy for EUV at one time, but "this may not be true anymore" reinforces my discussion about page 3 above. It is an assumption that proxy relationships do not change over time, and this assumption must factor into the estimation of model uncertainties.

Overall, I think this manuscript does an excellent job in describing the recommended solar forcing for the climate community.

---

## Author Comment (AC1) · 7 Jul 2016

Dear Dr. Luening,

We appreciate your interest for our manuscript. Since the CMIP6 modeling exercise starts in 1850, the time series for the solar forcing is provided exactly for this period. Our intention is to provide a best estimate solar forcing data set that will be used by all CMIP6 modeling groups in order to test how well the models can reproduce observations.

We agree that it is important to also look further into the past. This is actually done within the PMIP (Paleoclimate Modeling Intercomparison Project) Exercise (Kagaeyama et al., GMDD 2016). PMIP has an experiment in Tier 1, which means highest priority, covering the last 1000 years and therefore also the Medieval Warm Period and the Little Ice Age. The solar forcing provided for PMIP will be coordinated with our

solar forcing, in order to ensure consistency. In addition, as part of the PMIP "Tier 2" experiments, a variety of solar forcing reconstruction will be provided covering uncertainty in base data (14C and 10Be isotope records), as well as different assumptions for long-term changes, to reflect uncertainties in the magnitude of secular variations (Jungclaus, et al., manuscript in preparation for Clim. Past).

Our 1000-year solar forcing time series is meant solely for sensitivity experiments in order to understand physical mechanisms for internal natural climate variability such as a potential synchronization of North Atlantic climate variability by the 11-year solar cycle (Thieblemont et al., 2015) in the atmosphere-ocean system. It avoids on purpose any long-term trend in solar activity, and should therefore not be used for historical model simulations and/or solar forcing reconstructions.

We will include a comment and a link to the PMIP publication (Kageyama et al., 2016) in our revised manuscript.

Best regards,

Katja Matthes and Bernd Funke

Kageyama, M., Braconnot, P., Harrison, S. P., Haywood, A. M., Jungclaus, J., Otto-Bliesner, B. L., Peterschmitt, J.-Y., Abe-Ouchi, A., Albani, S., Bartlein, P. J., Brierley, C., Crucifix, M., Dolan, A., Fernandez-Donado, L., Fischer, H., Hopcroft, P. O., Ivanovic, R. F., Lambert, F., Lunt, D. J., Mahowald, N. M., Peltier, W. R., Phipps, S. J., Roche, D. M., Schmidt, G. A., Tarasov, L., Valdes, P. J., Zhang, Q., and Zhou, T.: PMIP4-CMIP6: the contribution of the Paleoclimate Modelling Intercomparison Project to CMIP6, Geosci. Model Dev. Discuss., doi:10.5194/gmd-2016-106, in review, 2016.

Thiéblemont, R., Matthes, K., Omrani, N.-E., Kodera, K., and Hansen, F.: Solar forcing synchronizes decadal North Atlantic climate variability, Nature Commun., 6, 8268, doi:10.1038/ncomms9268, 2015.

---

## Referee Comment (RC2) · Anonymous Referee #2 · 13 Jul 2016

This is an important paper as it is meant to be a standard reference for the climate models taking part in the CMIP6 exercise. It is rather comprehensive, covering many topics. Some parts are well written, easy to understand and, as far as I can see, are internally consistent. However, other parts are inconsistent or unclear. Below I list the points where the paper could be improved. Many of these are minor issues such as typos etc. that should be easy to take care of, but there are a few points that are more fundamental and which the authors must deal with carefully. They might require bigger changes to the manuscript.

1) page 1, line 10 (abstract): "The TSI and SSI time series are defined as averages of two (semi-) empirical solar irradiance models, namely the NRLTSI2/NRLSSI2 and SATIRE-TS." Two comments: a) NRLTSI2/NRLSSI2 are empirical, not semi-empirical models. This is correctly explained later in the text of the manuscript but the statement in the abstract is misleading. b) More importantly, could the authors describe why they

have taken recourse to this unusual step of averaging two independent models? Both models are constructed differently, and both have to some extent been tested against data. What the authors do is to provide a new model that is largely untested with regard to solar data. More comments on this aspect will be made later.

2) page 2, lines 1-2: "The slight negative trend in TSI during the last three solar cycles in CMIP6 is statistically indistinguishable from available observations". What do the authors mean by this? Different TSI composites indicate different trends. There are implicit assumptions underlying this statement that need to be spelt out and the reasoning clarified.

3) page 2, line 5: CMIP6 cannot be tested against CMIP 5. It can only be compared to CMIP5.

4) page 2, line 6: The expression "background SSI" is neither clear nor used in the literature.

5) page3, line 5: "Because of its prominent 11-year cycle, solar variability may offer a degree of predictability for regional climate and could therefore help reduce uncertainties in decadal climate predictions." However, the solar cycle itself is notoriously hard to predict and predictions of upcoming solar minima have not been particularly successful, so that the statement does seem too optimistic. But possibly the authors are not concerned with such niceties here?

6) page 3, lines 23-24: "The quantitative assessment of radiative solar forcing has been systematically hampered so far by the large uncertainties and the instrumental artifacts that plague TSI and SSI observations" The TSI observations are significantly more precise than those of SSI (especially SSI in the near UV and visible spectral domains). This makes the quoted sentence misleading.

7) page 3, line 28: The IAU resolution (Mamajek et al. 2015) adopted the result of Kopp & Lean (2011). The latter is a well-known and well-cited paper in a refereed journal,

while the former is not a scientific paper at all, is not properly published, and the text in the reference is not really useful. Please replace the reference to IAU resolution with the Kopp & Lean paper everywhere in the text.

8) page 3, line 29: The proxy reconstructions of the TSI do not exhibit "occasional phases of unusually low nor high activity", the Sun does. Also, Usoskin et al. 2014 did not mention TSI reconstructions at all.

9) page 3, line 32: "There is growing evidence for the Sun to enter a phase of low activity near 2050, after a grand maximum that peaked during the 20th century." The authors should explain what they base this claim on. The solar dynamo is chaotic, possibly even stochastic and it is even difficult to predict the next solar cycle much before the previous minimum. Going beyond the next cycle is even tougher. See e.g. * J. Jiang, R.H. Cameron, M. Schüssler, 2015: The cause of the weak solar cycle 24; ApJL 808 L28 * Cameron et al. 2013: Limits to solar cycle predictability: Cross-equatorial flux plumes, A&A 557, A141

10) page 4, line 12: " . . . with respect to the CMIP5 solar forcing recommendation." Is the CMIP5 paper supplying the solar input cited? At least so far "CMIP5" is not associated with such a paper.

10a) page 4, line 20: "Lockwood et al., 2019" This is likely "Lockwood et al., 2010"; also to be corrected on page 74

11) page 6, line 15: " . . . and one observational estimate (SOLID), . . ." Is this correct? From the evidence given in the paper, SOLID is not an "observational estimate" but an empirical model (see below).

12) page 6, lines 26-27: The sentence simplifies things too much. Faculae are not "the Mg II index" and dark sunspots are not "sunspot area". These proxies are used to represent the contributions from faculae and sunspots to the irradiance. This should be written more carefully to reflect the actual relationships.

13) page 7, line 8: ". . . .SORCE SSI observations Lean and DeLand (2012) the wavelength dependent scaling coefficients . . ." Change to: ". . . .SORCE SSI observations (Lean and DeLand, 2012) the wavelength dependent scaling coefficients . . ."

14) page 7, lines 5-15: The description of the adjustments in the NRLSSI2 model is not clear. The authors should explain better how the adjustments are done.

15) page 7, line 19: What is SATIRE-TS? It does not become clear from reading the paper, as references are given only to SATIRE-T and SATIRE-S. TS is used multiple times, so that it cannot be a typo. Is TS some combination of the two models? What kind of combination?

16) page 7, line 28: The reconstruction made by Dasi-Espuig et al. (2014) is not based on sunspot number.

17) page 8, line 3: "In NRLSSI2, this internal consistency also applies to the integral of the facular and sunspot contributions to SSI to their respective counterparts in TSI (see Coddington et al. (2015) for more details). Why are the authors stressing this point? This sounds rather trivial. Doesn't every model that distinguishes between spots and faculae do that? Or are they implying that SATIRE does not fulfil such internal consistency? They should either explain why this is such an achievement and is unique to NRLSSI, or they should remove this unnecessary sentence.

18) page 8, lines 5-6: "The controversial out-of-phase behavior of SORCE/SIM observations in that band (Harder et al., 2009) are likely to be an instrumental artefact (Lean and DeLand, 2012; Ermolli et al., 2013) but this has not yet been corrected in the SOLID composite." If SORCE is wrong, is then the SOLID composite the best one to use to test the SSI models? If models are chosen by how close they are to SOLID, this could have unwanted effects for the atmospheric chemistry and finally climate modelling. As shown by Haigh et al. (2010, An influence of solar spectral variations on radiative forcing of climate, Nature, 467, 696), the SSI variability found by SORCE has a major effect on atmospheric ozone, changing its concentration in a way contrary to

expectations.

19) page 8, line 14 "In NRLSSI2, the proxy index for sunspot darkening is the sunspot area as recorded by ground-based observatories in white light images since 1882 (Lean et al., 1998). Values prior to that are estimated from the sunspot number." Does not also the SATIRE model rely on sunspot areas over the period of time covered by Greenwich observatory? It seems to according to * Krivova et al. 2010: Reconstruction of solar spectral irradiance since the Maunder minimum; JGRA, DOI: 10.1029/2010JA015431 Please comment.

20) Sect. SOLID composite After reading this section the exact nature of the SOLID composite is still very unclear. A single paper (Schöll et al., 2016) is cited, which explains only the preprocessing of the data. As long as SOLID is not properly documented, it is important to make the description here sufficiently clear so that readers who don't know the nitty gritty of the SOLID approach can follow and, if necessary make their own judgement. So far, this composite (if it is really a composite at all and not an empirical model, see below) is not widely-known or established in the solar community.

21) page 9, line14: "..., the most reasonable approach consists in averaging both reconstructions, weighted by their uncertainty. But this means that yet another model is produced, one that is untested, except for the rudimentary tests briefly discussed in the paper. It is not clear why this is "the most reasonable approach"

22) page 9, line 17: "... SATIRE-TS (Yeo et al., 2014)." What is SATIRE-TS? Is it a typo or a combination of SATIRE-T and SATIRE-S? Please define it. In Yeo et al. (2014) I could find a description of SATIRE-S, but no mention of SATIRE-TS, so that this reference seems not to be relevant (unless TS is a combination of T and S).

23) page 9, line 24: "The EUV band (10-121 nm) is required for CMIP6 but is absent from NRLSSI and SATIRE. We added it with spectral bins from 10.5-114.5 nm by using a nonlinear regression from the SSI in the 115.5-188.5 nm band, trained with

TIMED/SEE data from 2002 to 2009." Indeed, this is a difficult situation and the authors have done the reasonable thing and somehow modelled this wavelength region themselves. The procedure chosen may be fine and may produce a good result, but without being shown anything it is hard for the reader to judge. Please provide a figure and some more explanation.

24) page 9, line 29: "Let us note that while the F10.7 index is a good proxy for EUV variability on daily to yearly time scales, this may not be true anymore on multi-decadal time scales." This is a good point.

25) page 10, line 3: "... observed composite from PMOD" This sounds strange since the composite was not observed. Rather it is based on a set of observations of TSI.

26) page 10, line 7: "All TSI records agree well on daily to yearly time scales, and in some cases (e.g. NRLTSI1 and NRLTSI2) they match as well on multi-decadal time scales." Aren't both models part of the same model family and are founded on the same proxies? I am not sure what is remarkable about them being consistent with each other? The authors can leave this – it is not an important point – but I would like to understand why they stress this agreement, which I would have naively thought to be trivial.

27) page 11, Figure 1: The orange curve described as CMIP5 goes till 2015. What exactly is plotted? Are these extrapolated CMIP5 data? Or are the plotted CMIP5 data regularly updated and are available up to 2015?

28) page 11, line 12: "The major difference come from SATIRE-TS .." "The major difference comes from SATIRE-TS .."

29) page 12, Figure 2, bottom left diagram Why is NRLSSI2 producing such strange cycles between ∼1940 and 1960? These certainly do not look realistic. How come, these strange cycle shapes are restricted mainly to the visible (although there may be some sign of similar behavior in the IR)? Should not also the UV cycles behave strangely

in order to compensate for the behavior in the visible and produce a reasonable TSI? Has the TSI produced by NRLSSI2 been compared with measured time series? Also, something seems to be going wrong right at the start of the time series. How do the authors explain and compensate for these problems?

30) page 12, line 4: " . . . and higher-quality data from the SORCE mission on the rotational timescale in NRLSSI2, . . ." But isn't SORCE giving wrong trends for SSI? This is what the authors claim multiply elsewhere in the paper.

31) page 13, Figure 3: Fig. 3 is confusing and does not agree with the text of the paper, mainly regarding SOLID. Earlier it was said that SOLID is based entirely on observed data (page 8, line 20: "More specifically it is derived as the weighted mean of all available SSI observations in the satellite era.") If that were true, then the green curve should be a lot closer to the plotted observations. Thus, it is not clear why The SOLID composite departs so strongly from SORCE at a time when that is the only data set used (according to the authors)? Or are other data sets also used after all? The SSI variations shown by SOLID seem to be smaller than of all the instrumental records, except maybe UARS SOLSTICE (it is hard to see - there are too many light colors in this plot). Anyway, SOLSTICE shows a behavior completely inconsistent with an SSI composite of the observations. This is a serious problem that points to a fundamental inconsistency in the paper. Another strange feature of this plot is that SOLID covers also the 1950s and 1960s when there were no SSI data available. How does SOLID produce something at those times if it is purely based on SSI data? The description given in the paper is totally inadequate and obviously seriously misleading. However, Fig. 3 very strongly suggests that the SOLID "composite" uses either a proxy (possibly something like 10.7 cm flux?) or makes really strong changes to the data while processing them. In either case, I would strongly oppose calling it a composite of SSI observations. Rather Fig. 3 clearly shows that it is an empirical model. If the authors want to maintain that SOLID is a composite of observed SSI, then they should provide a detailed explanation that goes far beyond the inadequate one in the current version

of the paper. This should include a list of all data sets that enter into the SOLID "composite" and all the steps that are undertaken to produce it. Also, they should provide a convincing explanation why SOLID differs so strongly from the observational data. Fig. 3 raises an issue regarding fairness and bias in the paper. If SOLID is indeed an empirical model, and I have seen no evidence to counter this in the paper, I see no advantage in using SOLID to "test" the other two models. Indeed, if SOLID is a model (and an unpublished one at that), why is it being discussed ahead of the numerous other (published!) models in the literature. I see only two paths that that the authors can follow: a) Either remove SOLID completely from the publication and instead compare the averaged model that the authors have produced more rigorously with the observations directly, b) or discuss SOLID on an equal footing with the other SSI models that the authors simply ignore in this version of the paper. Irrespective of which of these paths the authors follow, I strongly urge them to use the original SSI observations to test the new model data set obtained by averaging NRLSSI2 and SATIRE (-TS?).

32) page 13, Figure 4 and its discussion in the text: Averaging over one month at activity maximum and minimum does not allow eliminating the rotational cycle in solar variability, so that this figure mixes information on shorter timescales into the solar cycle variability that the authors want to show. The figure should be redone using at least 81-day averaging. Why are the comparisons in Figures 3 and 4 being done in such broad, seemingly arbitrary wavelength bands, rather than broken up according to the important molecular band listed in Table 1? What is the advantage for the climate community of following the bands used by Ermolli et al. (2013). Also, where would the observations lie in Fig. 4 (to the extent available for exactly these times, which is a limitation of the figure)?

33) page 13, lines 1-2: " . . . the only available measurements are from the SORCE/SIM instrument, which has calibration issues (Lean and DeLand, 2012) . . ." Until now no calibration issues in SORCE/SIM instrument have been reported by the instrumental

team. In particular, the paper by Lean & DeLand does not identify any calibration issue.

34) page 14, line 5: "In the NIR CMIP6 shows slightly larger variability than CMIP5 and remarkable here is the largest variability in NRLSSI2." Why is this remarkable? Both SATIRE & NRLSSI2 reproduce TSI. NRLSSI2 has a smaller variability in the UV and this must be compensated by NRLSSI2 in the IR. Or is there something more complex at work here that I am missing? Please explain or simplify the text.

35) page 17, line 1: "Solar activity and hence spectral irradiance vary between different solar cycles. However, these differences are small compared to the total 11 year solar cycle amplitude . . ." This has been the case in the second half of the 20th century, but the sizes of cycles vary between zero and the very high amplitude of cycle 19. This is only a minor quibble, however.

36) page 19, line 18: ". . . produces slightly higher SW heating rates than NRLSSI1(CMIP5)" ". . . produces slightly higher SW heating rate differences than NRLSSI1(CMIP5)"; the diagram shows the differences between heating rates for solar minimum and solar maximum.

37) page 20, Figure 5: "Impact of solar forcing according . . ." add: for perpetual solar minimum conditions

38) page 21, Figure 6: "CMIP6 SSI differences in % for perpetual solar minimum conditions . . ." It may not be immediately clear what differences are actually meant here, i.e. differences in which parameter; add e.g.: "CMIP6 SSI differences of the solar irradiance in % . . ."

39) page 21, line 1: "More important for the solar ozone signals seems to be the choice of the CCM (with its specific photolysis scheme, see also Fig. 8), especially for the lower stratosphere (10 hPa and below)." This is true for the lower stratosphere only; above that the dataset-induced differences are larger than the model-induced ones, in particular in the lower Mesosphere.

40) page 21, line 12: "Note that statistically significant irradiance differences between CMIP5 and CMIP6-SSI irradiances are particularly observed between 300 and 350nm . . . (Fig. 8)." In Figure 8 differences in the irradiance amplitude between solar minimum and solar maximum are shown. i.e. "irradiance differences" should be replaced by "differences in the irradiance amplitude".

41) page 25, line 19: ". . .for mesospheric OH production Fytterer et al. (2015b), and for . . . ". . . for mesospheric OH production (Fytterer et al., 2015b), and for . . ."

42) page 27, Figure 11: SSN scaled by a factor of 0.67 should have larger values (in Figure 13 SSN scaled by a factor of 0.741 has values above 200); a factor of 0.067 appears to be much more reasonable.

43) page 33, Figure 16: There are differences between the caption and the labels in the diagram. caption: 70–90oS (left) and 70–90oN (right); in the diagram: 70–90oS (right); 70–90oN (left) caption: 0.01 hPa (upper panel) and 0.1 hPa (lower panel); in the diagram: 0.1 hPa (upper panel) and 1 hPa (lower panel)

44) page 36, line 13: "Since fast transient solar energetic particle events often occur at the background of enhanced geomagnetic disturbances, straight-forward computation of the particle trajectories in a realistic geomagnetic field is needed" Why are fast transient solar energetic particle events relevant in this context? This paragraph deals with the penetration of GCRs in the Earth's atmosphere.

45) page 38, line 25: The CMIP6 future solar forcing is different from that of CMIP5. However the manuscript does not demonstrate that it is "more realistic". The authors need to provide solid arguments for this realism or remove any such claims.

46) page 38, lines 29-31: "We ignore scenarios with high levels of solar activity because the Sun just left such an episode (called grand solar maximum), and several studies suggest that it is very unlikely to return to one in the next 300 years". As pointed out above (see point 9 of this report), predictions of anything beyond the next cycle are

affected by chance (in the sense that the activity level can be changed significantly by singular events that in turn cannot be predicted). It also seems that statistically, from the record of past solar activity, a grand minimum is equally unlikely as another grand maximum. According to Solanki and Krivova (2011, Science 334, 916), "Half the grand maxima in (6) were followed by one or more subsequent grand maxima before a grand minimum finally occurred." (The reference (6) in this sentence is to Usoskin et al. 2007, Astron. Astrophys. 471, 301.) Consequently, the authors should revise the above statement and find new arguments for why they choose to concentrate on just low values of solar activity for the coming centuries.

47) "Nonetheless, memory effects associated with these periodic reversals play a major role in determining solar variability on multi-decadal time scales, and to some degree are decoupled from the short-term variability. This is our prime motivation for considering predictions on multi-decadal time scales." As given, this is just a statement without a physical basis. As this is their prime motivation for the predictions, the authors should provide solid evidence for such memory effects and the decoupling of multi-decadal from decadal variability.

48) Sections 3.1-3.4. I have significant doubts about the results presented in these sections. Section 3.1 presents three forecast methods, chosen seemingly arbitrarily from all those that have been proposed. As far as I can tell, they are all in one way or another linear. For a strongly non-linear system such as the solar dynamo, I see little value in using linear forecasting methods. I argue that applying inappropriate, but complex sounding forecasting techniques projects a sense of accuracy where none is present in reality. The performance of the techniques is discussed in Sect. 3.3 and Fig. 21c. From Fig. 21a I get the impression that the errors in the Phi forecast are comparable (and over some periods exceed by a factor of 2-3, e.g. around 2080-2090) the values of Phi. Around 2200 various methods give Phi of about 100 to 400, and the forecast error is 150 for all methods. This essentially means the range of 0 to 550 (with ∼600 being the highest value measured during the modern Grand max). In summary,

[Figure]

Fig 21 shows that the three methods often give hugely different results (which is not surprising and simply reinforces that solar activity cannot be reliably predicted using such simple techniques and possibly cannot be predicted at all on these time scales). The authors then consider the mean of these three results, claiming that it represents "the most likely level of solar activity". Such an approach can hardly be called scientific and cannot lead to a "more realistic" forecast than CMIP5. Thus, the mean of 3 more or less random numbers is still a more or less random number of little value. The construction of the "extreme" scenario is also difficult to follow. A lot of the description is rather opaque. All this seems such a complicated way of computing something that is likely very unreliable anyway and does not provide that much reliable information. For example, the reference scenario seems to be somewhat below present conditions and stays nearly constant at that level, while the extreme scenario drops down to the Maunder minimum and basically stays there. Is that so much different from what was done for CMIP5. I would find it a lot more honest towards the reader to not invoke all these different methods, but rather to make a clear and simple assumption and to show the result it gives. This result may turn out not to be very different from what the authors are proposing now, if the authors make the appropriate assumptions. BTW, what is the meaning of a negative modulation potential (Fig. 21B) and how is it obtained?

49) page 39, line 17: "... is certainly presentat some level." "... is certainly present at some level."

50) page 39, line 35: "... using the geomagnetic reconstruction of the open solar flux Lockwood et al. (2014)." "... using the geomagnetic reconstruction of the open solar flux (Lockwood et al., 2014)."

51) p. 40, line 11-12: "According to solar dynamo models, the solar-cycle averaged modulation potential (and the sunspot number) cannot be predicted more than a few decades ahead." This statement is not entirely consistent with p. 39, line 7-8 "As of today, even predicting the cycle amplitude one cycle ahead remains a major challenge (Pesnell, 2012)." Which statement do the authors actually support, one cycle ahead or

multiple cycles ahead? According to Cameron et al. and Jie et al. (referred to earlier in this report; see point 9), the statement on p. 39 appears to be the valid one and the statement on p. 40 should be changed accordingly.

52) page 40, line 12-14: "The observed modulation potential has an autocorrelation function that decays exponentially with a characteristic time of 48 $\pm$ 5 years. This quantity can be interpreted as the time beyond which memory is lost." This is weak und unconvincing evidence of memory. Earlier in the same paragraph, the authors state that they are dealing with 22-year averages of the modulation potential Phi. The decay time of 48 years is hence basically the time resolution of the data (based on the Nyquist frequency). I do not see this as evidence for a memory, just that the true resolution of the data is not very high. In addition to this argument, there may well be hidden connections between data points lying close in time, so that they are not entirely independent. This can be the case in cosmogenic isotope data, so that using this as an argument of memory should be done with considerable care. The authors should first convincingly show that individual data points are completely independent, before making claims of memory.

53) Page 40, line 15-16: "To the best of our knowledge, no existing method has been able to meaningfully predict solar activity more than 60 years ahead." This seems to imply that there are methods that can predict up to 60 years ahead. I would like to hear more about these. E.g. how do the authors know that they work up to 60 years ahead, without waiting for another 60 years to find out? Unless, of course, they are referring to methods that are at least 60 years old and that I seem to have missed. I have seen many so-called predictions tuned to reproduce past data exceptionally well, but then do rather less well when predicting even the next cycle. This statement also is not consistent with p. 39, line 7-8 and other work.

54) Page 45, line 1: "The resulting SSN time series of both future scenarios have then be used" "The resulting SSN time series of both future scenarios have then been used"

[Figure]

55) page 53, line 5: NRLTSI2/NRLSSI2 are empirical models not semi-empirical ones.

56) page 54, lines 8-10: the statement is not clear (see above);

57) page 54, line 13: The statement is too ambiguous. Which satellite measurements are meant?

---

## Author Comment (AC2) · 19 Sep 2016

During the last few months we realized that we did a mistake in constructing the variable preindustrial control forcing dataset. Therefore we would like to revise it and will change the text in the paper as indicated below. Instead of scaling solar cycle amplitudes to the rather small amplitudes of the preindustrial period around 1850, we will scale the solar cycle amplitude to be representative of the last grand solar maximum. We will however make sure that the mean of this constructed solar cycle is in agreement with the pi control solar forcing value. By prescribing a rather strong solar cycle amplitude, we expect to learn how the solar cycle contributes to natural climate variability and whether the synchronization between the NAO and the solar cycle (Thiéblemont et al., 2015) is a robust feature.

The new text reads as follows:

[Figure]

For those groups that are interested, we also provide a 1000-year solar forcing time series with 11-year solar cycle variability included but without long-term trend (Fig. 25). This time series still has slightly different solar cycle amplitudes and also preserves the variable phase of the solar cycle, however, the solar cycle mean activity level is held constant as compared to the reference scenario in Fig. 23. By running a second PI control experiment with solar cycle variability, this provides one additional periodic forcing on top of the seasonal cycle. Since the PI control is also used to determine model variability at decadal timescales, including a solar cycle would certainly change the mean climate and the variance of the control experiment as compared to the "standard" control experiment with constant 1850 solar forcing. However, not including the solar cycle variability may underestimate the variance of the climate system and may lead to climate system biases. Ideally the groups would do two PI control experiments: one with and one without solar cycle variability.

The variable PI control forcing has been generated by scaling the annual and sub-annual components of the REF forcing dataset to a constant solar cycle mean activity level representative for the time period last grand solar maximum.. The scaling procedure for SSI and F10.7 is described in Appendix G. Constant background components have been added. These have been adjusted such that the mean values of the resulting SSI and F10.7 time series are consistent with the constant PI control forcing. The scaling of geomagentic Ap and Kp indices is described in Appendix H. MEE-induced ion-pair production rates for the variable PI control forcing have then been calculated from the scaled Ap data. GCR-induced ion-pair production rates have been calculated using a constant value representative for the 1850–1873 period. The variable PI control proton forcing is identical to the REF forcing since it does not include any long-term trend. Note that the temporal averages of SSI, TSI, F10.7, Ap, Kp, as well as GCR-induced ion-pair production rates are fully consistent wit the values provided in the constant PI control forcing dataset. This, however, is not the case for the proton and MEE forcings, which, in the latter case, do not account for large, sporadic events. The variable PI control dataset (see Fig. 25) covers the time period from 1.1.1850 until

9.9.2053 (end of solar cycle 27). The dataset can be extended to cover 1000 years by multiple repetition of the solar cycle sequence 12–27. The first 450 years of the resulting forcing time series are consistent in solar cycle phase and short-term fluctuations with the REF and EXT datasets. Solar forcing only experiments based on variable PI control, REF, and EXT forcing data would therefore be ideally suited to address the impact of long-term solar activity variations on the climate system.

Appendix G, last paragraph: For the construction of the variable PI control forcing, anual and sub-annual SSI components are scaled individually to grand solar maximum average conditions at each wavelength bin. The scaling has been performed for the D+ and D− components based on SD(D+) and MAD(D−), respectively, as in the the future scenario construction. For A+ and A−, we use the corresponding solar cycle averages to construct time-resolved scaling functions. The background contributions SSIbg were set to constant values. These have been adjusted such that the mean values of the resulting SSI and F10.7 time series are consistent with the constant PI control forcing. The same procedure was also applied to the F10.7 radio flux.

---

## Author Comment (AC3) · 20 Sep 2016

Dear Marty,

thank you very much for your suggestions and comments. We specify in our point-by-point response below how we plan to include them into the revised version of the paper.

Katja and Bernd on behalf of all authors

P3 L29: The text mentions uncertainties in the irradiance measurements, and there is no question that these uncertainties can be sometimes larger than solar variability at some wavelengths. There is a brief discussion that uncertainties in models are sometimes difficult to assess. Measurements only go back a few solar cycles, and both irradiance models are based on interpretation of those measurements. Extrapolating these proxy models to the past and future requires several assumptions about

proxy relationships remaining invariant. For example, the numerical relationship between sunspot number (or area) and the sunspot blocking function in NRLSSI2 could change if the sunspot contrast evolves over time. So it is not automatically true that we understand proxy relationships well enough over long timescales. Model development will continue to improve as we continue to make better measurements. State of the art model reconstructions as described in this manuscript are the best we currently have, but their uncertainties are also still significant.

Reply: We fully agree that all models (whether empirical or semi-empirical) most likely underestimate the SSI when moving away from the space age. You are right in pointing that out, and this will be emphasized more strongly in the text.

P9 L14: Averaging two quantities that disagree produces a result that is also not likely to be correct. Calling this "the most reasonable approach" is perhaps controversial. Maybe calling it "a reasonable approach" would be more appropriate.

Reply: This approach has indeed been met by some skepticism. We have here two models that have been derived almost independently, and as of today there are no objective criteria allowing us to prefer one to the other. In that context, from a statistical point of view, there is one clear and sound solution: just average the two. This solution is also what is advocated by the IPCC when combining GCM model predictions, see for example [D. Smith el al., Real-time multi-model climate predictions, Climate Dynamics 41 (2013), http://adsabs.harvard.edu/abs/2013ClDy...41.2875S]. We shall update our text to insist more heavily on this sound justification for averaging the two SSI models. We will also follow the reviewers suggestion and call our approach "a reasonable" one.

P9 L30: The comment that F10.7 was a good proxy for EUV at one time, but "this may not be true anymore" reinforces my discussion about page 3 above. It is an assumption that proxy relationships do not change over time, and this assumption must factor into the estimation of model uncertainties.

Reply: You are correct, and this is something that we should clarify in the text. We are

working on it in SOLID, but as you may imagine, it is difficult too to provide realistic confidence intervals for such long-term variations.

Overall, I think this manuscript does an excellent job in describing the recommended solar forcing for the climate community.

Reply: Thank you!

---

## Author Comment (AC4) · 20 Sep 2016

Dear reviewer,

thank you very much for your thorough and detailed review which clearly helps to improve the clarity of the manuscript. Please find below our detailed point-by-point response to your concerns which we all hope to address satisfactorily.

Katja Matthes and Bernd Funke on behalf of all coauthors

1) page 1, line 10 (abstract): "The TSI and SSI time series are defined as averages of two (semi-) empirical solar irradiance models, namely the NRLTSI2/NRLSSI2 and SATIRE-TS." Two comments: a) NRLTSI2/NRLSSI2 are empirical, not semi-empirical models. This is correctly explained later in the text of the manuscript but the statement in the abstract is misleading. b) More importantly, could the authors describe why they have taken recourse to this unusual step of averaging two independent models? Both

models are constructed differently, and both have to some extent been tested against data. What the authors do is to provide a new model that is largely untested with regard to solar data. More comments on this aspect will be made later.

Reply: a) We shall clarify the text accordingly. b) You are not the first one to object against this averaging (see e.g. also the first reviewers comment), which, however, is actually statistically rooted and well-justified. As of today, there is no community consensus (established by independent means) on which of the two SSI models provides a more accurate description of the SSI. For that reason we have no objective means for preferring one model to the other, or for assigning different weights to them. Meanwhile, these two models have - as you say - been constructed independently, and this precisely what gives us a strong statistical justification for just taking their average. This solution is also what is advocated by the IPCC when combining GCM model predictions, see for example [D. Smith el al., Real-time multi-model climate predictions, Climate Dynamics 41 (2013), http://adsabs.harvard.edu/abs/2013ClDy...41.2875S]. We shall update our text to insist more heavily on this sound justification for averaging the two SSI models. We will also follow the reviewers suggestion and call our approach "a reasonable" one.

2) page 2, lines 1-2: "The slight negative trend in TSI during the last three solar cycles in CMIP6 is statistically indistinguishable from available observations". What do the authors mean by this? Different TSI composites indicate different trends. There are implicit assumptions underlying this statement that need to be spelt out and the reasoning clarified.

Reply: This result is indeed quite recent, and is detailed in an article that is about to be submitted. In this article, we demonstrate that the latest TSI composite, when made with realistic confidence intervals, makes it very difficult, if not impossible to conclude about the existence of a downward trend. We shall cite this reference in the revised manuscript.

3) page 2, line 5: CMIP6 cannot be tested against CMIP 5. It can only be compared to CMIP5.

Reply: You are correct, we shall clarify the text.

4) page 2, line 6: The expression "background SSI" is neither clear nor used in the literature.

Reply: You are correct, we shall clarify the text.

5) page3, line 5: "Because of its prominent 11-year cycle, solar variability may offer a degree of predictability for regional climate and could therefore help reduce uncertainties in decadal climate predictions." However, the solar cycle itself is notoriously hard to predict and predictions of upcoming solar minima have not been particularly successful, so that the statement does seem too optimistic. But possibly the authors are not concerned with such niceties here?

Reply: Indeed, as you pointed out, we are dealing here with decadal time scales, ignoring the more subtle dynamics of sub-solar-cycle variations, which can be appropriately considered as a modulation of approximately 11-years.

6) page 3, lines 23-24: "The quantitative assessment of radiative solar forcing has been systematically hampered so far by the large uncertainties and the instrumental artifacts that plague TSI and SSI observations" The TSI observations are signiiÌĹnĒĞA Ìĺcantly more precise than those of SSI (especially SSI in the near UV and visible spectral domains). This makes the quoted sentence misleading.

Reply: We shall clarify the text to distinguish the relatively better shape of the TSI.

7) page 3, line 28: The IAU resolution (Mamajek et al. 2015) adopted the result of Kopp & Lean (2011). The latter is a well-known and well-cited paper in a refereed journal, while the former is not a scientific paper at all, is not properly published, and the text in the reference is not really useful. Please replace the reference to IAU resolution with the Kopp & Lean paper everywhere in the text.

Reply: A new and peer-reviewed publication (Prsa et al. (2016), http://adsabs.harvard.edu/abs/2016AJ....152...41P) now replaces the one by Mamajek et al. (2015). In contrast to the article by Kopp and Lean (2011), which focuses on the scientific background, the two articles on the IAU resolution explain what is exactly meant by a nominal TSI and hence are important too.

8) page 3, line 29: The proxy reconstructions of the TSI do not exhibit "occasional phases of unusually low nor high activity", the Sun does. Also, Usoskin et al. 2014 did not mention TSI reconstructions at all.

Reply: We shall rewrite the sentence accordingly.

9) page 3, line 32: "There is growing evidence for the Sun to enter a phase of low activity near 2050, after a grand maximum that peaked during the 20th century." The authors should explain what they base this claim on. The solar dynamo is chaotic, possibly even stochastic and it is even difficult to predict the next solar cycle much before the previous minimum. Going beyond the next cycle is even tougher. See e.g. * J. Jiang, R.H. Cameron, M. SchuÌĹssler, 2015: The cause of the weak solar cycle 24; ApJL 808 L28 * Cameron et al. 2013: Limits to solar cycle predictability: Cross-equatorial flux plumes, A&A 557, A141

Reply: Several recent studies (excluding here the more speculative ones) have suggested that the Sun may be entering a phase of low activity, and this issue was recently addressed at ISSI by a VarSITI forum. This will be clarified in the text.

Regarding predictions of the solar cycle, we explicitly state that "As of today, even predicting the cycle amplitude one cycle ahead remains a major challenge (Pesnell, 2012).", so we don't seem to disagree, do we ?

Fortunately, since we focus on multi-decadal time scales only, this complexity of solar cycle prediction is beyond the scope of our study.

10) page 4, line 12: " . . . with respect to the CMIP5 solar forcing recommendation."

Is the CMIP5 paper supplying the solar input cited? At least so far "CMIP5" is not associated with such a paper.

Reply: CMIP5 provides annually-resolved TSI since 1610, and monthly-resolved since 1882, see http://solarisheppa.geomar.de/cmip5. You are right, there is no paper associated with CMIP5 solar forcing recommendation, it is just described on the indicated website.

10a) page 4, line 20: "Lockwood et al., 2019" This is likely "Lockwood et al., 2010"; also to be corrected on page 74

Reply: Thank you for pointing this out.

11) page 6, line 15: " . . . and one observational estimate (SOLID), . . ." Is this correct? From the evidence given in the paper, SOLID is not an "observational estimate" but an empirical model (see below).

Reply: The SOLID composite, built by statistical means, and entirely based on SSI observations only, includes a few proxies to help fill the gaps. We call it an observational composite, and not en empirical model, because it is truly different from models such as NRLSSI2, which involve physical assumptions. As described below this will be better explained in the revised manuscript.

12) page 6, lines 26-27: The sentence simplifies things too much. Faculae are not "the Mg II index" and dark sunspots are not "sunspot area". These proxies are used to represent the contributions from faculae and sunspots to the irradiance. This should be written more carefully to reflect the actual relationships.

Reply: We shall rewrite the text as you suggest.

13) page 7, line 8: ". . ..SORCE SSI observations Lean and DeLand (2012) the wavelength dependent scaling coefficients ..." Change to: "....SORCE SSI observations (Lean and DeLand, 2012) the wavelength dependent scaling coefficients . . ."

Reply: Thank you for pointing this out.

14) page 7, lines 5-15: The description of the adjustments in the NRLSSI2 model is not clear. The authors should explain better how the adjustments are done.

Reply: As you propose, we will expand the NRLSSI2 model adjustments better in the revised manuscript.

15) page 7, line 19: What is SATIRE-TS? It does not become clear from reading the paper, as references are given only to SATIRE-T and SATIRE-S. TS is used multiple times, so that it cannot be a typo. Is TS some combination of the two models? What kind of combination?

Reply: The SATIRE model we use is indeed a blend of SATIRE-S and a SATIRE-T that has been run with the sunspot data we had specifically provided for CMIP6, and thus differs from the official SATIRE-T. We shall rephrase that in the text, and replace SATIRE-TS by SATIRE to avoid ambiguities.

16) page 7, line 28: The reconstruction made by Dasi-Espuig et al. (2014) is not based on to expectations.

Reply: We'll remove that citation.

17) page 8, line 3: "In NRLSSI2, this internal consistency also applies to the integral of the facular and sunspot contributions to SSI to their respective counterparts in TSI (see Coddington et al. (2015) for more details). Why are the authors stressing this point? This sounds rather trivial. Doesn't every model that distinguishes between spots and faculae do that? Or are they implying that SATIRE does not fulfil such internal consistency? They should either explain why this is such an achievement and is unique to NRLSSI, or they should remove this unnecessary sentence.

Reply: We'll remove lines 3 and 4.

18) page 9, lines 5-6: "The controversial out-of-phase behavior of SORCE/SIM observations in that band (Harder et al., 2009) are likely to be an instrumental artefact (Lean and DeLand, 2012; Ermolli et al., 2013) but this has not yet been corrected in the SOLID composite." If SORCE is wrong, is then the SOLID composite the best one to use to test the SSI models? If models are chosen by how close they are to SOLID, this could have unwanted effects for the atmospheric chemistry and finally climate modelling. As shown by Haigh et al. (2010, An influence of solar spectral variations on radiative forcing of climate, Nature, 467, 696), the SSI variability found by SORCE has a major effect on atmospheric ozone, changing its concentration in a way contrary to expectations.

Reply: You are right, the text in its current form is misleading. The SOLID team has decided to deliver a first version of the SOLID observational composite that is totally devoid of model assumptions, and does not rely on nudged data. This raw composite can then be tested against SSI models, and from there onwards we may decide if some of the original datasets should be corrected (or rather, if their confidence intervals should be increased to reflect our understanding of the data). The description of the SOLID database is now extended in the paper (see answer to comment 31) which will hopefully also clarify this point.

19) page 8, line 14 "In NRLSSI2, the proxy index for sunspot darkening is the sunspot area as recorded by ground-based observatories in white light images since 1882 (Lean et al., 1998). Values prior to that are estimated from the sunspot number." Does not also the SATIRE model rely on sunspot areas over the period of time covered by Greenwich observatory? It seems to according to * Krivova et al. 2010: Reconstruction of solar spectral irradiance since the Maunder minimum; JGRA, DOI: 10.1029/2010JA015431 Please comment.

Reply: The referee is right, SATIRE relies on sunspot areas and then sunspot number to expand the model in time. We shall modify the text accordingly.

20) Sect. SOLID composite After reading this section the exact nature of the SOLID

composite is still very unclear. A single paper (SchoÌĹll et al., 2016) is cited, which explains only the preprocessing of the data. As long as SOLID is not properly documented, it is important to make the description here sufficiently clear so that readers who don't know the nitty gritty of the SOLID approach can follow and, if necessary make their own judgement. So far, this composite (if it is really a composite at all and not an empirical model, see below) is not widely-known or established in the solar community.

Reply: Please see our detailed answer to comment 31 below.

21) page 9, line14: "..., the most reasonable approach consists in averaging both reconstructions, weighted by their uncertainty. But this means that yet another model is produced, one that is untested, except for the rudimentary tests briefly discussed in the paper. It is not clear why this is "the most reasonable approach"

Reply: As already explained above, the averaging is well justified. As of today we do not see a better approach and call our approach "a reasonable" one. Obviously, the outcome can be considered as another model, or rather meta-model, but for obvious reasons it is premature to expect the outcome to be thoroughly tested against observations. Having said that, we do provide some comparisons in Fig. 3 during the satellite era. We rely on SATIRE and NRLSSI precisely because these have been tested extensively against observational data by their respective teams.

22) page 9, line 17: "... SATIRE-TS (Yeo et al., 2014)." What is SATIRE-TS? Is it a typo or a combination of SATIRE-T and SATIRE-S? Please define it. In Yeo et al. (2014) I could find a description of SATIRE-S, but no mention of SATIRE-TS, so that this reference seems not to be relevant (unless TS is a combination of T and S).

Reply: Please see answer to comment 15.

23) page 9, line 24: "The EUV band (10-121 nm) is required for CMIP6 but is absent from NRLSSI and SATIRE. We added it with spectral bins from 10.5-114.5 nm

by using a nonlinear regression from the SSI in the 115.5-188.5 nm band, trained with TIMED/SEE data from 2002 to 2009." Indeed, this is a difficult situation and the authors have done the reasonable thing and somehow modelled this wavelength region themselves. The procedure chosen may be fine and may produce a good result, but without being shown anything it is hard for the reader to judge. Please provide a figure and some more explanation.

Reply: We'll add a figure on the EUV band.

24) page 9, line 29: "Let us note that while the F10.7 index is a good proxy for EUV variability on daily to yearly time scales, this may not be true anymore on multi-decadal time scales." This is a good point.

Reply: Thank you!

25) page 10, line 3: ". . . observed composite from PMOD" This sounds strange since the composite was not observed. Rather it is based on a set of observations of TSI.

Reply: This should indeed be "observational composite".

26) page 10, line 7: "All TSI records agree well on daily to yearly time scales, and in some cases (e.g. NRLTSI1 and NRLTSI2) they match as well on multi-decadal time scales." Aren't both models part of the same model family and are founded on the same proxies? I am not sure what is remarkable about them being consistent with each other? The authors can leave this – it is not an important point – but I would like to understand why they stress this agreement, which I would have naively thought to be trivial.

Reply: NRLTSI1 and NRLTSI2 are based on the same method and proxy, but were trained with different datasets: NRLTSI1 were trained with a previous version of the PMOD composite while NRLTSI2 was trained with SORCE/TIM observations. Even if not remarkable, it is worth mentioning it.

27) page 11, Figure 1: The orange curve described as CMIP5 goes till 2015. What

exactly is plotted? Are these extrapolated CMIP5 data? Or are the plotted CMIP5 data regularly updated and are available up to 2015?

Reply: Plotted are the extended NRLTSI1 data that have been the basis for CMIP5 (from 1882 until 2010). We will adopt the plot and the figure caption to make this clear.

28) page 11, line 12: "The major difference come from SATIRE-TS .." "The major difference comes from SATIRE-TS .."

Reply: Has been corrected.

29) page 12, Figure 2, bottom left diagram Why is NRLSSI2 producing such strange cycles between âĹij1940 and 1960? These certainly do not look realistic. How come, these strange cycle shapes are restricted mainly to the visible (although there may be some sign of similar behavior in the IR)? Should not also the UV cycles behave strangely in order to compensate for the behavior in the visible and produce a reasonable TSI? Has the TSI produced by NRLSSI2 been compared with measured time series? Also, something seems to be going wrong right at the start of the time series. How do the authors explain and compensate for these problems?

Reply: We have sought clarification from the NRL team. The rather modest solar cycle variation in the visible radiation in the new NRLSSI2 model is thought to be related to the scaling of the two sunspot area data bases, the Royal Greenwich Observatory (RGO) sunspot areas (before 1976) and the areas in the SOON (USAF) database, which have to be combined. This scaling affects the relative roles of sunspots and faculae before (and including) 1976. Apparently, there is quite bit of debate about how these two datasets relate - with scaling factors ranging from the RGO being 20% to 50% larger than SOON areas. NRLSSI2 uses a scaling factor of 0.67, in which the GW sunspot areas are assumed to be $1/0.67 = 1.5$ times larger than SOON. The problem at the beginning and end of the time series is caused by smoothing of the data taken between 1882 and 2010. We will update the plot and take the longer timeseries (available on an annual basis before 1882) in order to avoid this smoothing problem.

[Figure]

30) page 12, line 4: " ...  and higher-quality data from the SORCE mission on the rotational timescale in NRLSSI2, . . ." But isn't SORCE giving wrong trends for SSI? This is what the authors claim multiply elsewhere in the paper.

Reply: Yes indeed, but rotational time scales (<81 days) used to train the NRLSSI2 model are not affected by long-term drifts of the SORCE instrument.

31) page 13, Figure 3: Fig.  3 is confusing and does not agree with the text of the paper, mainly regarding SOLID. Earlier it was said that SOLID is based entirely on observed data (page 8, line 20: "More specifically it is derived as the weighted mean of all available SSI observations in the satellite era.") If that were true, then the green curve should be a lot closer to the plotted observations. Thus, it is not clear why The SOLID composite departs so strongly from SORCE at a time when that is the only data set used (according to the authors)? Or are other data sets also used after all? The SSI variations shown by SOLID seem to be smaller than of all the instrumental records, except maybe UARS SOLSTICE (it is hard to see - there are too many light colors in this plot). Anyway, SOLSTICE shows a behavior completely inconsistent with an SSI composite of the observations. This is a serious problem that points to a fundamental inconsistency in the paper.  Another strange feature of this plot is that SOLID covers also the 1950s and 1960s when there were no SSI data available. How does SOLID produce something at those times if it is purely based on SSI data?  The description given in the paper is totally inadequate and obviously seriously misleading. However, Fig.  3 very strongly suggests that the SOLID "composite" uses either a proxy (possibly something like 10.7 cm flux?) or makes really strong changes to the data while processing them.  In either case, I would strongly oppose calling it a composite of SSI observations. Rather Fig.  3 clearly shows that it is an empirical model. If the authors want to maintain that SOLID is a composite of observed SSI, then they should provide a detailed explanation that goes far beyond the inadequate one in the current version of the paper. This should include a list of all data sets that enter into the SOLID "composite" and all the steps that are undertaken to produce it.  Also, they should provide

a convincing explanation why SOLID differs so strongly from the observational data. Fig. 3 raises an issue regarding fairness and bias in the paper. If SOLID is indeed an empirical model, and I have seen no evidence to counter this in the paper, I see no advantage in using SOLID to "test" the other two models. Indeed, if SOLID is a model (and an unpublished one at that), why is it being discussed ahead of the numerous other (published!) models in the literature. I see only two paths that that the authors can follow: a) Either remove SOLID completely from the publication and instead compare the averaged model that the authors have produced more rigorously with the observations directly, b) or discuss SOLID on an equal footing with the other SSI models that the authors simply ignore in this version of the paper. Irrespective of which of these paths the authors follow, I strongly urge them to use the original SSI observations to test the new model data set obtained by averaging NRLSSI2 and SATIRE (-TS?).

Reply: Let us answer stepwise:

i) Earlier it was said that SOLID is based entirely on observed data (page 8, line 20: "More specifically it is derived as the weighted mean of all available SSI observations in the satellite era.") If that were true, then the green curve should be a lot closer to the plotted observations.

The previous version of Fig. 3 did not include all observational datasets going into SOLID. Figure 3, right panel, has been revised and now also includes the scaled SORCE/SOLSTICE observations which were not shown in the previous plot as the dataset does not cover the full spectral range from 200-400nm. In addition a table with all observational estimates used in the SOLID composite has been added to the SOLID description. Also, Fig. 3 has been restricted now to the satellite era and starts only in 1980.

ii) Thus, it is not clear why The SOLID composite departs so strongly from SORCE at a time when that is the only data set used (according to the authors)?

For the spectral range (200 - 400nm) shown in Fig. 3 apart from SORCE/SIM many other observations are available, such as NIMBUS, NOAA9, NOAA11, UARS/SOLSTICE, UARS/SUSIM. Moreover, SORCE/SOLSTICE also enters the SOLID composite for the spectral range 200-320nm. SORCE/SOLSTICE has a relatively low uncertainty (compared to the other instruments) and as such a relatively high weight in the composite. Therefore, it is plausible that the integrated SSI from 200-400nm deviates from the SORCE/SIM dataset. Note, SORCE/SOLSTICE was not shown in Fig 3 as it does not cover the full spectral range of the figure. It has now been added so that the influence of the various datasets is visible. Some text to explain this will be added to the paper.

iii) Another strange feature of this plot is that SOLID covers also the 1950s and 1960s when there were no SSI data available. How does SOLID produce something at those times if it is purely based on SSI data? The description given in the paper is totally inadequate and obviously seriously misleading. However, Fig. 3 very strongly suggests that the SOLID "composite" uses either a proxy (possibly something like 10.7 cm flux?) or makes really strong changes to the data while processing them.

We thank the reviewer for pointing this out. One technicality to derive the SOLID composite is that in order to apply the same scale-wise decomposition, all datasets have to cover the full time interval under consideration. This means for some dataset gaps have to be filled or the datasets have to be extended in time. To achieve this we use (observed) proxy data and the "maximization expectation" technique which makes use of the original signal in the data. This technique also allows us to cover the times before observations are available, e.g. 1959-1960. We agree with the reviewer that more details need to be given in the paper and will add those. However to avoid confusion, we will restrict the figure to the satellite era as already explained above.

iv) In either case, I would strongly oppose calling it a composite of SSI observations. Rather Fig. 3 clearly shows that it is an empirical model. If the authors want to maintain that SOLID is a composite of observed SSI, then they should provide a detailed

explanation that goes far beyond the inadequate one in the current version of the paper. This should include a list of all data sets that enter into the SOLID "composite" and all the steps that are undertaken to produce it. Also, they should provide a convincing explanation why SOLID differs so strongly from the observational data.

The SOLID composite is produced using a statistical framework in a maximum-likelihood sense and no physical assumptions go into it. More details will be given in the revised SOLID section including information on the individual data sources used in the composite as well as their statistical combination.

v) Fig. 3 raises an issue regarding fairness and bias in the paper. If SOLID is indeed an empirical model, and I have seen no evidence to counter this in the paper, I see no advantage in using SOLID to "test" the other two models. Indeed, if SOLID is a model (and an unpublished one at that), why is it being discussed ahead of the numerous other (published!) models in the literature.

As already stated above, in our view the SOLID composite is not an empirical model as no physical assumptions go into it. We use the composite as an independent source of (observed) information, and as such it is a very valid dataset to which the recommended CMIP6 dataset is compared to.

vi) I see only two paths that that the authors can follow: Either remove SOLID completely from the publication and instead compare the averaged model that the authors have produced more rigorously with the observations directly, b) or discuss SOLID on an equal footing with the other SSI models that the authors simply ignore in this version of the paper.

To follow option b) more details will be given in the SOLID section including information on the individual data sources used in the composite as well as their statistical combination.

vi) Irrespective of which of these paths the authors follow, I strongly urge them to use

the original SSI observations to test the new model data set obtained by averaging NRLSSI2 and SATIRE (-TS?).

In Fig. 3 we do compare the CMIP6 recommended SSI model data with the SSI observations, along with the SOLID composite.

32) page 13, Figure 4 and its discussion in the text: Averaging over one month at activity maximum and minimum does not allow eliminating the rotational cycle in solar variability, so that this iÌĹnËĞA Ìĺgure mixes information on shorter timescales into the solar cycle variability that the authors want to show. The figure should be redone using at least 81-day averaging. Why are the comparisons in Figures 3 and 4 being done in such broad, seemingly arbitrary wavelength bands, rather than broken up according to the important molecular band listed in Table 1? What is the advantage for the climate community of following the bands used by Ermolli et al. (2013). Also, where would the observations lie in Fig. 4 (to the extent available for exactly these times, which is a limitation of the figure)?

Reply: The referee is correct and Fig.4 now uses a 81-day averaging. This does not change the differences between the reconstruction but the numbers have changed slightly (less cycle variability in general) and we will update the related discussion in the revised manuscript. We think that it is important for the climate community to use the same bins than in Ermolli et al. (2013).

33) page 13, lines 1-2: " . . . the only available measurements are from the SORCE/SIM instrument, which has calibration issues (Lean and DeLand, 2012) . . ." Until now no calibration issues in SORCE/SIM instrument have been reported by the instrumental team. In particular, the paper by Lean & DeLand does not identify any calibration issue.

Reply: You are correct, Lean and Deland do not explicitly evoke calibration issues. These are unofficial. We shall clarify this.

34) page 14, line 5: "In the NIR CMIP6 shows slightly larger variability than CMIP5 and remarkable here is the largest variability in NRLSSI2." Why is this remarkable? Both SATIRE & NRLSSI2 reproduce TSI. NRLSSI2 has a smaller variability in the UV and this must be compensated by NRLSSI2 in the IR. Or is there something more complex at work here that I am missing? Please explain or simplify the text.

Reply: We shall reformulate the text and simply remove the part "...and remarkable here is the largest variability in NRLSSI2".

35) page 17, line 1: "Solar activity and hence spectral irradiance vary between different solar cycles. However, these differences are small compared to the total 11 year solar cycle amplitude . . ." This has been the case in the second half of the 20th century, but the sizes of cycles vary between zero and the very high amplitude of cycle 19. This is only a minor quibble, however.

Reply: You are correct. We shall clarify this.

36) page 19, line 18: "... produces slightly higher SW heating rates than NRLSSI1(CMIP5)" "... produces slightly higher SW heating rate differences than NRLSSI1(CMIP5)"; the diagram shows the differences between heating rates for solar minimum and solar maximum.

Reply: Yes, thank you, we will change the sentence accordingly.

37) page 20, Figure 5: "Impact of solar forcing according . . ." add: for perpetual solar minimum conditions

Reply: We shall add this.

38) page 21, Figure 6: "CMIP6 SSI differences in % for perpetual solar minimum conditions . . ." It may not be immediately clear what differences are actually meant here, i.e. differences in which parameter; add e.g.: "CMIP6 SSI differences of the solar irradiance in % . . ."

Reply: We shall add this.

39) page 21, line 1: "More important for the solar ozone signals seems to be the choice of the CCM (with its specific photolysis scheme, see also Fig. 8), especially for the lower stratosphere (10 hPa and below)." This is true for the lower stratosphere only; above that the dataset-induced differences are larger than the model-induced ones, in particular in the lower Mesosphere.

Reply: Yes, we agree. We will therefore add: "In the lower mesosphere however, the dataset-induced differences are larger than the model-induced ones."

40) page 21, line 12: "Note that statistically significant irradiance differences between CMIP5 and CMIP6-SSI irradiances are particularly observed between 300 and 350nm . . . (Fig. 8)." In Figure 8 differences in the irradiance amplitude between solar minimum and solar maximum are shown. i.e. "irradiance differences" should be replaced by "differences in the irradiance amplitude".

Reply: Will be changed to "difference in irradiance amplitude".

41) page 25, line 19: ". . .for mesospheric OH production Fytterer et al. (2015b), and for . . . ". . . for mesospheric OH production (Fytterer et al., 2015b), and for . . ."

Reply: Thank you for spotting this typo.

42) page 27, Figure 11: SSN scaled by a factor of 0.67 should have larger values (in Figure 13 SSN scaled by a factor of 0.741 has values above 200); a factor of 0.067 appears to be much more reasonable.

Reply: The factor is indeed 0.067. Thank you for spotting this.

43) page 33, Figure 16: There are differences between the caption and the labels in the diagram. caption: 70–90oS (left) and 70–90oN (right); in the diagram: 70–90oS (right); 70–90oN (left) caption: 0.01 hPa (upper panel) and 0.1 hPa (lower panel); in the diagram: 0.1 hPa (upper panel) and 1 hPa (lower panel)

Reply: The caption is wrong: NH is shown on the left and SH on the right. The pressure levels are 0.1 hPa (top) and 1 hPa (bottom).

44) page 36, line 13: "Since fast transient solar energetic particle events often occur at the background of enhanced geomagnetic disturbances, straight-forward computation of the particle trajectories in a realistic geomagnetic field is needed" Why are fast transient solar energetic particle events relevant in this context? This paragraph deals with the penetration of GCRs in the Earth's atmosphere.

Reply: Since the reviewer thinks this is confusing, we can modify this sentence as follows: the text "Since fast transient ... are fully rejected (Cooke et al., 1991)." can be substituted by "This shielding is usually parameterized in the form of the effective geomagnetic rigidity cutoff, so that only particles with rigidity/energy exceeding the cutoff can penetrate to the atmosphere at a given location while less energetic particles are fully rejected (Cooke et al., 1991)."

45) page 38, line 25: The CMIP6 future solar forcing is different from that of CMIP5. However the manuscript does not demonstrate that it is "more realistic". The authors need to provide solid arguments for this realism or remove any such claims.

Reply: The CMIP5 solar forcing recommendation was to simply repeat solar cycle 23. Many climate modeling groups ended up repeating the last four solar cycles because they argued that solar cycle 23 was special. Observations of a wide range of solar activity indices clearly demonstrate significant cycle to cycle variability, and long-term trends [Solanki et al 2000, Usoskin et al 2016]. Therefore the CMIP5 solar forcing recommendation was certainly unrealistic. For CMIP6 we have developed solar forcing scenarios which include cycle to cycle variability, and long term trends, both of which are established from observations or reconstructions of solar activity metrics. Therefore, taken purely as a scenario rather than a prediction, we fail to see how the CMIP6 future solar forcing recommendation can be less realistic than the CMIP5 solar forcing recommendation.

Solanki S.K., Schussler M., Fligge M.: Evolution of the Sun's Large-Scale Magnetic Field Since the Maunder Minimum. Nature 408 , p. 445-447 (2000).

I.G. Usoskin, G.A. Kovaltsov, M. Lockwood, K. Mursula, M. Owens and S.K. Solanki, A new calibrated sunspot group series since 1749: Statistics of active day fractions, Sol. Phys., p.1-24, doi:10.1007/s11207-015-0838-1, 2016 ADS

46) page 38, lines 29-31: "We ignore scenarios with high levels of solar activity because the Sun just left such an episode (called grand solar maximum), and several studies suggest that it is very unlikely to return to one in the next 300 years". As pointed out above (see point 9 of this report), predictions of anything beyond the next cycle are affected by chance (in the sense that the activity level can be changed significantly by singular events that in turn cannot be predicted). It also seems that statistically, from the record of past solar activity, a grand minimum is equally unlikely as another grand maximum. According to Solanki and Krivova (2011, Science 334, 916), "Half the grand maxima in (6) were followed by one or more subsequent grand maxima before a grand minimum iÌLnËĞA ÌÍnally occurred." (The reference (6) in this sentence is to Usoskin et al. 2007, Astron. Astrophys. 471, 301.) Consequently, the authors should revise the above statement and find new arguments for why they choose to concentrate on just low values of solar activity for the coming centuries.

Reply: There is indeed strong evidence against the occurrence of another Grand Solar Maximum within the 21st century. We say this for several reasons. The first is that runs made with various dynamo models (Charbonneau, private comm.) indicate that after a state of Grand Maximum, the Sun is much more likely to move into a Grand Minimum, rather than into another Grand Maximum (see our detailed discussion to point 47 below). Another piece of evidence comes from the various empirical models that we tested using a Monte-Carlo approach, with various parameters. Among the several hundred reconstructions that we made, not a single one showed a grand maximum in the 21st century.

For this particular study, our prime objective is to provide a likely scenario. As stated in the text, the low activity scenario is much less likely, and is merely given to test climate model sensitivity.

47) "Nonetheless, memory effects associated with these periodic reversals play a major role in determining solar variability on multi-decadal time scales, and to some degree are decoupled from the short-term variability. This is our prime motivation for considering predictions on multi-decadal time scales." As given, this is just a statement without a physical basis. As this is their prime motivation for the predictions, the authors should provide solid evidence for such memory effects and the decoupling of multi-decadal from decadal variability.

Reply: It is certainly the case that we currently do not have a reliable scheme for forecasting of solar activity on centennial timescales, whether empirical (based on data) or physical (based on dynamo models). At the purely empirical level, the low probability of another soon-to-occur Grand Maximum can be argued on the basis of (1) the fact that the sun just exited a Grand Maximum, and that return to something closer to the historical norm appears more likely than the opposite; (2) naive extrapolation of the so-called Gleissberg modulation points to lower-than-average activity in the middle of the twenty-first century; (3) more elaborate reconstruction schemes (such as those cited in the paper), notwithstanding their potential failings and uncertainties, also point towards reduced activity in the 21st century. At the physical level, in the (broad) class of non-kinematic dynamo models which generate Grand Minima through non-linear backreaction on large-scale flows, periods of much Higher-than-average activity (arguably equivalent to Grand Maxima in such models) are more likely to be followed by a Grand Minimum than another Grand Maximum; this is because, in such models showing intermittency, collapse to the trivial solution often requires a large excursion at the boundary of the attractor associated with "normal" cyclic behavior, such excursions corresponding to periods of higher-than-average cycle amplitudes. Examples include the models presented in Passos et al. (2012, Solar Physics 279, 1).

[Figure]

Moreover, in models that achieve Grand Minimum-like behavior exclusively through magnetically-mediated amplitude/parity modulation (without intermittent behavior), recurring epochs of much-higher-than-average cycle amplitude are typically separated by epochs of much-lower-than-average amplitude often of similar temporal duration. Examples include the models presented in Tobias 1996 (A&A 307, L21) and Moss and Brooke 2000 (MNRAS 315, 521). In the model of Brooke et al. 2002 (A&A 395, 1013), Grand Maxima are always followed by a long period of low amplitude cyclic behavior. "Memory" in these dynamical models results from the time required for the field to dissipate back to "normal" values, and large-scale flow to then re-establish themselves to "normal" magnitude; and depending on parameter values, that "memory" can be much longer than the primary cycle period. The empirical study by Inceoglu et al. (2016) (Solar Phys., 291, 303) claims that a grand maximum is at reduced probability to be followed by another grand maximum on short timescales.

Of course, there also exist classes of dynamo models that enter and exit Grand Minima (and Maxima) primarily through stochastic driving, making any long term prediction truly impossible. Stochastic driving certainly takes place in the sun via the impact of convective turbulence and vaguaries of active region emergence; but nonlinear magnetic backreaction on the inductive flows also certainly takes place; therefore some level of long term memory must remain, unless stochastic effects completely dominate the fluctuating behavior even on multidecadal timescales.

The only tentative conclusion that emerges from all these various models is thus that Grand Maxima appear more likely to be followed by a Grand Minimum than by another Grand Maximum. This is the basis of our choice to restrict our extreme scenario for the 21st century to a Grand Minimum rather than a Grand Maximum. We have modified/expanded the text in section 1 (p 5), section 3 (p 39) and section 6 (p 55) to be more explicit about the rationale underlying this choice. We have also "softened" our statements regarding the specific activity predictions, to avoid giving the impression that they are physically sounder than they really are.

48) Sections 3.1-3.4. I have signiiÌĹnËĞA ÌĨcant doubts about the results presented in these sections. Section 3.1 presents three forecast methods, chosen seemingly arbitrarily from all those that have been proposed. As far as I can tell, they are all in one way or another linear. For a strongly non-linear system such as the solar dynamo, I see little value in using linear forecasting methods. I argue that applying inappropriate, but complex sounding forecasting techniques projects a sense of accuracy where none is present in reality. The performance of the techniques is discussed in Sect. 3.3 and Fig. 21c. From Fig. 21a I get the impression that the errors in the Phi forecast are comparable (and over some periods exceed by a factor of 2-3, e.g. around 2080-2090) the values of Phi. Around 2200 various methods give Phi of about 100 to 400, and the forecast error is 150 for all methods. This essentially means the range of 0 to 550 (with âĹij600 being the highest value measured during the modern Grand max). In summary, Fig 21 shows that the three methods often give hugely different results (which is not surprising and simply reinforces that solar activity cannot be reliably predicted using such simple techniques and possibly cannot be predicted at all on these time scales). The authors then consider the mean of these three results, claiming that it represents "the most likely level of solar activity". Such an approach can hardly be called scientiiÌĹnËĞA ÌĨc and cannot lead to a "more realistic" forecast than CMIP5. Thus, the mean of 3 more or less random numbers is still a more or less random number of little value. The construction of the "extreme" scenario is also difiÌĹnËĞ A ÌĨcult to follow. A lot of the description is rather opaque. All this seems such a complicated way of computing something that is likely very unreliable anyway and does not provide that much reliable information. For example, the reference scenario seems to be somewhat below present conditions and stays nearly constant at that level, while the extreme scenario drops down to the Maunder minimum and basically stays there. Is that so much different from what was done for CMIP5. I would find it a lot more honest towards the reader to not invoke all these different methods, but rather to make a clear and simple assumption and to show the result it gives. This result may turn out not to be very different from what the authors are proposing now, if the authors make the appropriate assumptions.

[Figure]

BTW, what is the meaning of a negative modulation potential (Fig. 21B) and how is it obtained?

Reply: Let us reply pointwise - Arbitrary: the choice is by no means arbitrary. These techniques are part of the basic set that is advocated for time series prediction e.g. [Brockwell and Davis, Introduction to time series and forecasting. Springer, 2010]. Analogue forecasts have the advantage of not requiring any parametric model. Autoregressive models is by far the most widely used class of models for linear differential equations. And the harmonic model is directly motivated by the evidence for periodicities in the level of solar activity. One could have added analogue neural networks, etc, but then one would be transitioning toward a black box approach. The three models we advocate on the contrary each have their justification. - Linearity: the analogue forecast does not make linearity assumptions. The harmonic one is linear by construction, but can handle both linear and nonlinear systems. Only the autoregressive model explicitly assumes that the system is described by a linear differential equation. - Performance: we do not claim that our models are capable of providing good forecasts, especially since their predictive capacity drops within a few decades (but still does better than the persistence scenario that has been recommended for CMIP5). Probably no model ever will. However, we use these model to get the 'best' possible scenarios with the information at hand. - Taking the mean: the justification is exactly the same as for the averaging of the SSI models: in the absence of objective criteria for preferring one model to the other, the recommendation is to average them all, which provides a mean state of phi. In this particular context, however, since we have prediction skills, we use these to weight the models. - Negative modulation potential: this is a consequence of the prediction methods being unable to provide positive definite values, except for the analogue forecast. Furthermore, these negative phi values are also present in the Steinhilber 2012 phi record, resulting from the modulation potential reconstruction techniques. Therefore, although the negative phi values are unphysical, they are not necessarily caused by the forecast techniques. Also Solanki et al. (2004), McCracken et al. (2007) obtained formally negative phi values in their reconstructions, which are

however, consistent with zero phi within the uncertainties. Strictly speaking, negative phi is not necessarily a physical nonsense but may be related to a poor knowledge of the low-energy part of the local interstellar spectrum (LIS) of cosmic rays outside the heliosphere, which is not very well known.

49) page 39, line 17: ". . . is certainly presentat some level." ". . . is certainly present at some level."

Reply: Thank you for pointing this out.

50) page 39, line 35: ". . . using the geomagnetic reconstruction of the open solar flux Lockwood et al. (2014)." ". . . using the geomagnetic reconstruction of the open solar flux (Lockwood et al., 2014)."

Reply: Thank you for pointing this out.

51) p. 40, line 11-12: "According to solar dynamo models, the solar-cycle averaged modulation potential (and the sunspot number) cannot be predicted more than a few decades ahead." This statement is not entirely consistent with p. 39, line 7-8 "As of today, even predicting the cycle amplitude one cycle ahead remains a major challenge (Pesnell, 2012)." Which statement do the authors actually support, one cycle ahead or multiple cycles ahead? According to Cameron et al. and Jie et al. (referred to earlier in this report; see point 9), the statement on p. 39 appears to be the valid one and the statement on p. 40 should be changed accordingly.

Reply: The text should indeed say "one solar cycle ahead".

52) page 40, line 12-14: "The observed modulation potential has an autocorrelation function that decays exponentially with a characteristic time of 48 $\pm$ 5 years. This quantity can be interpreted as the time beyond which memory is lost." This is weak und unconvincing evidence of memory. Earlier in the same paragraph, the authors state that they are dealing with 22-year averages of the modulation potential Phi. The decay time of 48 years is hence basically the time resolution of the data (based on

the Nyquist frequency). I do not see this as evidence for a memory, just that the true resolution of the data is not very high. In addition to this argument, there may well be hidden connections between data points lying close in time, so that they are not entirely independent. This can be the case in cosmogenic isotope data, so that using this as an argument of memory should be done with considerable care. The authors should first convincingly show that individual data points are completely independent, before making claims of memory.

Reply: Regarding the wording "memory", this is conventionally used for the decay time of the autocorrelation function, though the more technical "decorrelation time" is more accurate.

We agree that the measured decay time is indeed not so much longer than the sampling period. We did discuss the way the Steinhilber (2012) record has been made and found no evidence for a smoothing that may artificially increase the correlation between samples. Note that the relatively larger value of the decay time is also confirmed by the duration of the grand minima/maxima, which always take a finite time to start and to recover.

53) Page 40, line 15-16: "To the best of our knowledge, no existing method has been able to meaningfully predict solar activity more than 60 years ahead." This seems to imply that there are methods that can predict up to 60 years ahead. I would like to hear more about these. E.g. how do the authors know that they work up to 60 years ahead, without waiting for another 60 years to find out? Unless, of course, they are referring to methods that are at least 60 years old and that I seem to have missed. I have seen many so-called predictions tuned to reproduce past data exceptionally well, but then do rather less well when predicting even the next cycle. This statement also is not consistent with p. 39, line 7-8 and other work.

Reply: We should indeed rephrase this more carefully. As of today, very few models are able to shed light on horizons beyond the 22-year timeline. What we meant to say,

is that to the best of our knowledge, that no single realistic approach has gone beyond 60 years.

As you probably know, few of the more elaborate methods have been tested against observations because they rely on space age data.

54) Page 45, line 1: "The resulting SSN time series of both future scenarios have then be used" "The resulting SSN time series of both future scenarios have then been used"

Reply: Thank you for pointing this out.

55) page 53, line 5: NRLTSI2/NRLSSI2 are empirical models not semi-empirical ones.

Reply: This has been corrected.

56) page 54, lines 8-10: the statement is not clear (see above);

Reply: Please see reply to comment 2.

57) page 54, line 13: The statement is too ambiguous. Which satellite measurements are meant?

Reply: This refers to the observations used in the SOLID composite. We shall update the text accordingly.
* * *

---

## Editor Decision (ED1)

Referee's Report on the revised version

Many thanks to the authors for taking many of my numerous suggestions into account and for making substantial changes to the manuscript. It has improved considerably compared with the first version. Many parts of the manuscript, such as that on particle forcings, are well balanced, well written and a pleasure to read. I do not agree with some of the arguments/answers put forward by the authors, but am willing to let them pass in the interests of time. However, there are still two important issues to be clarified/dealt with before the manuscript can be published. There are also a few minor points that the authors may care to look at. Let me begin with the two major issues still present in the revised version of the manuscript.

1. Description of the SOLID composite and Fig. 3: The authors are to be lauded for substantially extending the description of the SOLID composite and for introducing a table describing which data sets have been used. They have now also given a reference to Haberreiter et al. (2017) in which a more detailed description is promised. This is a significant improvement over the first version of the manuscript. All the same, there are still a number of unanswered questions.

For example, the manuscript states that the significant deviations of the SOLID composite from the observations shown in Fig. 3 are explained by the existence of other datasets listed in Table 1, but not shown in Fig. 3. Coincidently, these other, not shown, datasets almost exactly compensate the difference between the data that are plotted and CMIP6. If this is indeed the case, then the authors are to be congratulated, but from the brief description in the paper, it does not become clear how this happens. For instance, the disagreement between SOLID and SORCE/SIM in the 200-400 nm range is explained by averaging with SORCE/SOLSTICE. However, SORCE/SOLSTICE data are only available shortward of 309 nm (Table 1 in the manuscript), whereas much of the flux variability measured by SIM in the 200-400 nm range comes from the 309-400 nm domain not covered by SOLSTICE (see Fig. 1 from Haigh et al. 2010). According to the manuscript this flux should enter SOLID with weight 1 so that it remains unclear why SOLID is so different from the SIM data.

Likely this and other questions that I still have will be reslved by the more detailed description in the manuscript submitted to JGR by Haberreiter et al. (2017), which unfortunately is not accessible. I therefore ask the authors to make this paper available to me via the editor, to enable evaluating the strengths and weaknesses of this approach. The information likely provided there will hopefully quickly clarify the issues and questions I still have regarding the SOLID composite.

2. Future scenarios: Section 3 has improved, but is still substantially weaker than most of the rest of the paper. It also remains misleading, maybe partly because the authors seem to have misunderstood my comments in the last report. I apologize if they were unclear. Let me try to be clearer this time. In spite of the "scenarios" in the title, large parts of Sect. 3 give a false sense that in effect some kind of prediction of future solar activity is being made. True, in the revised version "predict" has been replaced by "forecast" in a few places, but dictionaries (e.g., Collins English dictionary) use the two words synonymously (although they do have somewhat different meanings in meteorology). Also, in spite of these changes in name, a lot of trouble is taken (and quite some text has been added) to argue that the two provided time series are reliable forecasts and not just scenarios.

However, uncertainties in the adopted approach are likely strongly underestimated.

For me the claimed ability to predict or forecast solar activity on longer timescales, although even the amplitude of the next cycle cannot be predicted reliably by dynamo models (or by other means except partly from the strength of the polar fields during the previous minimum), is an assumption, and not a statement of fact, as the current version purports it to be. Assumptions should be clearly identified as such to the reader. On the basis of this assumption and various purely statistical models the authors then "forecast" future solar activity, while the assumption itself remains completely untested in this paper and very poorly tested otherwise. E.g., given the many unknowns affecting the solar dynamo and consequently the very wide variety of dynamo models that have found their way into the recent literature, it may be questioned whether the predictions of any one current dynamo model will stand the test of time.

Even if this basic assumption of Sect. 3 were correct, it is unclear that any of the statistical methods used by the authors, or an ensemble of such methods, will be any good in predicting solar activity in 50, 100 or 300 years. I cannot find a reliable reason for the belief apparently held by the authors that purely statistical "more elaborate reconstruction schemes" will give the correct behavior a number of cycles down the line, while failing already at the next cycle.

The trouble with all the methods used by the authors is that they have not been tested on solar activity on the time scales of interest to the authors (and cannot be properly tested for 50, or 100 years, or even longer periods without waiting that length of time). Therefore, the value of all these extrapolations, be they called "predictions" or "forecasts", is fairly limited.

Take the argument that the Sun has just left a grand maximum and is now likely to stay at a low level of activity and consider the recent revision of the sunspot number. Until a new consensus is reached a number of different sunspot number data sets vie with each other, some of which do not even agree with the statement that the last 6-7 decades was a grand maximum. E.g., according to Svalgaard and Schatten (2016) there have been at 3 episodes since the Maunder minimum, when the activity was approximately as high as in the last 60-70 years. Therefore, the authors possibly have been extrapolating on the basis of wrong assumptions regarding past solar activity. Please note that I am not arguing that Svalgaard and Schatten are correct, but that currently even the sunspot number record is very much under debate, placing a big question mark behind some of the assumptions made here (which, yet again, are sold for a fact). Consequently, the uncertainties in the extrapolations made in the present paper are much larger than the authors acknowledge.

In summary, I am arguing that the "forecasts" or "scenarios" of future activity proposed here are reasonable-looking courses that future activity may possibly take, but they are not more probable than any number of other such scenarios. This is because we do not have reliable, i.e. thoroughly and successfully tested means of forecasting solar activity on the timescales of interest. To be clear, I am not arguing that the authors remove or change their time series (although they show only part of the possible range of solar activity in the upcoming decades, thus restricting the range of solar forcing in CMIP6 climate runs). Rather I am arguing that the authors acknowledge that these are 2 of many equally possible scenarios.

It is necessary to strongly tone down the claims that actual forecasts of activity are presented, removing the impression that the time series being provided are better than many other time series that could be proposed. Ideally, much of the introductory part of Sect. 3, as well as almost the entire Sect. 3.1 should be removed. This could become part of a separate paper in an appropriate journal. In such a paper the

authors could provide more details, much more stringent tests etc. It could allow readers to estimate the reliability of the employed statistical methods when applied to solar data.

However, as this means a major rewrite of this section, I would be satisfied if the authors changed the text to reflect the fact that what they are proposing really are to some extent random scenarios and should not be taken as reliable forecasts or predictions.

Minor issues

1. page 2. lines 1-2: The manuscript still contains the confusing sentence about CMIP6 "statistically indistinguishable from available observations". The answer to the referee explains what the authors mean by this. Such an explanation should also be added to the manuscript to avoid confusing readers

2. page 3. lines 30-31. The sentence it is still not fully correct. Please note that Usoskin et al. (2014) do not discuss TSI reconstruction. Possibly the authors mean reconstructions of solar activity?

3. SATIRE-TS has been converted everywhere to SATIRE, but SATIRE-TS is still mentioned in the legends of Figures 1, 2, 4, 6 (see previous report).

4. page 11, lines 25-28. "In both models, the historical reconstructions are sensitive to the assumptions made when constraining them to direct (satellite era) observations that suffer from large uncertainties. This mainly explains why these models differ before 1990 by an offset." Which assumptions are meant? Why do they lead to an offset? Please explain in the manuscript.

5. page 8. line 16. "a disk-integrated ratio of the core to the-wings of the Mg II emission line at 280 nm". This is not how the Mg II index is defined (a disk integrated ratio is not the same as the ratio of disk integrated quantities); see the paper by Viereck et al. (2001), cited in the same sentence of the revised manuscript. Please correct.

6. Figure 4 may be misleading to some readers, as the sum of the SSI variability of the various models does not seem to be the same (and hence cannot all be equal to the TSI variability). This is likely because the IR is missing. I would propose to (ideally) add the IR band to the figure (as done by Ermolli et al. 2013, although a larger wavelength coverage would be better), or to at least point this out in the text.

---

## Author Response (AR2)

**Response to the reviewer comments on the revised version 15.02.2017**

Dear Dr. Stenke,

we have revised the manuscript according to your and the reviewer comments. Please find our detailed responses below. We hope that the manuscript now fulfills all requirements and can be published.

Thank you very much,

Katja Matthes and Bernd Funke (on behalf of all co-authors)

**Comments by the editor from December 20th 2016:**

Dear Katja Matthes,

your efforts in revising your manuscript according to the referees' comments are very much appreciated. I agree with the referee that the manuscript has clearly improved compared to the previous version. However, there are still some open questions that need clarification before the paper can be finally accepted. Attached I am sending you the referee's comments on the revised version.

Furthermore, I would like to ask you for clarification of the following point: At a project meeting last week there was a talk on solar irradiance observations and data sets. In the presentation it was mentioned that the CMIP6 solar forcing data set is not exactly the mean of the NRLSSI2 and SATIRE data sets as stated in your manuscript, but that somehow modified data sets had been used for the CMIP6 solar forcing data. Could you please comment on this? In case the cited NRLSSI2 and SATIRE data sets had indeed been modified for the CMIP6 input data, the applied modifications have to be documented in the paper. The description of the data sets should be sufficiently complete and precise to allow traceability of methods and results by other scientists.

Yes, it is correct that the NRL and SATIRE datasets in the CMIP6 solar forcing dataset are not identical with the published individual datasets by the two groups. The use of this for CMIP6 modified version of the datasets have been documented, but we have highlighted it even more in the manuscript (in the abstract, the data description section, and in the conclusions). We also added this to the netcdf headers of the published datasets to make sure that there is no confusion. We hope that this has now been sufficiently clarified.

Finally, the editors handling manuscripts on CMIP6 data sets have been asked to make sure that the data are uploaded to ESGF and that the DOIs are cited in the paper before final acceptance. I checked the solarisheppa website, but could not find a link to ESGF or DOIs of the data sets. Please make sure that these information are made available, also in the paper.

Meanwhile the data has been uploaded to ESGF (https://pcmdi.llnl.gov/search/input4mips/). This has been clarified in Section 7 (Data

Availability). Regarding DOIs, the infrastructure that will be issuing DOIs for the CMIP6 contributed data is currently still in development. It is hoped that DOIs will be come available soon.

Best regards, Andrea Stenke

**Referee's Report on the revised version**

Many thanks to the authors for taking many of my numerous suggestions into account and for making substantial changes to the manuscript. It has improved considerably compared with the first version. Many parts of the manuscript, such as that on particle forcings, are well balanced, well written and a pleasure to read. I do not agree with some of the arguments/answers put forward by the authors, but am willing to let them pass in the interests of time. However, there are still two important issues to be clarified/dealt with before the manuscript can be published. There are also a few minor points that the authors may care to look at. Let me begin with the two major issues still present in the revised version of the manuscript.

**1. Description of the SOLID composite** and Fig. 3: The authors are to be lauded for substantially extending the description of the SOLID composite and for introducing a table describing which data sets have been used. They have now also given a reference to Haberreiter et al. (2017) in which a more detailed description is promised. This is a significant improvement over the first version of the manuscript. All the same, there are still a number of unanswered questions.

For example, the manuscript states that the significant deviations of the SOLID composite from the observations shown in Fig. 3 are explained by the existence of other datasets listed in Table 1, but not shown in Fig. 3. Coincidently, these other, not shown, datasets almost exactly compensate the difference between the data that are plotted and CMIP6. If this is indeed the case, then the authors are to be congratulated, but from the brief description in the paper, it does not become clear how this happens. For instance, the disagreement between SOLID and SORCE/SIM in the 200-400 nm range is explained by averaging with SORCE/SOLSTICE. However, SORCE/SOLSTICE data are only available shortward of 309 nm (Table 1 in the manuscript), whereas much of the flux variability measured by SIM in the 200-400 nm range comes from the 309-400 nm domain not covered by SOLSTICE (see Fig. 1 from Haigh et al. 2010). According to the manuscript this flux should enter SOLID with weight 1 so that it remains unclear why SOLID is so different from the SIM data.

Likely this and other questions that I still have will be reslved by the more detailed description in the manuscript submitted to JGR by Haberreiter et al. (2017), which unfortunately is not accessible. I therefore ask the authors to make this paper available to me via the editor, to enable evaluating the strengths and weaknesses of this approach. The information likely provided there will hopefully quickly clarify the issues and questions I still have regarding the SOLID composite.

Thanks for asking for further clarification. We apologize that our previous answer was not complete. The datasets that are available for the 200-400nm spectral range are: NIMBUS7/SBUV (170-399nm), NOAA9/SBUV (170-399nm), NOAA11/SBUV2 (170-399nm), SME/UV (115-302.5nm), SORCE/SIM (240- 2412.3 nm), SORCE/SOLSITCE (115-309nm), UARS/SOLSTICE (115.5 - 410.5 nm) and UARS/SUSIM (115.5 - 410.5 nm). (Note that following the comment of a reviewer, NOAA16/SBUV2 has been taken out). Through the temporal extension of all these datasets, and their subsequent scale-wise

decomposition and recomposition these datasets all enter into the final SOLID composite and moderate the SIM data. So, up to 415 nm, the weight of SIM is smaller than 1. This explains why the SOLID composite is in good agreement with CMIP6. SIM data do get a weight of "1" if no other dataset is available.

We are happy to provide the manuscript for Haberreiter et al. (2017), which hopefully answers the other open questions.

**2. Future scenarios: Section 3** has improved, but is still substantially weaker than most of the rest of the paper. It also remains misleading, maybe partly because the authors seem to have misunderstood my comments in the last report. I apologize if they were unclear. Let me try to be clearer this time. In spite of the "scenarios" in the title, large parts of Sect. 3 give a false sense that in effect some kind of prediction of future solar activity is being made. True, in the revised version "predict" has been replaced by "forecast" in a few places, but dictionaries (e.g., Collins English dictionary) use the two words synonymously (although they do have somewhat different meanings in meteorology). Also, in spite of these changes in name, a lot of trouble is taken (and quite some text has been added) to argue that the two provided time series are reliable forecasts and not just scenarios.

We agree with the reviewer that there was some confusion about the terminology of using "forecast" and "prediction". We have changed the text in section 3 significantly in order to avoid confusion. We provide a scenario for future solar activity, this terminology is perfectly in line with the IPCC terminology for greenhouse gas scenarios. For the scenario construction we use time series analysis, that are forecasts, in order to come up with a likely scenario under the assumption that past statistical behaviour of the Sun can be extrapolated into the future.

However, uncertainties in the adopted approach are likely strongly underestimated. For me the claimed ability to predict or forecast solar activity on longer timescales, although even the amplitude of the next cycle cannot be predicted reliably by dynamo models (or by other means except partly from the strength of the polar fields during the previous minimum), is an assumption, and not a statement of fact, as the current version purports it to be. Assumptions should be clearly identified as such to the reader. On the basis of this assumption and various purely statistical models the authors then "forecast" future solar activity, while the assumption itself remains completely untested in this paper and very poorly tested otherwise. E.g., given the many unknowns affecting the solar dynamo and consequently the very wide variety of dynamo models that have found their way into the recent literature, it may be questioned whether the predictions of any one current dynamo model will stand the test of time.

Even if this basic assumption of Sect. 3 were correct, it is unclear that any of the statistical methods used by the authors, or an ensemble of such methods, will be any good in predicting solar activity in 50, 100 or 300 years. I cannot find a reliable reason for the belief apparently held by the authors that purely statistical "more elaborate reconstruction schemes" will give the correct behavior a number of cycles down the line, while failing already at the next cycle.

The trouble with all the methods used by the authors is that they have not been tested on solar activity on the time scales of interest to the authors (and cannot be properly tested for 50, or 100 years, or even longer periods without waiting that length of time). Therefore, the value of all these extrapolations, be they called

"predictions" or "forecasts", is fairly limited.

Take the argument that the Sun has just left a grand maximum and is now likely to stay at a low level of activity and consider the recent revision of the sunspot number. Until a new consensus is reached a number of different sunspot number data sets vie with each other, some of which do not even agree with the statement that the last 6-7 decades was a grand maximum. E.g., according to Svalgaard and Schatten (2016) there have been at 3 episodes since the Maunder minimum, when the activity was approximately as high as in the last 60-70 years. Therefore, the authors possibly have been extrapolating on the basis of wrong assumptions regarding past solar activity. Please note that I am not arguing that Svalgaard and Schatten are correct, but that currently even the sunspot number record is very much under debate, placing a big question mark behind some of the assumptions made here (which, yet again, are sold for a fact). Consequently, the uncertainties in the extrapolations made in the present paper are much larger than the authors acknowledge.

**We would like to note here that our forecasts are based on Phi rather than SSN.**

In summary, I am arguing that the "forecasts" or "scenarios" of future activity proposed here are reasonable-looking courses that future activity may possibly take, but they are not more probable than any number of other such scenarios. This is because we do not have reliable, i.e. thoroughly and successfully tested means of forecasting solar activity on the timescales of interest. To be clear, I am not arguing that the authors remove or change their time series (although they show only part of the possible range of solar activity in the upcoming decades, thus restricting the range of solar forcing in CMIP6 climate runs). Rather I am arguing that the authors acknowledge that these are 2 of many equally possible scenarios.

It is necessary to strongly tone down the claims that actual forecasts of activity are presented, removing the impression that the time series being provided are better than many other time series that could be proposed. Ideally, much of the introductory part of Sect. 3, as well as almost the entire Sect. 3.1 should be removed. This could become part of a separate paper in an appropriate journal. In such a paper the authors could provide more details, much more stringent tests etc. It could allow readers to estimate the reliability of the employed statistical methods when applied to solar data.

However, as this means a major rewrite of this section, I would be satisfied if the authors changed the text to reflect the fact that what they are proposing really are to some extent random scenarios and should not be taken as reliable forecasts or predictions.

We have updated the text along the lines you suggest. However, we would like to highlight that our scenarios are not just random (which would have made our task much easier). Nor are they predictions. They describe a likely scenario of solar activity.

**Minor issues**

1. page 2. lines 1-2: The manuscript still contains the confusing sentence about CMIP6 "statistically indistinguishable from available observations". The answer to the referee explains what the authors mean by this. Such an explanation should also be added to the manuscript to avoid confusing readers

We removed this confusing sentence in the abstract and expanded that part to be more specific in the respective section where Fig. 1 is described.

2. page 3. lines 30-31. The sentence it is still not fully correct. Please note that

Usoskin et al. (2014) do not discuss TSI reconstruction. Possibly the authors mean reconstructions of solar activity?

We failed to correct this in the previous version by replacing TSI with solar activity, with our apologies.

3. SATIRE-TS has been converted everywhere to SATIRE, but SATIRE-TS is still mentioned in the legends of Figures 1, 2, 4, 6 (see previous report).

**Thank you for pointing this out. This has now been corrected.**

4. page 11, lines 25-28. "In both models, the historical reconstructions are sensitive to the assumptions made when constraining them to direct (satellite era) observations that suffer from large uncertainties. This mainly explains why these models differ before 1990 by an offset." Which assumptions are meant? Why do they lead to an offset? Please explain in the manuscript.

NRLTSI2 has been trained with TSI data from SORCE/TIM whereas SATIRE has mainly used the PMOD composite to assess its performance. This, by itself, may already explain why the two models behave differently before the 1990's. We cut out some text because this is not the main message we want to convey.

5. page 8. line 16. "a disk-integrated ratio of the core to the-wings of the Mg II emission line at 280 nm". This is not how the Mg II index is defined (a disk integrated ratio is not the same as the ratio of disk integrated quantities); see the paper by Viereck et al. (2001), cited in the same sentence of the revised manuscript. Please correct.

**Thank you, we corrected this.**

6. Figure 4 may be misleading to some readers, as the sum of the SSI variability of the various models does not seem to be the same (and hence cannot all be equal to the TSI variability). This is likely because the IR is missing. I would propose to (ideally) add the IR band to the figure (as done by Ermolli et al. 2013, although a larger wavelength coverage would be better), or to at least point this out in the text.

We added the following sentence to the description of Fig. 4 to avoid confusion: "Please note that the sum of the SSI variability of the various models is not equal to the TSI variability since the IR part is missing in Fig. 4."

**Solar Forcing for CMIP6 (v3.2)**

Katia Matthes1,2, Bernd Funke3, Monika E. Andersson18, Luke Barnard4, Jürg Beer5, Paul Charbonneau6, Mark A. Clilverd7, Thierry Dudok de Wit8, Margit Haberreiter9, Aaron Hendry14, Charles H. Jackman10, Matthieu Kretzschmar8, Tim Kruschke1, Markus Kunze11, Ulrike Langematz11, Daniel R. Marsh19, Amanda C. Maycock12, Stergios Misios13, Craig J. Rodger14, Adam A. Scaife15, Annika Seppälä18, Ming Shangguan1, Miriam Sinnhuber16, Kleareti Tourpali13, Ilya Usoskin17, Max van de Kamp18, Pekka T. Verronen18, and Stefan Versick16 1GEOMAR Helmholtz Centre for Ocean Research Kiel, Kiel, Germany 2Christian-Albrechts Universität zu Kiel, Kiel, Germany 3Instituto de Astrofísica de Andalucía (CSIC), Granada, Spain 4University of Reading, Reading, United Kingdom 5EAWAG, Dübendorf, Switzerland 6University of Montreal, Canada 7British Antarctic Survey (NERC), Cambridge, UK 8LPC2E, CNRS and University of Orléans, France 9Physikalisch-
[revised manuscript text omitted]

---

## Author Response (AR3)

**Additional Response to the reviewer comments on the revised version 16.03.2017**

Dear Dr. Stenke, dear Reviewers,

we are very sorry for this additional extra revision round.

We were contacted by the groups who delivered the input datasets for the CMIP6 dataset to modify/clarify the description of how the original SATIRE and NRL datasets differ from the CMIP6-adapted datasets, In particular different sunspot datasets have been used for the original and the CMIP6-adapted SATIRE dataset which affect the long-term trend in the pre-satellite era. We've adapted the description in the paper again and all changes can be seen in the track changes version which we provide with this reply.

We feel that this should resolve the issues raised by the reviewer in the second round.

We therefore kindly ask you to add this extra addition to the current revision cycle.

Thank you very very much for your understanding,

Katja Matthes and Bernd Funke (on behalf of all co-authors)

[revised manuscript text omitted]

---

## Author Response (AR4)

**Reply to Editor Review from 19 April 2017, 28.04.2017**

**Topical Editor Decision: Publish subject to minor revisions (Editor review)** (19 Apr 2017) by Andrea Stenke
Comments to the Author:

Dear Katja Matthes and Coauthors,

your efforts on addressing the reviewer comments and changing the manuscript accordingly are very much appreciated. The latest version of the manuscript has clearly improved. I think the paper is now in publishable state and the discussion should be continued within the scientific community.

A reviewer asked for a few additional, minor revisions. As soon as the points listed below have been taken into account, I will accept the manuscript for publication.

Best regards,
Andrea Stenke

Dear Dr. Stenke,

we have addressed all remaining minor comments from the referees as detailed below and in our attached highlighted manuscript and hope that the paper is now ready for publication.

Thank you very much and we are looking forward to your reply,

Katja Matthes and Bernd Funke (on behalf of all coauthors)

Referee comments:
The line numbers mentioned below refer to the manuscript version of 160317.

- On lines 6-8 of P. 11 the authors mention that that since 2003 there are no SSI measurements in the visible and near-UV except for SIM. This is not correct. SOLSPEC data are published -http://adsabs.harvard.edu/abs/2016SoPh..291.3527M There are also various filter radiometers that cover this period of time.

We have reformulated the sentence as follows: „After 2003, the only remaining observations are from SORCE/SIM...." changed to „After 2003, the only remaining observations that are continuous are from SORCE/SIM....."

- Fig. 3: it is still not clear why the SOLID time series departs so strongly from SOLSTICE at 120-200nm during recent years when SOLSTICE was the only instrument providing data at these wavelengths whose data were used.

We have reproduced Figure 3 and adjusted the figure caption to make this point clear. The figure caption now reads as follows: „CMIP6 recommended SSI time series (black) from 1980 to 2015 together with the SOLID data composite (green) and relevant instrument observations for the following wavelength bins: 120nm--

200nm (left), 200nm--400nm (right). The SOLID and instrument time series have been adjusted to match the average level of the CMIP6 time series. Note that the longest wavelength observed by TIMED/SEE is 189nm, the longest observed wavelength by SORCE/SOLSTICE is 309nm and the shortest observed wavelength by SORCE/SIM is 240nm. All time series are running averages over 2 years."

All individual datasets contributing to SOLID in the 120-200 nm range have been incorporated into Fig. 3.
The TIMED/SEE data was forgotten in the figure only, but were considered in the SOLID composite, taking into account that the TIMED/SEE degradation after about 2010 is underestimated (Del Zanna & Andretta, A&A, 2015).

- Sect. 3: This remains the weakest section of the paper.
Lines 21-24 of p. 42: Vague "memory effects" of unknown origin are still invoked to argue for the approach taken by the authors. No references are provided for these effects (except later mentioning that most dynamo models show a memory of a few cycles). Please provide references.

We now provide several references (see a list of references below) and have adjusted the text accordingly. The following paragraph of the previous version

Most models exhibit some persistence in the solar cycle-averaged level of activity, with a memory of up to a few cycles. However, among models that do succeed in producing deep activity minima similar to the Maunder minimum, most show onsets occurring surprisingly fast, typically within one or two cycles. At present these models are still not detailed enough to warrant their use in producing physics-based forecasts. The second approach consists in making a probabilistic statement about future solar activity based on present conditions and by learning from past variations of solar activity.

has been replaced with

In stochastically-forced kinematic dynamo models, persistence in the solar cycle-averaged level of activity (hereafter ``memory") can extend from less than one and up to three cycles, depending on details of the models and of the physical parameter regime in which they operate \citep[e.g.][]{stjean07,yeates08,cameron13,munoz-jaramillo13}. In non-kinematic dynamo models incorporating the magnetic backreaction on large-scale inductive flows, deterministic modulation of the primary cycle amplitude can be produced, amounting to a form of memory that can extend over tens of activity cycles in a wide range of parameter regimes \citep[e.g.][]{tobias97,bushby06}. Among models that do succeed in producing deep activity minima similar to the Maunder minimum, most show onsets occurring surprisingly fast, typically within one or two cycles. At present these models are still not detailed enough to warrant their use in producing physics-based forecasts.

The second approach consists in making a probabilistic statement about future solar activity based on present conditions and by learning from past variations of solar activity.

- I also noticed that the authors still do not cite the work of Cameron and Schussler, which provides the clearest evidence and explanation yet that predictions beyond half a solar cycle are going to be extremely difficult. I quote from my first report: "The solar dynamo is chaotic, possibly even stochastic and it is even difficult to predict the next solar cycle much before the previous minimum. Going beyond the next cycle is even tougher. See e.g.
* J. Jiang, R.H. Cameron, M. Schüssler, 2015: The cause of the weak solar cycle 24; ApJL 808 L28
* Cameron et al. 2013: Limits to solar cycle predictability: Cross-equatorial flux plumes, A&A 557, A141"

As mentioned above, we have added a list of references, also striving to give a balanced view of the question. The model used by R. H. Cameron indeed has no memory beyond one cycle, while some others do.

- Various places in Sect. 3: The word "realistic" is often used when mentioning the scenarios of future solar activity (e.g. pages 42, 43, 47, 50). This should be replaced by a more appropriate and indeed realistic expression, such as "reasonable" or "statistically appropriate".

This suggestion has been taken into account and realistic has been replaced appropriately.

References added:

[revised manuscript text omitted]